# Inherent fast inactivation particle of Nav channels as a new binding site for a neurotoxin

Xi Zhou [1,2,3,7] ✉, Haiyi Chen [4,5,7], Shuijiao Peng[1,2,3,6], Yuxin Si[1,2,3], Gaoang Wang [4], Li Yang[1,2,3], Qing Zhou[1,2,3], Minjuan Lu[1,2,3], Qiaoling Xie[1,2,3], Xi He[1,2,3], Meijing Wu[1,2,3], Xin Xiao[1,2,3], Xiaoqing Luo[1,2,3], Xujun Feng[1,2,3], Wenxing Wang[1,2,3], Sen Luo[1,2,3], Yaqi Li[1,2,3], Jiaxin Qin[1,2,3], Minzhi Chen[1,2,3], Qianqian Zhang[1,2,3], Weijun Hu[1,2,3], Songping Liang[1,2,3], Tingjun Hou [4] ✉ & Zhonghua Liu [1,2,3] ✉

## Abstract

**Neurotoxins derived from animal venoms are indispensable tools for probing the structure and function of voltage-gated sodium (Nav) channels. Utilizing a novel centipede peptide toxin called rpTx1, we show that the "inherent inactivation particle" of Nav channels represents a binding site for a neurotoxin. The toxin comprises two functional domains: one for cell penetration and one for modulating Nav channel activity. After crossing the cell membrane, rpTx1 preferentially binds to and stabilizes the IFMT motif (the conserved core region of the fast inactivation particle in mammalian Nav channels) in the unbound state, preventing this motif from associating with its receptor site and thereby inhibiting the fast inactivation of Nav channels. This competition between rpTx1 and the receptor site for interacting with the IFMT motif may account for the higher activity of rpTx1 on Nav1.8 than on other Nav subtypes, given the weaker relative affinity between the receptor site and the IFMT motif of Nav1.8. Overall, this study should promote the investigation of the intracellular modulation of Nav channels by neurotoxins.**

**Keywords** Sodium Channels; Peptide Toxins; Fast Inactivation Gate; Neurotoxins Binding Sites; Intracellular Modulation
**Subject Categories** Neuroscience; Pharmacology & Drug Discovery

## Introduction

Voltage-gated sodium (Nav) channels play a key role in membrane excitation, which is essential for the generation and propagation of action potentials in many excitable tissues, such as nerves and muscles (Catterall, 2012; Hodgkin and Huxley, 1945). To date, nine Nav channels have been cloned and characterized in mammals.

These channels have distinct biophysical and pharmacological properties, although they share high sequence similarity (Catterall, 2012). In humans, mutations of Nav channels are associated with various channelopathies, such as pain, epilepsy, and cardiac and muscle paralysis syndromes (Dib-Hajj and Waxman, 2019; Huang et al, 2017; Mantegazza et al, 2021). The pore-forming subunit of mammalian Nav channels consists of four asymmetric homologous repeat domains (DI–IV), each domain containing six transmembrane segments (S1 to S6) (Fig. EV1). The S5 and S6 segments from each domain form a central pore domain (PD) for Na⁺ conductance, which is surrounded by four voltage-sensing domains (VSD I–IV) formed by S1 to S4 (Catterall, 2000). The process of Nav channel pore opening involves a mechanism of electromechanical coupling. Upon membrane depolarization, the outward and rotational movements of the S4 segments (gating charges) appear to couple through intracellular S4–S5 linkers to pull S6 outward, resulting in iris-like dilation to initiate ionic conductance (Catterall et al, 2020; Yan et al, 2017). According to the asynchronous gating model, the outward movements of VSD I–III are primarily responsible for channel activation, while the activation of VSD IV is involved in the channel's activation and fast inactivation (Ahern et al, 2016; Armstrong, 2006; Goldschen-Ohm et al, 2013). Fast inactivation is a hallmark of mammalian Nav channel kinetics and is critical for the repetitive firing of action potentials (Catterall, 2012; Hodgkin and Huxley, 1952). Decades of characterization have established that the short intracellular linker between DIII and DIV (DIII–DIV linker) is a key element for fast inactivation (Vassilev et al, 1988). The key hydrophobic motif Ile-Phe-Met-Thr (the IFMT motif), located in the N-terminal of this linker, is defined as the inactivation particle (Fig. EV1) (Catterall, 2012; West et al, 1992). To date, the mechanism of the development and electromechanical coupling of fast inactivation is not entirely clear.

Many venomous animals, plants, and microbes have evolved neurotoxins that modulate the activities of Nav channels to subdue their prey and deter predators (Gilding et al, 2020; Klint et al, 2012; War et al, 2012; Zancolli and Casewell, 2020). These neurotoxins affect Na⁺ conductance through two main mechanisms: (1) acting

[1]The National and Local Joint Engineering Laboratory of Animal Peptide Drug Development, College of Life Sciences, Hunan Normal University, Changsha 410081, China. [2]Peptide and Small Molecule Drug R&D Platform, Furong Laboratory, Hunan Normal University, Changsha 410081 Hunan, China. [3]Institute of Interdisciplinary Studies, Hunan Normal University, Changsha 410081, China. [4]Innovation Institute for Artificial Intelligence in Medicine of Zhejiang University, Hangzhou, Zhejiang 310058, China. [5]School of Pharmacy, Hangzhou Normal University, Hangzhou, Zhejiang 311121, China. [6]Hunan Provincial Key Laboratory of Microbial Molecular Biology, Hunan Provincial Center for Disease Control and Prevention, Changsha 410000, China. [7]These authors contributed equally: Xi Zhou, Haiyi Chen. ✉E-mail: xizh@hunnu.edu.cn; tingjunhou@zju.edu.cn; liuzh@hunnu.edu.cn

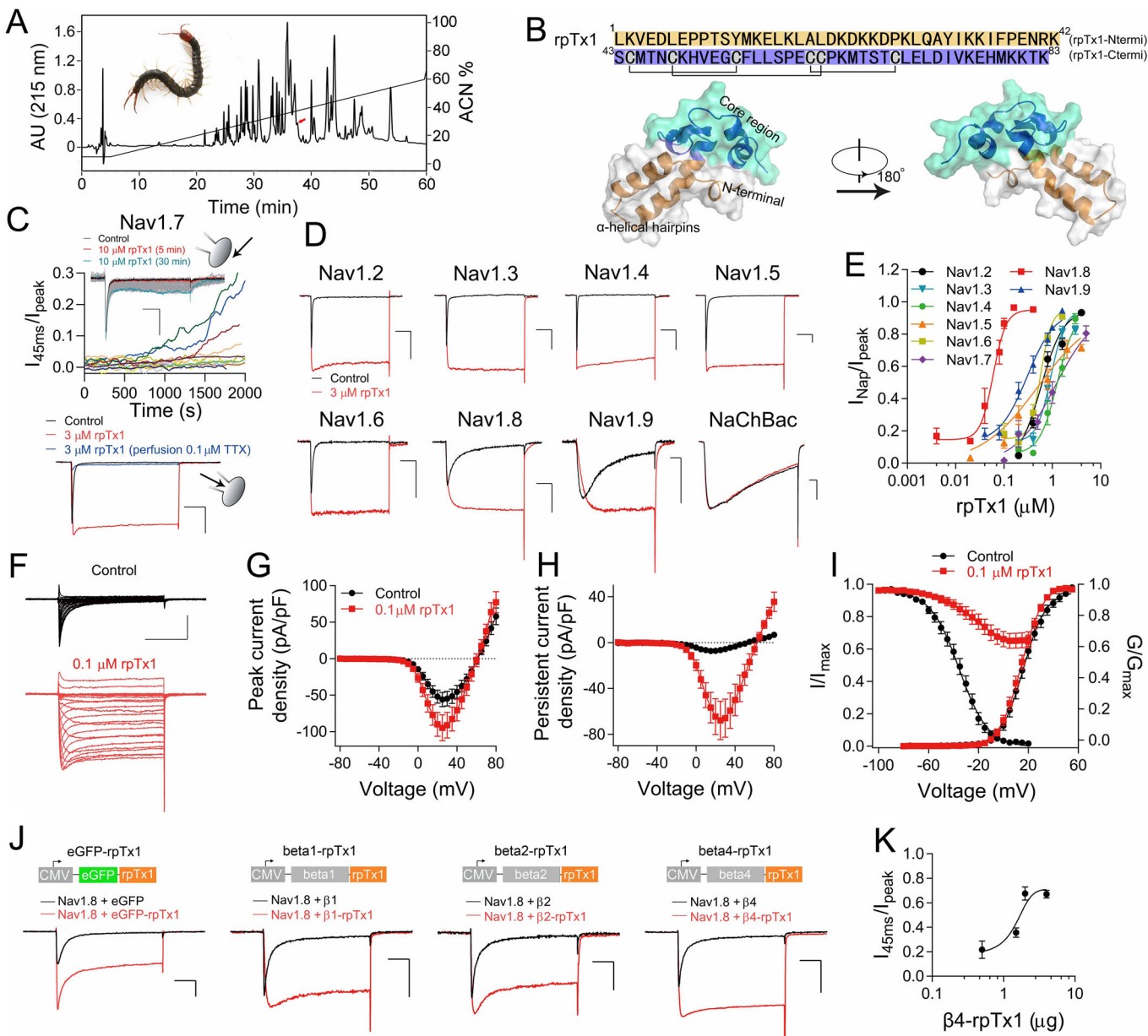

as blockers to physically occlude the pore, exemplified by the small-molecule neurotoxins tetrodotoxin (TTX) and saxitoxin (STX) and some peptide toxins, namely, μ-conotoxins (Green et al, 2014; Kao, 1966; Mosher et al, 1964) or (2) serving as gating modifiers to modulate the channel activity by an allosterically coupled mechanism, usually binding to VSDs to alter the equilibrium among the open, closed, and inactivated states (Catterall et al, 2007). Furthermore, it has also been found that venom peptide toxins can enter cells to bind to the intracellular regions of ion channels or ion channels located on organelles, such as scorpion toxins WaTx targeting TRPA1 (Lin King et al, 2019), and Maurocalcine, Hadrucalcin and IpTxa acting on ryanodine receptor type 1 (RyR1) (Estève et al, 2005; Gurrola et al, 2010; Schwartz et al, 2009). Therefore, neurotoxins have proven to be useful pharmacological probes for Nav channel structure and function investigation due to their action specificity. To date, at least nine distinct

neurotoxin/drug receptor sites have been identified on Nav channels, located in voltage sensors or in/around the pore region (Fig. EV1) but not in intracellular regions (Catterall et al, 2007; Klint et al, 2012; Luo et al, 2024; Stevens et al, 2011; Wang et al, 2025). In particular, because of the physiochemical characteristics of peptide toxins, their binding sites (namely, sites 1, 3, 4, and 6) are distributed around the extracellular regions or the outer vestibule of the pore of Nav channels (Fig. EV1). On the other hand, intracellular domains of Nav channels continue to be interesting sites and are crucial for channel modulation and the regulation of gating processes, including membrane traffic, activation and the fast inactivation gate (Catterall, 2012; Nathan et al, 2021; Oelstrom et al, 2014; Sharkey et al, 2009; Zhang et al, 2008). Notably, dozens of mutations targeting intracellular activation or the fast inactivation gate influence the relevant gating kinetics to cause gain-of-function phenotypes in the clinic

**Figure 1.    The centipede venom peptide toxin rpTx1 inhibits the fast inactivation of Nav channels inside cells.**

(A) The $C_{18}$ RP-HPLC profile of ~3 mg crude venom which was collected from the centipede *S. subspinipes mutilans* by electrical stimulation. The red arrow indicates the fraction containing rpTx1, which accounts for approximately 1% of the crude venom. Inset: a photo of *S. subspinipes mutilans* by X.Z. (B) Primary structure of rpTx1 (upper). Black lines show the disulfide linkage. Surface and cartoon of rpTx1 (lower). The structure was predicted by the AlphaFold2. (C) Upper panel: the time course of rpTx1 (10 μM) acting on hNav1.7 following extracellular application. Varied line colors indicate recordings from different cells (*n* = 11). The inset shows representative current traces exhibiting an rpTx1-induced response following 30 min of extracellular treatment with 10 μM rpTx1. These currents were elicited by a series 50-ms depolarization pulses to 0 mV from the holding potential of −90 mV at a frequency of 0.1 Hz. Lower panel: the fast inactivation of hNav1.7 was removed by intracellular application of rpTx1, and the rpTx1-evoked currents were completely inhibited by extracellular perfusion 0.1 μM TTX (*n* = 8). Scale bar: 1 nA, 10 ms. (D) Representative current traces show intracellular application of 3 μM rpTx1 completely removed the fast inactivation of mammalian Nav channels, but did not affect bacterial Nav channel currents (NaChBac) (*n* = 4–10). Scale bar: 1 nA, 10 ms. (E) Concentration–response inhibitory curves show the effect of intracellular application of rpTx1 on the fast inactivation of Nav1.2-1.9. The current (persistent currents) at the time point of 45-ms (Nav1.2-1.8) or 95-ms (hNav1.9) of the depolarization pulse was normalized to the corresponding peak current to quantify the magnitude of rpTx1's effect (*n* = 3–10 per concentration). (F) Representative traces of current families were recorded from ND7/23 cells expressing rNav1.8 in the absence or presence of 0.1 μM rpTx1 in intracellular solution. The cell was held at −90 mV (*n* = 16 for control, *n* = 18 for rpTx1). Scale bar: 1 nA, 20 ms. (G, H) Current density and voltage relationships of the peak currents (G) and the persistent currents (H) of rNav1.8 before (*n* = 16) and after the intracellular treatment of 0.1 μM rpTx1 (*n* = 18). (I) The effect of rpTx1 on the steady-state activation (*n* = 16 for control, *n* = 18 for rpTx1) and inactivation (*n* = 20 for control, *n* = 18 for rpTx1) of rNav1.8. (J) Inward currents elicited by 50-ms voltage steps from −90 mV to 20 mV in ND7/23 cells expressing rNav1.8 co-expressed with eGFP or eGFP-rpTx1 (*n* = 5), β1 or β1-rpTx1 (*n* = 5), β2 or β2-rpTx1 (*n* = 5), or β4 or β4-rpTx1 (*n* = 5). Scale bar: 1 nA, 10 ms. (K) Concentration–response curves of the transfected β4-rpTx1 plasmid inhibiting the fast inactivation of rNav1.8 (*n* = 3 per concentration). The horizontal axis shows the dosage of the β4-rpTx1 plasmid used during cell transfection. Data are presented as mean ± S.E.M. \**p* < 0.05, \*\**p* < 0.01. Source data are available online for this figure.

(Huang et al, 2017). Neurotoxins targeting intracellular domains would be indispensable probes to study the action and regulation of these domains.

In this study, we discovered a unique peptide toxin, rpTx1, from a centipede venom. It penetrates the cell membrane and completely inhibits the fast inactivation of Nav channels on the intracellular side by directly binding to the IFMT motif. Thus, rpTx1 may be a valuable tool to probe the underlying mechanism of the inactivation gate of Nav channels. Here, we employed a multifaceted approach encompassing electrophysiology, fluorescence imaging, mutagenesis, surface plasmon resonance, mutant cycle analysis, and molecular dynamics simulations to elucidate the molecular basis of the toxin and Nav channel interaction.

## Results

### RpTx1 intracellularly inhibits the fast inactivation of mammalian Nav channels

While screening peptide toxins acting on Nav channels, we discovered an RP-HPLC fraction (Fig. 1A) from the venom of the Chinese red-headed centipede *Scolopendra subspinipes mutilans* that completely inhibited the fast inactivation of human Nav1.7 (hNav1.7) currents in a unique manner (see below). This fraction could be further purified by cation-exchange HPLC and analytical RP-HPLC to yield a single active peptide (Fig. EV2A,B). Its molecular weight was 9625.4 Da as measured by ESI mass spectrometry (Fig. EV2C). Its full sequence was determined by N-terminal Edman degradation sequencing and 3'-RACE (Fig. 1B, upper panel, Appendix). Initially, Edman degradation sequencing yielded the N-terminal 20 amino acid sequence: LKVE-DLEPPTSYMKELKLAL (Appendix). Subsequently, based on the N-terminal sequence VEDLEPPT, degenerate primers (5'-GTNGAR-GAYYTNGARCCNCCNAC-3', where N = A, C, G, or T; Y = C or T; R = A or G) were designed for 3' RACE. PCR and DNA sequencing were then used to obtain the gene sequence and amino acid sequence of the toxin, which contains 83 amino acid residues, and we named it rpTx1 (Fig. 1B, upper panel, Appendix). According to a BLAST search, rpTx1 exhibits 35–73% sequence identities with several peptide toxins

from centipedes. Among these toxins, Ssd1a and Ssd1b are isolated from the venom of *Scolopendra subspinipes dehaani* (Liu et al, 2012), and the other are derived from the transcriptome analysis of venom glands of *Scolopendra morsitans* (Sm1a, Sm2a and Sm3a), *Cormocephalus westwoodi* (Cw1a and Cw2a), *Scolopendra alternans* (Sa2a), and *Ethmostigmus rubripes* (Er1a) (Undheim et al, 2014). Their functions remain unknown. The sequence comparison shows they share the same sequence architecture, composing of the long flexible N-terminal sequences and the C-terminal cysteine-rich sequences, and the arrangement patterns of cysteine residues are identical (Fig. EV2D). This result indicates that these toxins may be conserved in centipede venoms. Based on the BLAST results, its function described below, and the rational nomenclature rules introduced by King et al (King et al, 2008), the toxin is also named δ-SLPTX$_{10}$-Ssm1a. A high-confidence three-dimensional structure of rpTx1 was predicted by AlphaFold2 (Jumper et al, 2021; Varadi et al, 2022) (Fig. 1B, lower panel, Fig. EV2E). The sequence and structure analysis showed that the structure of rpTx1 could be separated into two domains: the N-terminal domain (rpTx1-Ntermi, with amino acid residues 1–42) and the C-terminal domain (rpTx1-Ctermi, with amino acid residues 43–83) (Fig. 1B). RpTx1-Ntermi is enriched with lysine residues and forms α-helical hairpins, and rpTx1-Ctermi adopts a compact structure stabilized by three disulfide bonds, which is commonly found in animal venom peptide neurotoxins. In order to determine the disulfide linkage of rpTx1, the truncated rpTx1 with short N-terminal was yielded by expression in *Escherichia coli*. Partial reduction of the truncated rpTx1 with Tris (2-carboxyethyl) phosphine (TCEP) resulted in the obtaining of partially reduced isomers through RP-HPLC purfication (Fig. EV3A). These isomers were subsequently alkylated with iodoacetamide and then subjected to Edman degradation, demonstrating the disulfide linkage of rpTx1: C1–C3, C2–C5, and C4–C6 (the numbers represent the relative positions of the cysteine residues in the peptide sequences) (Fig. EV3B,C), which is consistent with the disulfide connectivity in the Alphafold2 predicted structure.

Previously reported peptide toxins usually modulate the function of Nav channels by acting on the extracellular regions, and the action often displays fast dynamics. However, rpTx1 seems to have no such property. In our patch-clamp recording experiments, when 10 μM rpTx1 was added to the recording chamber, we

did not observe any change in the hNav1.7 currents within 5 min. However, when the treatment time was extended to more than 10 min, it was observed that the fast inactivation of hNav1.7 was gradually inhibited in some cells (4 of 11 cells) (Fig. 1C, upper panel). We speculated that rpTx1 did not directly target the hNav1.7 channel or mediate its effect intracellularly rather than extracellularly. Indeed, when rpTx1 was added directly to the intracellular solution by pipette, it rapidly removed the fast inactivation and caused a persistent current of hNav1.7 in a concentration-dependent manner. The rpTx1-induced persistent currents were inhibited by tetrodotoxin (TTX), a central pore blocker of TTX-S Nav channels (Fig. 1C, lower panel), indicating that the non-inactivated currents were mediated by rpTx1-treated hNav1.7 rather than leak currents or other ion channel currents. These data suggested that rpTx1 might cross cell membranes and inhibit the fast inactivation of hNav1.7 on the intracellular side.

In addition, we found that rpTx1 potently inhibited the fast inactivation of the mammalian Nav channels Nav1.2-1.9 but not the bacterial Nav channel NaChBac heterologously expressed in HEK293T or ND7/23 cells (Fig. 1D). Interestingly, rpTx1 preferentially inhibited the fast inactivation of two channels upon intracellular application, rat Nav1.8 (rNav1.8) (half-maximum effective concentration ($EC_{50}$) = $59.8 \pm 5.6$ nM, data are presented as mean $\pm$ S.E.M) and hNav1.9 ($EC_{50}$ = $301.7 \pm 58.9$ nM), both of which have far slower fast inactivation than Nav1.1-1.7 (Fig. 1E). RpTx1 has more than 10-fold higher potency for rNav1.8 over the Nav1.2-1.7 subtypes. As a further test of toxin specificity, rpTx1 produced neither activation nor persistent inhibition when applied intracellularly to other ion channels, including voltage-gated potassium channels (mKv1.3, rKv2.1, rKv4.2, and hKCNQ2) and calcium channels (mCav1.2, rCav1.3, and rCav3.1) (Fig. EV4A).

Intracellular application of rpTx1 suppressed the fast inactivation currents of rNav1.8 at all tested voltages and enhanced the peak and persistent currents (Fig. 1F–H), but it did not change the threshold of the initial activation voltage or the reversal potential of the rNav1.8 currents (Fig. 1G). In addition, rpTx1 did not alter the voltage dependence of activation but significantly shifted the steady-state inactivation curve to a positive direction by approximately 14 mV, and it caused a largely non-inactivated component in the steady-state inactivation curve around the test potentials (Fig. 1I and Table EV1). Due to the removal of fast inactivation, no components of fast inactivation recovery were observed (Fig. EV4B). Similar results were found for other Nav channels, such as rNav1.4 and hNav1.7, but the activation curve of hNav1.7 was moderately shifted to the hyperpolarization direction by rpTx1 (Fig. EV4C,D and Table EV1). Generally, rpTx1 has a similar effect on Nav channels to some site 3 toxins which inhibit the fast inactivation of Nav channel currents by binding to the DIV extracellular S3-S4 linker (Catterall et al, 2007; Stevens et al, 2011; Zhou et al, 2020).

The production of recombinant rpTx1 by overexpression in *Escherichia coli* yielded a single dominant isomer (Fig. EV5A–C). The application of recombinant rpTx1 to rNav1.8 likewise abolished fast inactivation with an $EC_{50}$ value of $61.8 \pm 10.5$ nM (Fig. EV5D,E). Recombinant rpTx1 fully recapitulated the activity of the native species, further validating rpTx1 as an active component in the venom. All subsequent experiments were performed with recombinant rpTx1 unless otherwise stated.

To further confirm the effect of rpTx1 on Nav channels on the intracellular side, we generated two types of fusion proteins by covalently linking the peptide to the C-terminal of eGFP (eGFP-rpTx1) or Nav channel β-subunits (β1/2/4-rpTx1) by the linker RILENLYFQG (Fig. 1J). By this fusion expression, rpTx1 could be expressed intracellularly (Fig. EV6). Unsurprisingly, we observed a persistent current when coexpressing Nav1.8 and eGFP-rpTx1 or β1/2/4-rpTx1 in ND7/23 cells (Fig. 1J), similar to the effect when rpTx1 was added in the pipette. In addition, these effects were positively correlated with the amounts of transfected plasmids (Fig. 1K). Considering rpTx1 added intracellularly could possibly pass through the cell membrane into the extracellular space, we cannot exclude the possibility that intracellular rpTx1 exits into the extracellular space and exerts its effect of inhibiting the fast inactivation of Nav channels.

## RpTx1 contains two functional domains

We next asked whether rpTx1 could cross the plasma membrane barrier. Confocal imaging was used to visualize the cell permeability of fluorescein isothiocyanate (FITC)-labeled rpTx1 (rpTx1-FITC). HEK293T cells were incubated with rpTx1-FITC (5 μM) for 30 min, and approximately 29.8% (50 of 168) of the cells were FITC-labeled, with the green fluorescence localized in the cytoplasm, suggesting that rpTx1 was successfully internalized by cells (Fig. 2A). As mentioned above, the N-terminal of rpTx1 is enriched with lysine residues and might be important for the cell-penetrating ability of rpTx1. When the cluster of the eight lysine residues in the N-terminal was replaced with alanine residues (rpTx1-8K/A), this mutant significantly reduced the efficiency of cellular uptake compared to rpTx1, resulting in 8.0% of the cells (18 of 225 cells) being FITC-labeled (Fig. 2A), but it maintained the ability to inhibit the fast inactivation of rNav1.8 when administered intracellularly (Figs. 2B and EV7A,B). If rNav1.8-expressing ND7/23 cells were incubated with either rpTx1 or rpTx1-8K/A for 30 min, the response of the cells to rpTx1 was found to be concentration-dependent. Specifically, 30.6% (19 of 62 cells) exhibited a response in the 5 μM rpTx1 treatment group, whereas in the 1 μM rpTx1 treatment group, this response rate decreased to 18.6% (11 of 59 cells) (Fig. 2C). However, in the 5 μM rpTx1-8K/A treatment group, only few cells (8.3%, 5 of 60 cells) exhibited the inhibition of Nav1.8's fast inactivation (Fig. 2C). These results are consistent with the confocal images and suggest that rpTx1 can cross the plasma membrane barrier to enter the cell, possibly depended on its N-terminal region. In order to validate this, two truncated peptides of rpTx1, rpTx1-Ntermi and rpTx1-Ctermi were further synthesized and analyzed. As shown in Fig. 2A, by using the same treatment method as rpTx1-FITC, FITC-labeled rpTx1-Ntermi was capable of entering HEK293T cells on its own (27.2% for rpTx1-Ntermi vs 29.8% for rpTx1, $p = 0.85$). However, rpTx1-Ntermi had no effect on rNav1.8 currents upon either intracellular or extracellular administration (Fig. 2D). In contrast, intracellular but not extracellular application of rpTx1-Ctermi significantly inhibited the fast inactivation of rNav1.8 (Fig. 2E), suggesting that rpTx1-Ctermi itself does not have the ability to enter cells, and the C-terminal of rpTx1 might be required for the peptide action on Nav channels.

To further explore critical residues for rpTx1's activity toward Nav channels, alanine scanning was conducted. We observed that three amino acid residue mutations significantly decreased the peptide's activity on rNav1.8 (Fig. EV7C). The $EC_{50}$ values of the

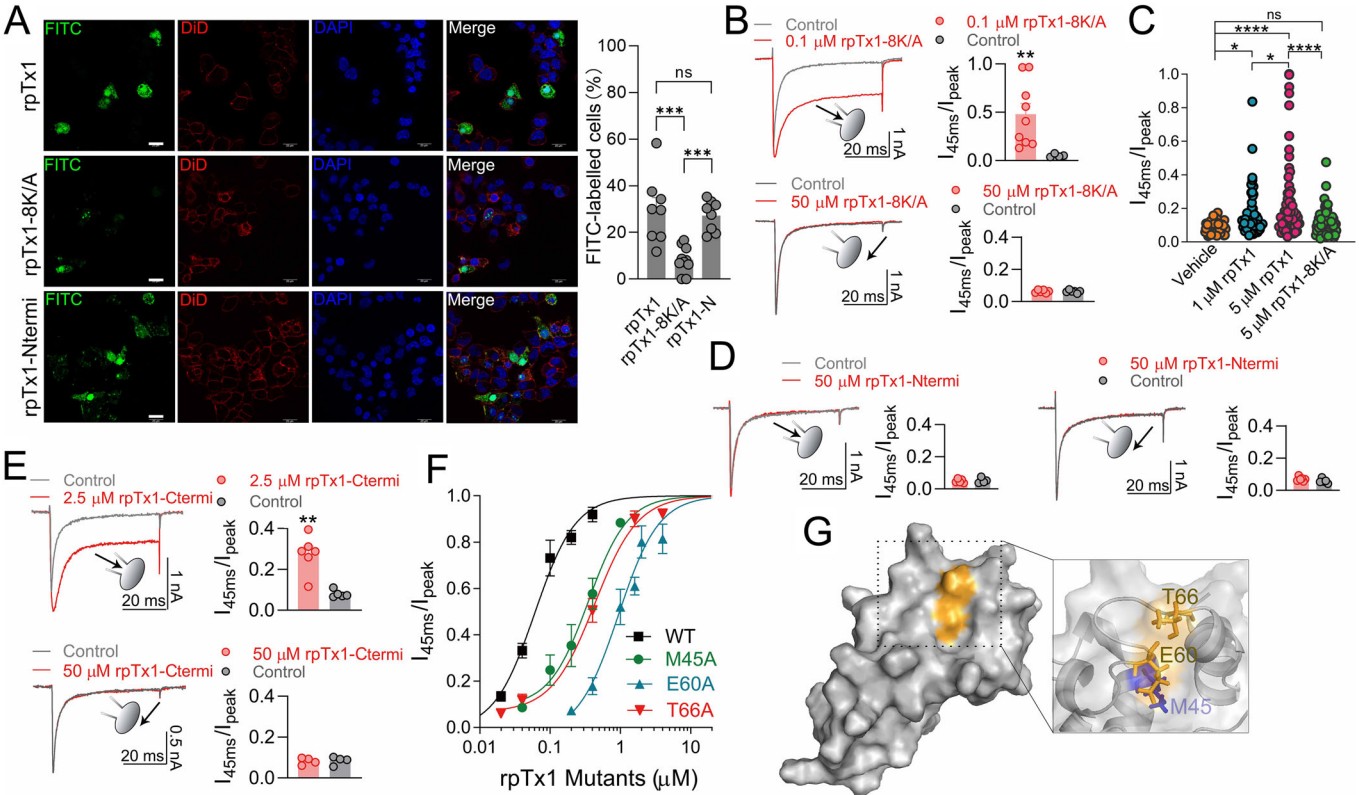

**Figure 2. RpTx1 contains two functional domains.**

(A) Left panel: confocal images of HEK293T cells treated with 5 μM FITC-labeled rpTx1, rpTx1-8K/A or rpTx1-Ntermi for 30 min. Cell membranes and nuclei were labeled by DiD or DAPI, respectively. Scale bar: 20 μm ($n = 3$ independent experiments). Right panel: quantitative analysis of FITC-positive cells in the confocal images in the left panel (one-way ANOVA with post hoc analysis using the Dunnett's multiple comparisons test, $n = 8$–10) (rpTx1 vs rpTx1-8K/A, $p = 0.0002$; rpTx1 vs rpTx1-Ntermi, $p = 0.8543$; rpTx1-Ntermi vs rpTx1-8K/A, $p = 0.0009$). (B) Representative current traces and scatter dot plot showing the effect of rpTx1-8K/A on rNav1.8 expressed in ND7/23 cells upon intracellular (upper) ($n = 6$, non-parametric Mann–Whitney two-tailed test, $p = 0.0028$) or extracellular application (lower) ($n = 5$). (C) Comparison of the persistent currents of rNav1.8-expressing ND7/23 cells incubated with or without rpTx1 or rpTx1-8K/A. The cells were pre-treated with peptides for 30 min at 25 °C ($n = 63$ for vehicle, $n = 59$ for 1 μM rpTx1, $n = 62$ for 5 μM rpTx1, and $n = 60$ for 5 μM rpTx-8K/A) (one-way ANOVA with post hoc analysis using the Dunnett's multiple comparisons test) (vehicle vs 1 μM rpTx1, $p = 0.0368$; vehicle vs 5 μM rpTx1, $p < 0.0001$; vehicle vs 5 μM rpTx1-8K/A, $p = 0.7532$; 1 μM rpTx1 vs 5 μM rpTx1, $p = 0.0191$; 5 μM rpTx1 vs 5 μM rpTx1-8K/A, $p < 0.0001$). (D) Representative current traces and scatter dot plot showing the effect of rpTx1-Ntermi on rNav1.8 upon intracellular (left) ($n = 5$, non-parametric Mann–Whitney two-tailed test) or extracellular application (right) ($n = 5$). (E) Representative current traces and scatter dot plot showing the effect of rpTx1-Ctermi on rNav1.8 upon intracellular (upper) ($n = 6$, non-parametric Mann–Whitney two-tailed test, $p = 0.0043$) or extracellular application (lower) ($n = 5$). (F) Inhibition of the fast inactivation of rNav1.8 by various concentrations of WT-rpTx1, M45A, E60A, and T66A mutants, respectively ($n = 3$–17 per concentration). (G) The locations of the three key residues M45, E60, and T66 are shown on the rpTx1 structure. Data are presented as mean ± S.E.M. **$p < 0.01$, ***$p < 0.001$, ****$p < 0.0001$. Source data are available online for this figure.

three mutants M45A, E60A and T66A were 0.35 ± 0.08 μM, 0.94 ± 0.24 μM and 0.44 ± 0.05 μM, respectively (Fig. 2F). Compared to wild type rpTx1 (WT-rpTx1), these mutants displayed approximately 6–22-fold lower potency, indicating that the three residues might contribute to the toxin activity. In addition, the CD spectra of these mutant toxins showed that their structural properties were similar to those of their WT counterpart, suggesting that the reduced activity is due to mutation that damaged the direct interaction between rpTx1 and rNav1.8 (Fig. EV7D). They are all distributed within the C-terminal sequence (Fig. 2G), confirming that rpTx1-Ctermi contributes to the binding of rpTx1 to rNav1.8. It is worth noting that E60 and T66 are spatially adjacent, located in a concave region. In addition, disulfide bonds are crucial for structure and function of these peptide toxins. Numerous evidence suggests that disrupting the disulfide bonds specifically in disulfide-containing toxins

completely abolishes their activity (Han et al, 2016; Ojeda et al, 2014; Wright et al, 2017). We mutated two cysteine residues (C61A and C62A) located in the C-terminal of rpTx1. Intracellular application of the mutant C61A&C62A revealed that its ability to inhibit the fast inactivation of rNav1.8 was completely lost (Fig. EV7D). Similarly, disrupting the disulfide bonds of the toxin in the fusion protein β4-rpTx1-C61A&C62A mutant also rendered it completely inactive (Fig. EV7E). Taken together, these data suggested that rpTx1 might contain two functional domains, rpTx1-Ntermi for cell penetration and rpTx1-Ctermi for the inhibition of the fast inactivation of Nav channels.

## RpTx1 directly targets the IFMT motif

The action of rpTx1 could be explained as follows: (1) indirect modulations by rpTx1 targeting other proteins that can regulate Nav

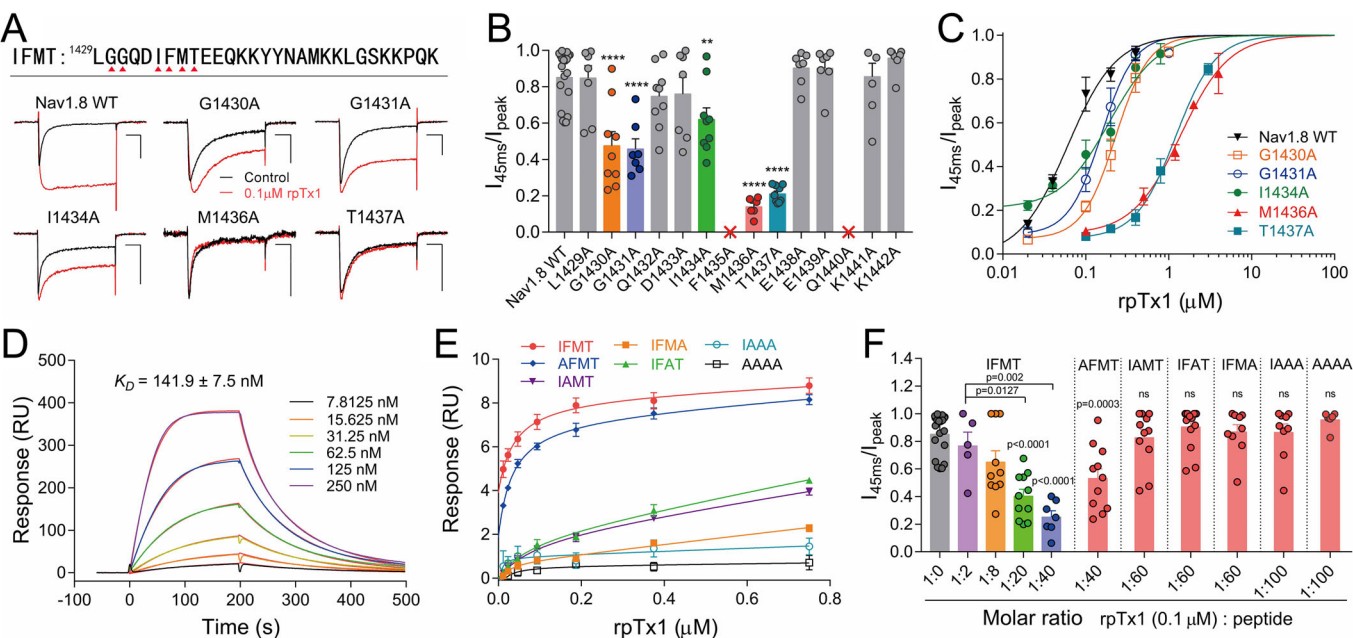

**Figure 3. RpTx1 interacts directly with the IFMT motif of Nav channels.**

(A) The sequence shows the IFMT motif of rNav1.8. Representative current traces from WT and mutant rNav1.8 channels in the absence or presence of 0.1 μM rpTx1 ($n = 6$–20). Scale bar: 1 nA, 10 ms. (B) Scatter dot plot shows the effect of 0.1 μM rpTx1 on the persistent currents ($I_{45\,ms}/I_{peak}$) of WT and mutant rNav1.8 channels (one-way ANOVA with post hoc analysis using the Dunnett's multiple comparisons test, G1430A, $p < 0.0001$; G1430A, $p < 0.0001$; I1434A, $p = 0.0066$; M1436A, $p < 0.0001$; T1437A, $p < 0.0001$; $n = 6$–20). The cross marks indicate the mutant rNav1.8 channels could not functionally express in ND7/23 cells. (C) Concentration-dependent curves of rpTx1 affecting WT and mutant rNav1.8 channels, respectively ($n = 4$–17 per concentration). (D) Representative SPR traces of the immobilized IFMT peptide in response to the series of concentrations of rpTx1 as measured with the kinetics assay mode ($n = 3$). (E) The IFMT and mutant peptides were injected on an SPR sensor immobilized with rpTx1, respectively. The fit curves were constructed by using the steady-state affinity model. SPR experiments were repeated three times. (F) Compare the effects of intracellularly applied pre-mixed rpTx1 and the IFMT peptide, or IFMT peptide mutants, on rNav1.8 currents at varying molar ratios, and assess whether IFMT or the mutant peptides can neutralize the influence of rpTx1 on Nav1.8 channel activity. The concentration of rpTx1 was fixed at 0.1 μM, and mixtures with varying molar ratios were prepared by adding different concentrations of IFMT or its mutant peptides (one-way ANOVA with post hoc analysis using the Dunnett's multiple comparisons test, $n = 5$–20). Data are presented as mean ± S.E.M. **$p < 0.01$, ***$p < 0.001$, ****$p < 0.0001$. Source data are available online for this figure.

channel activity through protein–protein interaction, posttranslational modification, or transcriptional regulation, and (2) direct interaction with Nav channels. The rpTx1-evoked response was quick (Fig. EV8), and by combining it with the fusion proteins (eGFP-rpTx1, β1/2/4-rpTx1) data, we speculated that a direct rpTx1-channel interaction was more likely. The IFMT motif, which is highly conserved, serves as the inactivation particle of Nav channels (Fig. EV9A). The four residues (I, F, M, and T) were found to interact with the receptor site formed by residues located in the S4–S5 linkers in DIII and DIV and the intracellular ends of the S5 and S6 segments of DIV. We asked whether the IFMT motif was a binding site for rpTx1. Alanine scanning was performed among the residues located in the intracellular linkers of rNav1.8, including the IFMT motif between DIII and DIV. Intriguingly, I1434, M1436 and T1437 in the core IFMT region and G1430 and G1431 in the N-terminal of the inactivation particle were important for rpTx1-induced activity (Fig. 3A–C). The EC$_{50}$ of rpTx1 was 228.3 ± 26.2 nM for G1430A, 154.0 ± 20.4 nM for G1431A, 193.5 ± 26.7 nM for I1434A, 1.40 ± 0.13 μM for M1436A and 1.24 ± 0.11 μM for T1437A (Fig. 3C). Furthermore, compared with the WT, the mutants I1434A, M1436A, and T1437A also eliminated or reduced the effect of rpTx1 on the steady-state inactivation curve of the channel (Figs. 1I and EV9C). Although these amino acid residues are coupled to the inactivation gate, mutating them to alanine did not significantly affect the fast inactivation of the channel, except for the I1434A mutation in rNav1.8, which significantly increased the persistent

currents (I1434A: $I_{45ms}/I_{peak} = 0.25 \pm 0.03$ for; WT: $I_{45ms}/I_{peak} = 0.1 \pm 0.01$, $p = 0.002$) (Figs. 3A and EV9B). Similar results were obtained with hNav1.5 and hNav1.7 channels, and two additional mutants (F1435A and Q1440A, which were not functionally expressed in the rNav1.8 channel) in the inactivation particle also significantly reduced toxin activity (Fig. EV9D,E). In contrast, mutations of the intracellular S4–S5 linker of each domain and the intercellular loop (including N-terminal, C-terminal, Loop1 and Loop2) of the channel did not significantly affect toxin sensitivity (Fig. EV9F).

To validate whether the IFMT motif region formed a direct binding site for the toxin, surface plasmon resonance (SPR) was used. A 29-residue peptide containing the inactivation particle sequence was chemically synthesized (Fig. EV9G). This IFMT peptide was immobilized by amine coupling onto a CM5 sensor chip, and increased concentrations of rpTx1 were injected into the microfluidic system. As shown in Fig. 3D, a robust binding response was observed between the IFMT peptide and rpTx1, and the binding was concentration-dependent with an equilibrium dissociation constant of 141.9 ± 7.5 nM. In turn, rpTx1 was immobilized, while increased concentrations of the IFMT peptide or mutant peptides (AFMT, IAMT, IFAT, IFMA, IAAA, or AAAA) were injected (Fig. EV9G). As we expected, the IFMT peptide exhibited specific binding to rpTx1. Once a certain amino acid residue of the four residues IFMT was mutated, the binding affinity

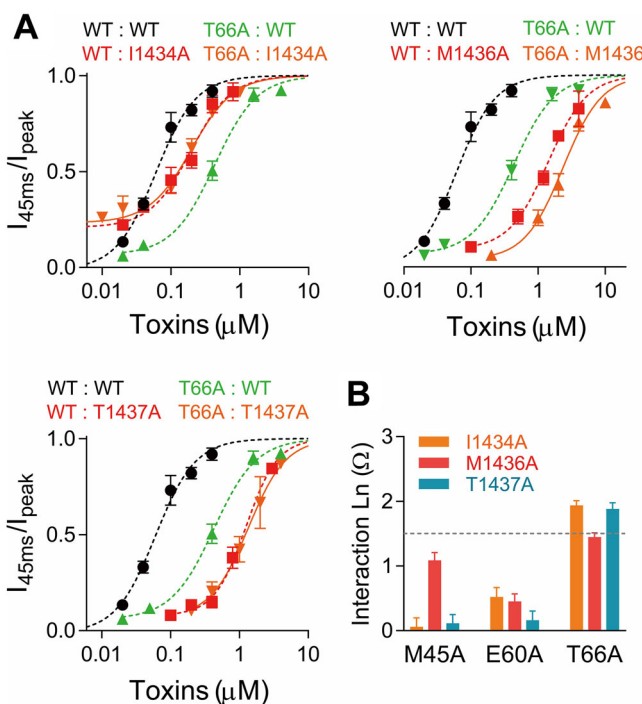

**Figure 4. The T66 of rpTx1 interacts specifically with I1434 and T1437 of rNav1.8.**

(A) Concentration–response curves for determining the interaction Ln(Ω) values between the rpTx1 T66 and rNav1.8 I1434 pair (n = 3–17 per concentration), the rpTx1 T66 and rNav1.8 M1436 pair (n = 4–17 per concentration), the rpTx1 T66 and rNav1.8 T1437 pair (n = 4–17 per concentration). (B) Summary of the interaction Ln(Ω) values. Data are presented as mean ± S.E.M. Source data are available online for this figure.

was significantly reduced (Fig. 3E). In particular, both the IAAA and AAAA mutants fully lost the ability to bind to rpTx1 (Fig. 3E). Furthermore, the IFMT peptide could counteract the rpTx1-evoked effect on ND7/23 cells expressing rNav1.8 when it was premixed with rpTx1, and then the mixture was applied intracellularly, but the mutant peptides had no such effect (Fig. 3F). The IFMT peptide itself did not affect Nav channel currents (Fig. EV9H), which was consistent with previous reports (Eaholtz et al, 1994). Taken together, these results demonstrated that rpTx1 should inhibit the fast inactivation of Nav channels by directly interacting with the IFMT motif.

## The molecular interaction between rpTx1 and Nav1.8

Double-mutant cycle analysis was used to measure the coupling of amino acid residue pairs between rpTx1 and rNav1.8. For this method, if a residue of rpTx1 specifically interacts with a residue of rNav1.8, the double mutations of these two residues should show nonadditive effects on binding affinity compared with a single mutation in rpTx1 or rNav1.8, and a specific interaction is assumed when the calculated interaction Ln(Ω) is larger than 1.5 kT (equivalent to 0.89 kcal/mol at 24 °C) (Schreiber and Fersht, 1995; Yang et al, 2015). Among the M45A, E60A, and T66A mutants of rpTx1 and the I1434, M1436, and T1437 mutants of rNav1.8, we performed nine pairs of mutation analyses, where only the binding

affinity for the two mutants E60A-I1434A, T66A-I1434A, or T66A-T1437A was close to that of the corresponding single mutations (Figs. 4A and EV10), among the Ln(Ω) value for the interaction between the mutants rpTx1-T66 and Nav1.8-I1434 or T1437 was larger than 1.5 kT (Fig. 4B). Therefore, T66 of rpTx1 and I1434 as well as T1437 of rNav1.8 might constitute a specific interaction pair between the two interacting molecules.

Metadynamics (MD) simulations were conducted to explore the detailed binding pattern between rpTx1 and the IFMT motif. Based on the resolved structure of the human Nav1.8 (hNav1.8) (Huang et al, 2022), MD simulations were employed to investigate the conformational dynamics of the IFMT motif. It was noted that there is a one-position discrepancy in the amino acid residue numbering of the IFMT motif between rNav1.8 (I1434/F1435/M1436/T1437) and hNav1.8 (I1433/F1434/M1435/T1436). Two collective variables were defined to describe the binding and unbinding processes of the IFMT motif to the receptor site: (1) the distance between the centroids of residues I1433, F1434, and M1435 of the IFMT motif and its receptor site, and (2) the number of atomic contacts between two residues (for detailed definitions, see Methods). The free energy landscape obtained from the MD simulations reveals that the motion of the IFMT motif involves three major states: the Bound State I (B1, Fig. 5A), the Bound State II (B2, Fig. 5B), and the Unbound State (U, Fig. 5C). A transition state (T, Fig. 5D) between B2 and U was also identified, highlighting potential conformational changes during the dissociation of the IFMT motif from the receptor site. The positions of these states in the defined conformational space, along with their corresponding free energy values, are shown in Fig. 5E. The characteristics and clustering of these states are presented in Table EV2. Conformational clustering in each minimum was conducted based on heavy atoms of specific pocket residues (V1269, V1270, A1273, P1605, A1606, N1609, I1610, N1715, I1718, A1719) along with I1433, F1434, and M1435, using clustering threshold of 3.0 Å. Furthermore, three salt bridge interactions (E1730-K1447, E1731-K1453, and E1734-K1453) were found to potentially regulate the motion of the IFMT motif, which is consistent with the finding reported by Clairfeuille et al (Clairfeuille et al, 2019). Particularly, the frequency of the E1734-K1453 salt bridge increased as the IFMT motif dissociated (26.04% in B1, 32.00% in B2, 45.62% in T, and 58.32% in U), suggesting its involvement in regulating the unbinding of the IFMT motif.

Considering the potential interactions between hNav1.8-I1433/T1436 and rpTx1-T66 (Fig. 4), we conducted a protein-protein docking by restricting the distance between rpTx1-T66 and hNav1.8-I1433, as well as rpTx1-T66 and hNav1.8-T1436, to within 5 Å. Based on the conclusions drawn from the above analysis, we hypothesized that rpTx1 may bind to the IFMT motif in the U state. Our clustering results revealed that three major representative conformations (U1-3) in the U state (>20%, Table EV2). As shown in Fig. 5F, the predicted rpTx1 structure can be docked to representative conformations of hNav1.8 when the IFMT motif is in the U state, and the most representative conformation is the rpTx1-hNav1.8-U-3 complex. Unbiased MD simulations were then conducted to evaluate the stability of the rpTx1-hNav1.8-U-3 complexes. As anticipated, during the 200 ns MD simulation, rpTx1 consistently remained stably bound to the IFMT motif, keeping it in the U state and preventing transitioning to any bound state, as shown by the 200 ns snapshot in Fig. 5F. The distance-time curve

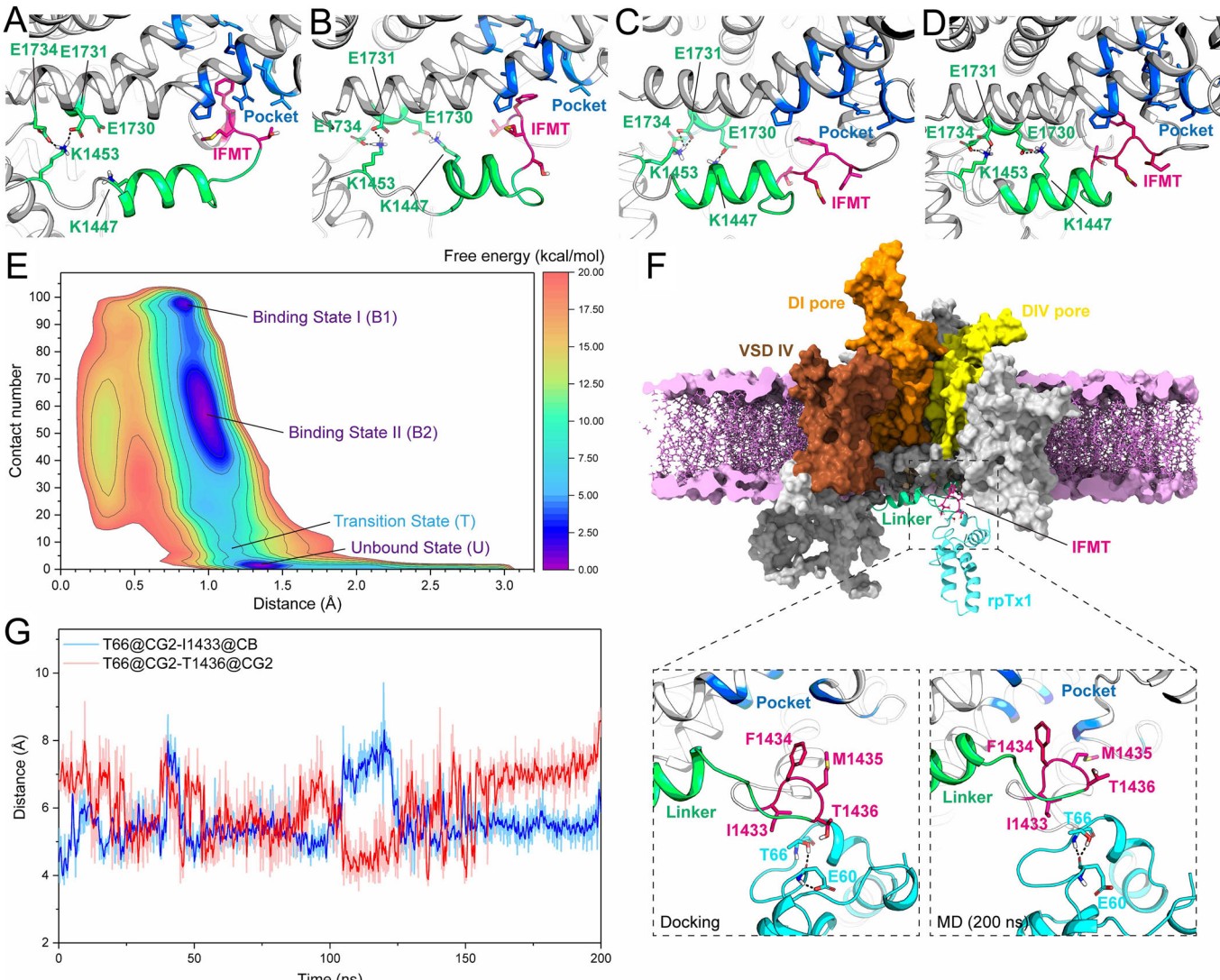

**Figure 5. The docking-based prediction of rpTx1-hNav1.8 complex.**

(A–D) As revealed by metadynamics (MD) simulation, there are four different states of the IFMT motif based on its binding and unbinding to the receptor site (pocket), including the Bound State I (B1), the Bound State II (B2), the Transitional state (T), and the Unbound state (U). (E) The free energy landscape given by MD simulation illustrates different states of the IFMT motif. (F) The global view of the docking complex of rpTx1-Nav1.8-U-3 (upper panel), and the local view of the binding between rpTx1 and the IFMT motif by the end of 200 ns MD simulation (lower panel). (G) The distance changes of two atom-pairs during the 200 ns MD simulation reveals continuous contacts between rpTx1-T66 and hNav1.8-I1433/T1436. In local view of (A–D), four residues I-F-M-T of the IFMT motif are colored in pink, while the linker region and the IFMT receptor site are colored in green and blue, respectively. In global view of (F), the rpTx1 molecule is represented by cyan cartoon, and hNav1.8 is drawn as surface with different region in varied colors, i.e., DI pore in dark orange, VSD IV in sienna, DIV pore in yellow, the IFMT motif in pink, and the linker region in green, respectively. As for membrane components, PA and OL molecules are given as violet stick models, while PC molecules are given as light violet surfaces. Source data are available online for this figure.

for the CG2 atom of rpTx1-T66 and the CB atom of hNav1.8-I1433, as well as the CG2 atom of hNav1.8-T1436, is presented in Fig. 5G. These distances remained within the range of 4–8 Å, indicating consistent contacts between rpTx1-T66 and hNav1.8-I1433/T1436. It is noted that considering the alkyl side chain of hNav1.8-I1433, this residue is more likely to play a dominant role in hydrophobic interactions at the interface. The distance between the CG2 atom of rpTx1-T66 and the CB atom of hNav1.8-I1433 remained within 6 Å for the majority of the MD simulation, suggesting a sustained hydrophobic interaction between these two

residues. Additionally, we observed that the side-chain hydroxyl group of rpTx1-T66 folded inward, forming a stable hydrogen bond with the backbone atoms of rpTx1-E60 (Fig. 5F). Simultaneously, the hydrophobic portion of rpTx1-T66's side chain was exposed at the protein-protein interface, potentially facilitating hydrophobic interactions between rpTx1-T66 and hNav1.8-I1433/T1436. These results support the experimental data, and suggested that rpTx1-T66 may interact directly with hNav1.8-I1433/T1436. Collectively, these data imply that rpTx1 preferentially targets the dissociated IFMT motif of Nav channels.

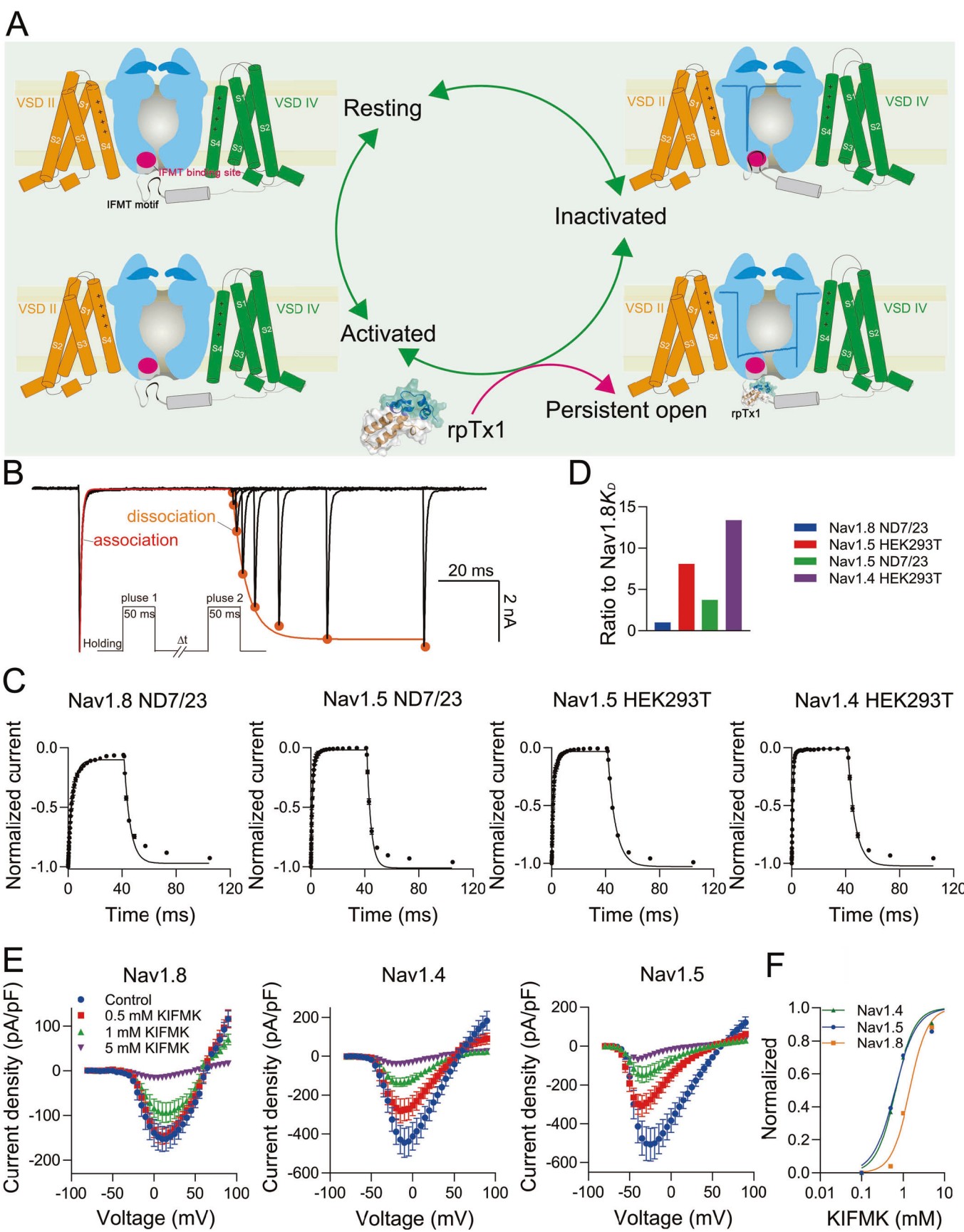

**Figure 6.  The action model underlying rpTx1 inhibiting the fast inactivation of Nav channels by directly targeting the IFMT motif.**

(A) The diagram shows that rpTx1 is in competition with the IFMT motif receptor site to bind to this motif, preventing this motif association with the receptor site and removing the fast inactivation of Nav channels. However, in the absence of rpTx1, when a Nav channel is activated, the IFMT motif rapidly binds to the receptor site, leading to the fast inactivation. (B) Representative current traces for the fast inactivation and recovery from the fast inactivation of Nav currents, which can be used to simulate the process of the IFMT motif binding to the receptor site (association, red) and its subsequent release (dissociation, orange) during the deactivation of Nav channels. The inset shows the current elicited protocol. To determine the recovery from fast inactivation, a 50-ms prepulse (pulse 1) was used: at −20 mV for hNav1.5, at 0 mV for rNav1.4, or at 20 mV for rNav1.8, to move channels into the fast inactivated state. This was followed by a pulse at −100 mV, of increased duration, to allow channels to recover from fast inactivation. Subsequently, a 50-ms test pulse (pulse 2) was applied to measure the available current. (C) The association and dissociation kinetics of the IFMT motif with the receptor sites of rNav1.8 ($n = 7$), rNav1.4 ($n = 7$), and hNav1.5 ($n = 9$ for HEK293T, $n = 7$ for ND7/23) which are expressed in HEK293T or ND7/23 of cells. (D) Bars show the ratio of the $K_D$ of the IFMT motif binding to the receptor site of rNav1.4 or hNav1.5, relative to that of rNav1.8. (E) The effect of the intracellular application of the peptide KIFMK on the current density of rNav1.8, rNav1.4, or hNav1.5 ($n = 8$–10 per concentration). (F) The concentration-dependent inhibition of rNav1.8, hNav1.4, and rNav1.5 currents by the intracellular application of KIFMK based on the data from (E). Data are presented as mean ± S.E.M. Source data are available online for this figure.

## Underlying mechanisms of rpTx1's high activity on rNav1.8

Based on the above results, a mechanism model was proposed to explain how rpTx1 inhibits Nav channel inactivation intracellularly (Fig. 6A). Generally speaking, Nav channels have three basic states: resting, activated, and inactivated. Upon cell membrane depolarization, Nav channels are activated, opening the pore to allow $Na^+$ to flow inward. Subsequently, the receptor site for the IFMT motif is quickly exposed and readily binds to this motif, leading to the inactivation of Nav channels and the termination of $Na^+$ flow. However, in the presence of rpTx1 inside cells, the receptor site is in competition with rpTx1 for binding to the IFMT motif during the activation/inactivation process, resulting in two channel conformations: one is the persistently open state where the IFMT motif is captured by rpTx1, and the other is the inactivated state where the IFMT motif binds to its receptor site, that is, the inactivation of Nav channels can be partially or completely inhibited by the intracellular application of rpTx1, depending on the concentration of rpTx1 and Nav channel subtypes.

Our data indicate that rpTx1 is able to completely remove the fast inactivation of rNav1.8 but not of other subtypes at low concentrations. However, the IFMT motifs are identical and should have similar binding affinity for rpTx1 in all Nav subtypes, and cell types could not influence rpTx1's effect (Fig. EV11), which cannot account for rpTx1's higher activity on rNav1.8. According to the action model as shown in Fig. 6A, we hypothesized that the binding affinity between the IFMT motif and the receptor site is important for rpTx1's potency on different Nav subtypes. Since the time course of the fast inactivation of Nav macroscopic currents after channel opening represents the process of the IFMT motif's association to the receptor site, and the recovery from the fast inactivation reflects the process of returning to the resting state from the fast inactivated state, and the dissociation of the IFMT motif from the receptor site (Fig. 6B), the binding affinity ($K_D$) can be calculated by the association-dissociation equation (see method) (Fig. 6C). Compared with hNav1.5 expressed in ND7/23 or HEK293T cells and rNav1.4 expressed in HEK293T cells, rNav1.8 expressed in ND7/23 cells displayed a relatively slow association rate ($K_{on}$), while the dissociation rate ($K_{off}$) was comparable to those of other Nav channel subtypes. Consequently, the equilibrium dissociation constant ($K_D$, calculated as $K_{off}/K_{on}$) of rNav1.8 was significantly lower than that of rNav1.4 or hNav1.5 (Fig. 6C,D). In addition, we synthesized the KIFMK peptide (a free peptide that can mimic the binding of the IFMT motif to the receptor site) and determined it blockage on Nav

currents (Eaholtz et al, 1994; Wang and Wang, 2005). As shown in Fig. 6E, the intracellular application of 0.5 mM KIFMK decreased the current densities of Nav1.4 and Nav1.5 by approximately 39.3% and 36.9%, respectively, but it hardly affected the current density of rNav1.8. The $IC_{50}$ values of KIFMK for rNav1.4 and hNav1.5 were similar, but were two time smaller than that for rNav1.8 (Fig. 6F), indicating lower binding affinity between KIFMK and the receptor site of rNav1.8. This might be derived from minor differences in the receptor sites of these Nav channels. Taken together, these data suggest that the high activity of rpTx1 should be resulted from the relatively low binding affinity between the IFMT motif and the receptor site of rNav1.8, as compared to other Nav subtypes.

## Discussion

In summary, in the present study, we identified that the centipede toxin modifies Nav channel activity via binding to the inactivation particle. This 83-residue toxin can penetrate the cell membrane and directly interact with the fast inactivation particle (the IFMT motif) of Nav channels, which consequently impedes the inactivation particle from binding to its receptor site and therefore delays the fast inactivation of Nav channels. This unique action mechanism might be derived from its distinctive structure containing two domains. The N-terminal domain forms an α-helical hairpin and mainly functions in cell penetration, while the C-terminal one consists of a disulfide bond-stabilized compact structure and mainly contributes to Nav channel modulation.

To our knowledge, no small molecules or peptide toxins have been reported to regulate Nav channel activity by directly targeting the IFMT motif region; therefore, our study defined the inactivation particle as a new binding site for neurotoxins (Fig. EV1), and rpTx1 is the first peptide toxin known to act on this site. Our multiple lines of evidence indicate that rpTx1 binds to the IFMT motif with high affinity, and four residues (I1434, F1435, M1436 and T1437) of Nav channels and three residues (M45, E60 and T66) of rpTx1 play key roles in this binding, especially I1434/T1437 and T66, which form a specific interaction (Figs. 2F, 3A–C, 4A,B, and EV9B,C). Different binding sites of Nav channels are distinguished not only by the localization of the toxin binding regions but also by the effects of the toxin's action. Site 3 peptide toxins, common to spider and scorpion venoms, are found to bind to the extracellular loop of VSD IV to slow or even completely abolish the fast inactivation, which is macroscopically similar to the

effect of rpTx1 on Nav channels (Catterall et al, 2007). However, their action mechanisms are distinct. Site 3 peptide toxins hinder the outward movement of gating charges of S4 to stabilize the voltage sensor of VSD IV in the deactivated state, thereby slowing the receptor site exposure of the IFMT motif in fast inactivation or destabilizing the fast-inactivated state (Clairfeuille et al, 2019; Jiang et al, 2021), whereas rpTx1 directly binds to the IFMT motif to prevent its interaction with the receptor site.

It is generally believed that the IFMT motif blocks the pore of Nav channels upon activation (Catterall, 2012). However, recent cryo-EM studies show that the IFMT motif targets the receptor site formed by residues located in the S4–S5 linkers in DIII and DIV and the intracellular ends of the S5 and S6 segments of DIV, causing the intracellular gate closure through allosteric effects rather than direct blocking (Jiang et al, 2020; Pan et al, 2018; Shen et al, 2019; Yan et al, 2017). Given the dynamic nature of Nav channels, resolving inactivation mechanisms remains challenging. However, recent research using genetically encoded photochemistry has revealed dynamic changes of the conformations of the IFMT motif during the fast inactivation (Goodchild and Ahern, 2024). Our findings also reveal that the IFMT motif exhibits distinct states (conformations) depending on its association and dissociation with the receptor site (Fig. 5A-D). Specifically, rpTx1 stabilizes the conformation of the IFMT motif which stays away from the receptor site, possibly representing an early state of the fast inactivation. Therefore, rpTx1 has the potential to capture the resting or intermediate conformations of the IFMT motif, aiding in the investigation of the fast inactivation mechanisms of Nav channels.

The IFMT motifs of different Nav subtypes are almost identical, but Nav1.8 as well as Nav1.9 exhibit slower fast inactivation rates, and the specific mechanism is still unclear. There may be several reasons: (1) different sensitivity to voltage, as the fast inactivation of Nav channels is voltage-dependent; (2) the existence of distinct interaction proteins that affect the fast inactivation (Gade et al, 2020; Goldfarb, 2012; Sarhan et al, 2012); (3) the influence of other regions of Nav channels, such as the C-terminal (Deschênes et al, 2001; Gade et al, 2020; Motoike et al, 2004; Shen et al, 2017); (4) different affinities of the IFMT motif binding to the receptor sites. Our results show that compared with other Nav subtypes, Nav1.8 demonstrates a low affinity between the IFMT motif and its receptor site, which is also more conducive to the competitive binding of rpTx1. Thereby, it provides a possible explanation for the stronger activity of rpTx1 on Nav1.8. Considering Nav1.9 has slower fast inactivation than Nav1.8, rpTx1 should be more potent on Nav1.9 than the latter, but our data are the opposite. This suggests that the explanation mentioned above may not be applicable to Nav1.9, and other factors may play a more significant role. Additionally, our MD simulations have revealed that, apart from the IFMT motif, more amino acid residues near this motif also participate in binding with rpTx1. This may also influence the varying activity of rpTx1 on different Nav channel subtypes. Therefore, we must acknowledge that even for Nav1.8, the mechanism we proposed is a simplified model. Actually, the fast inactivation mechanism of Nav channels is likely more complex than currently understood, and the underlying mechanisms responsible for rpTx1's selectivity on Nav channel subtypes remain unclear, necessitating further studies to elucidate the precise mechanism of action of the inactivation gate.

Animal venoms contain diverse peptide toxins with various activities, such as ion channel modulation, cell penetration, and cytotoxicity (Undheim et al, 2015). Neurotoxins with ion channel modulation activity commonly adopt disulfide bond-constrained compact structures (Klint et al, 2012), while cell-penetrating peptides are usually rich in

positively charged amino acid residues and form uncomplicated α-helical or β-sheet structures (Guidotti et al, 2017; Radis-Baptista, 2021). These toxins are all important to venomous animals for capturing prey and defending themselves. However, few toxins have been reported to combine these two functions. Unusually, King et al discovered a scorpion peptide toxin that can enter cells and activate TRPA1 by binding to the intracellular site (Lin King et al, 2019), and several scorpion peptide toxins were also found to cross the plasma membrane to activate the intracellular calcium channel type 1 ryanodine receptor (RyR1) (Estève et al, 2005; Gurrola et al, 2010; Schwartz et al, 2009). By incorporating these two kinds of toxin domains to form a single molecule, rpTx1 also achieves two functions, cell penetration and the intracellular modulation of Nav channels. Furthermore, the combination of two domains is not a simple connection but creates a synergistic effect. Our data demonstrated that rpTx1-Ntermi (the cell penetration domain) enhances the potency of rpTx1-Ctermi (the Nav channel modulation domain), and the reverse is also true. Our observations show that only a small subset of cells exhibit sufficient entry of rpTx1 and a noticeable inhibition of fast inactivation, which indicates the cell penetrating efficiency of the peptide is not high. Crossing the cell membrane is inherently challenging for macromolecules. Numerous studies have demonstrated that the cell membrane crossing efficiency of cell-penetrating peptides (CPPs) is generally low (Guidotti et al, 2017; Radis-Baptista, 2021); even the most effective CPPs identified to date do not achieve 100% efficiency. The N-terminal sequence of rpTx1 functions as a CPP, carrying the C-terminal sequence as a cargo into the cell. Compared to most CPPs, its molecular weight is significantly large, which understandably results in low cell penetrating efficiency. The cell penetrating efficiency of rpTx1 is influenced by various factors, and whether it shares the same mechanism as CPPs or operates through a different one is crucial for explaining the effect of rpTx1. This requires us to conduct further research. BLAST analysis reveals that peptides resembling rpTx1 are widely distributed in centipede venoms (Fig. S2D), implying shared functional mechanisms among them. Such toxins with comparable traits could also be present in other venomous animals.

# Methods

**Reagents and tools table**

| Reagent/Resource | Reference or Source | Identifier or Catalog Number |
|---|---|---|
| **Experimental models** | | |
| ND7/23 | Cell Bank of the Chinese Academy of Sciences (Zhang et al, 2008) | CSTR:19375.09.3101ETCSCSP5026 |
| HEK 293T | Cell Bank of the Chinese Academy of Sciences | CSTR:19375.09.3101HUMGNHu44 |
| **Recombinant DNA** | | |
| rNav1.2, rNav1.3, rNav1.4, hNav1.5, mNav1.6, hNav1.7, rNav1.8, beta1 and beta 2 | Plasmid backbone from Prof. Theodore R.Cummins (Indiana University Indianapolis) | N/A |

| Reagent/Resource | Reference or Source | Identifier or Catalog Number |
|---|---|---|
| Bacterial Nav channel NaChBac | Gift from Prof. David E. Clapham lab (Howard Hughes Medical Institute, Department of Cardiology, Children's Hospital Boston, Boston, USA) | N/A |
| hNav1.9-GFP | (Zhou et al, 2017) | N/A |
| Beta 1-rpTx1, beta 2-rpTx1, beta 4-rpTx1, and eGFP-rpTx1 | This study | Sequence available upon request |
| Beta 4 | This study | Sequence available upon request |
| The chimera channels (1815 NT, 1815 Loop1, 1815 Loop2, and 1815 CT) | (Zhou et al, 2018) | N/A |
| mCav1.2 | Addgene | #26572 |
| rCav1.3 | Addgene | #49333 |
| rCav3.1 | Addgene | #45812 |
| pET32a-rpTx1 and its mutants | This study | N/A |
| rNav1.8 mutants | This study | N/A |
| rNav1.4 mutants | This study | N/A |
| hNav1.5 mutants | This study | N/A |
| T-Vector pMD™20 | Takara | #3270 |
| **Antibodies** | | |
| Rabbit anti-GFP | Proteintech | #50430-2-AP |
| HRP-Goat Anti-Rabbit Recombinant Secondary Antibody | Proteintech | #RGAR011 |
| **Peptides, and recombinant peptides** | | |
| IFMT peptide and its mutants | This study | N/A |
| rpTx1-Ntermi | This study | N/A |
| rpTx1-Ntermi-FITC | This study | N/A |
| rpTx1 and its mutants | This study | N/A |
| KIFMK peptide | This study | N/A |
| **Chemicals, enzymes, and other reagents** | | |
| Lipofectamine 2000 | Invitrogen | #11668019 |
| X-tremeGENE HP DNA Transfection Reagent | Roche | #06366236001 |
| FluoroTag™ FITC Conjugation Kit | Sigma-Aldrich | #FITC1-1KT |
| RIPA lysis buffer | Sangon | #PL001 |

| Reagent/Resource | Reference or Source | Identifier or Catalog Number |
|---|---|---|
| DAPI | Beyotime | #C1005 |
| DiD | Beyotime | #C1995S |
| BCA Protein Assay Kit | Sangon | #SK3021 |
| Tetrodotoxin | Chengdu must bio-teachnology | #A0224 |
| TRIzol | Invitrogen | #15596026CN |
| KOD-plus-neo | TOYOBO | #KOD-401 |
| SMART-RACE Kit | TaKaRa | #6106 |
| Tks Gflex DNA Polymerase | TaKaRa | #R060A |
| Tris (2-carboxyethyl) phosphine (TCEP) | Invitrogen | #T2556 |
| Fetal bovine serum | Biological Industries | #04-001-1ACS |
| DMEM basic | Gibco | #C11995500BT |
| Poly-L-Lysine solution | Sigma-Aldrich | #P4707 |
| 0.25%Trypsin-EDTA | Gibco | #C25200-072 |
| Penicillin-streptomycin | Gibco | #15070063 |
| TEV Protease | Beyotime | #P2310M |
| Ni-NTA Sepharose 6FF (His-Tag) | Sangon | #C600033 |
| *S. subspinipes mutalans* Venom | This study | N/A |
| **Software** | | |
| PatchMaster | HEKA Elektronik | v2x73 |
| IgorPro6 | WaveMetrics | Version 6.1.0.9 |
| GraphPad Prism | GraphPad | Version 7.00 |
| Leica Application Suite X | Leica | Version 3.4.2.18368 |
| PPSQ Analysis | Shimadzu | Version 2.21 |
| Biacore Evaluation Software | GE Healthcare | Version 3.0 |
| AMBER20 | (Case et al, 2005) | N/A |
| Schrödinger 2020 | Schrödinger, Inc | https://www.schrodinger.com/ |
| Pymol | Schrödinger, Inc | https://pymol.org/ |
| GROMACS 2018 | (Abraham et al, 2015) | N/A |
| ChimeraX | Resource for Biocomputing, Visualization, and Informatics, University of California San Francisco, 1001 Potrero Ave, San Francisco, CA 94110 | N/A |
| Office Excel 2010 | Microsoft | Version 14.0.4760.1000 |

## Venom collection and toxin purification

The venom was obtained by electrical stimulation of the Centipedes of *S. subspinipes mutilans* (both sexes), and the freeze-dried crude venom was stored at −20 °C prior to analysis. Lyophilized venom was dissolved in double-distilled water. The venom was purified by semi-preparative reversed-phase high-performance liquid chromatography (RP-HPLC) on an Ultimate® XB-C18 column (300 Å, 10 mm × 250 mm, Welch Materials Inc., Shanghai, China) on the Hanbon HPLC system (Hanbon Sci&Tech., Jiangsu, China). A linear gradient of solvent A (0.1% formic acid in acetonitrile) in solvent B (0.1% formic acid in water) was used at a flow rate of 3 mL min$^{-1}$: 5% A for 5 min, then 5–60% A over 55 min. The absorbance was measured at 215 nm, and the eluted fractions were collected and lyophilized and then fractionated by using ion-exchange chromatography with an XB-SCX column (4.6 mm × 250 mm, 5 μm; Welch, China) on an analytical Waters 2795 HPLC system. The 1 M NaCl gradient increased at a rate of 1% per minute from 0–81% at a flow rate of 1 mL min$^{-1}$. RP-HPLC desalination was performed on fractions with rpTx1, using a C18 column (4.6 mm × 250 mm, 5 μm; Welch, China) with a slower increasing acetonitrile gradient (acetonitrile at an increasing rate of 0.5% per minute, and a flow rate of 1 mL min$^{-1}$) on an analytical Waters 2795 HPLC system to yield purified rpTx1. The molecular weights of the components separated by RP-HPLC were detected by matrix assisted laser desorption/ionization-time of-flight mass spectrometry (MALDI–TOF MS, AB SCIEX TOF/TOFTM 5800 system, Applied Biosystems, USA). The fractions containing rpTx1 were lyophilized and stored at −20 °C until further use.

## Mass spectrometry analysis and amino acid sequencing

The purity and molecular weights of rpTx1 was determined by electrospray ionization mass spectrometry (ESI-MS) analysis. The N-terminal amino acid sequence of rpTx1 was obtained by automated Edman degradation in a PPSQ-51A protein sequencer (Shimadzu Corporation, Kyoto, Japan).

## Assignment of the disulfide bonds of rpTx1

To determine the disulfide connections of rpTx1, a truncated rpTx1 with short N-terminal (NH$_2$-GIFPENRKSCMTNCKHVEGCFLL-SPECCPKMTSTCLELDIVKEHMKKTK-COOH) was obtained via expression in the *Escherichia coli* (*E.coli*) BL21 (DE3). The peptide (100 μg) was dissolved in 20 μL of 0.1 M citrate buffer (pH3.0) containing 5 M guanidine·HCl for 30 min. Then, added to the mixed solution was 20 μL of 0.1 M Tris (2-carboxyethyl) phosphine (TCEP), and placed at 40 °C for 8 min. The partially reduced sample was fractionated by RH-HPLC and identified by matrix assisted laser desorption/ionization-time of-flight mass spectrometry (MALDI–TOF MS, AB SCIEX TOF/TOF™ 5800 system, Applied Biosystems, USA). The partially reduced peptide fractions with free thiols were lyophilized and alkylated with 0.5 M iodoacetamide (pH 8.3). The alkylated peptides were desalted by RP-HPLC and then injected to automated Edman degradation in a PPSQ-51A protein sequencer (Shimadzu Corporation, Kyoto, Japan).

## RpTx1 cloning

Total RNA was extracted from the venom gland of centipedes *S. subspinipes mutilans* using TRIzol (Invitrogen). A degenerate primer (5'-GTNGARGAYYTNGARCCNCCNAC-3', where N = A, C, G or T; Y = C or T; R = A or G) derived from rpTx1 N-terminal peptide sequence (VEDLEPPT) was used to perform 3'RACE (SMART-RACE kit, Clontech). PCR was performed using Tks Gflex DNA Polymerase (TaKaRa), and the PCR products were cloned into T-Vector pMD™20, and DNA sequencing obtained the gene sequence of mature rpTx1.

## Production of recombinant rpTx1 and its mutants

RpTx1 cDNA was optimized for expression in BL21 (DE3), synthesized by Genscript Biotech Corporation (Nanjing, China), and cloned in the pET32a (+) plasmid (pET32a-rpTx1). RpTx1 was produced as fusion proteins containing the 6×His tag for affinity purification protein, a TrxA protein to aid solubility, and a tobacco etch virus (TEV) protease recognition site at the N-terminal (Fig. EV4A). A TEV protease was used to release rpTx1. For confirming the expression of rpTx1 in the soluble cell fraction, sodium dodecyl sulfate-polyacrylamide gel electrophoresis (SDS-PAGE) was used. Next, RP-HPLC on a C18 column (4.6 × 250 mm, 5 μm, Welch Materials Inc., Shanghai, China) was used to isolate and purify the peptide on an analytical Shimadzu LC20AT system using a 30 min linear acetonitrile gradient from 25% to 40% at a flow rate of 1 mL min$^{-1}$. Finally, the fraction containing rpTx1 was collected, freeze-dried, and stored at −20 °C until analysis. For C18 RP-HPLC, the solvent A was 0.1% TFA in water, the solvent B was 0.1% TFA in acetonitrile. The molecular weight was confirmed by ESI-MS. The theoretical mass of recombinant peptides include the non-native N-terminal Gly residue that is a vestige of the TEV protease recognition site. The protocol used for expression and purification of the mutant toxins was the same as that used for WT rpTx1.

## Peptide synthesis and purification

The IFMT peptide, its mutant peptides, rpTx1-Ntermi, and the KIFMK peptide were synthesized manually by using a Fmoc (N-(9-fluorenyl) methoxycarbonyl)/tert-butyl strategy and 1-hydroxyben-zotriazole/2-(1H-benzotriazole-1-yl)-1,1,3,3-tetramethyluronium tetrafluoroborate/N-methylmorpholine coupling method, as described in our previous study. Briefly, peptides synthesis was accomplished on a 0.1-mmol scale. The terminal Fmoc group was removed by treatment with 1:4 piperidine/N,N-dimethylformamide (v/v). Using a cocktail (82.5% trifluoroacetic acid, 5% double-distilled H$_2$O, 5% phenol, 5% thioanisole, and 2.5% ethanedithiol) for 2 h at room temperature allowed cleavage from the resin as well as removal of side chain protective groups. After filtering the resin, the free peptide was precipitated in cold ether at 4 °C. Centrifuged and washed with cold ether, the crude product was dissolved in ddH$_2$O and lyophilized. Then purified by a C18 column (4.6 × 250 mm, 5 μm, Welch Materials Inc., Shanghai, China) with a 20-min linear acetonitrile gradient ranging from 20% to 40% at a 1 mL min$^{-1}$ flow rate (Shimadzu LC20AT system). The molecular weight was determined by ESI-MS.

## FITC labeling of peptides

Purified rpTx1, rpTx1-8K/A, or rpTx1-Ntermi were conjugated by FluoroTag FITC Conjugation Kit (Sigma) according to the manufacturer's recommendation. Briefly, A peptide was incubated with FITC for 2 h. Then, apply the reaction mixture to top of a Sephadex G-25M column and collect the labeled peptide. The

conjugated peptides were further purified by RP-HPLC. FITC conjugation usually occurs through free amino groups of a peptide, forming a stable thiourea bond. Therefore, in this study, FITC may be labeled on the N-terminal of a peptide and/or the side chain of lysine (Lys, K) residues. The number of conjugated FITC to a peptide could be determined by MS analysis.

## Plasmid constructs and mutagenesis

Rat Nav1.2 (rNav1.2), rNav1.3, rNav1.4, human Nav1.5 (hNav1.5), mouse Nav1.6 (mNav1.6), hNav1.7, rNav1.8 clones and beta subunit (β1 and β2) clones were kindly gift from professor Theodore R.Cummins (Department of Biology, School of Science, Indiana University-Purdue University Indianapolis, Indianapolis, USA), and bacterial Nav channel NaChBac clones were kindly gift from professor David E. Clapham lab (Howard Hughes Medical Institute, Department of Cardiology, Children's Hospital Boston, Boston, USA), and were cloned in a pTracer-CMV2 vector. The cDNA clones of β1-rpTx1, β2-rpTx1, β4-rpTx1, and eGFP-rpTx1 were cloned in the pEGFP-C1. Both eGFP-rpTx1 and beta1/2/4-rpTx1 were constructed using an identical method, which involves fusion of rpTx1 to the C-terminal of either eGFP or β1/2/4 via a linker (RILENLYFQG) hNav1.9 clone was cloned in the pEGFP-N1. The C-terminal of hNav1.9 was linked a GFP to construct a fusion protein channel via a linker (ARDPPAA) (hNav1.9-GFP) (Zhou et al, 2017). All site mutations of hNav1.5, rNav1.7, and rNav1.8 were constructed by using the KOD-Plus-Neo (TOYOBO) according to the manufacture's instruction. The chimera channels (1815 NT, 1815 Loop1, 1815 Loop2, and 1815 CT) were from our previous study (Zhou et al, 2018). All mutations were verified by DNA sequencing.

## Cell culture and transfection

ND7/23 and HEK293T cells were maintained in Dulbecco's modified Eagle's medium (DMEM) supplemented with 10% fetal bovine serum, 2 mM L-glutamine, 100 U $L^{-1}$ penicillin and 100 μg $mL^{-1}$ streptomycin in a 5% $CO_2$ incubator at 37 °C. Cells were trypsinized, diluted with culture medium, and grown in 35 mm dishes. When grown to 90% confluence, ND7/23 cells were transfected with rNav1.8 and rNav1.8 mutants using Lipofectamine 2000 (Thermo Fisher Scientific) according to the manufacturer's instructions. The transfection of hNav1.7 and hNav1.7 mutants together with β1 and β2-eGFP other ion channels (hNav1.5, hNav1.5 mutants, rNav1.2-1.4 and mNav1.6, mKv1.3, rKv2.1, rKv4.2, hKCNQ2, mCav1.2, rCav1.3 and rCav3.1 with eGFP) into HEK293T cells using Lipofectamine 2000 and the transfections of hNav1.9-GFP were performed by using the transfection kit X-tremeGENE HP DNA Transfection Reagent (Roche, Basel, Switzerland) according to the manufacturer's instructions. Six hours after transfection, the cells were seeded onto poly-D-lysine-coated coverslips (Thermo Fisher Scientific) and maintained at 37 °C in 95% $CO_2$ for 24 h before whole-cell patch-clamp recording. For the visual identification of individual transfected cells, green fluorescence was used.

## Electrophysiology

Whole-cell patch-clamp recordings were performed at room temperature (25 ± 2 °C) using an EPC-10 USB patch-clamp amplifier operated by PatchMaster software (HEKA Elektronik, Lambrecht, Germany) or Axopatch 700B amplifier (Molecular Devices). Fire-polished electrodes (1.8–2.0 MΩ) were fabricated from 1.5 mm capillary glass using a PC-10 puller (NARISHIGE, Tokyo, Japan). The voltage errors were minimized with 80% series resistance compensation. The liquid junction potential was compensated using EPC-10 USB amplifier, and no salt bridge was used. Voltage-dependent currents were acquired with Patchmaster 4 min after establishing a whole-cell configuration, which was sampled at 30 kHz, and filtered at 2.9 kHz.

For recording the Nav channel currents, the extracellular solution contained (in mM) 150 NaCl, 2 KCl, 1.5 $CaCl_2$, 1 $MgCl_2$, 10 HEPES (pH 7.4 with NaOH) and was supplemented with 1 μM tetrodotoxin (TTX) to block endogenous $Na^+$ currents in ND7/23 cells. The pipette solution contained (in mM) 10 NaCl, 135 CsF, 10 EGTA, 10 HEPES (pH 7.3 with CsOH). For recording the Kv channel currents, the bath solution contained (in mM) 140 NaCl, 2 KCl, 1.5 $CaCl_2$, 1 $MgCl_2$, 10 HEPES (pH 7.4 with NaOH), and the pipette solution contained (in mM) 140 KCl, 2.5 $MgCl_2$, 10 HEPES, and 10 EGTA (pH 7.4 with NaOH). For recording voltage-gated Cav channels, the bath solution contains (in mM): 100 NaCl, 20 TEA-Cl, 5 $Ba_2Cl$, 1 $MgCl_2$, 10 HEPES, and 10 Glucose (pH 7.4 with NaOH). The corresponding pipette solution contains (in mM): 120 $CsMeSO_4$, 10 HEPES, 5 Mg-ATP, and 11 EGTA (pH 7.2 with CsOH). All reagents were purchased from Sigma.

To generate activation curves, cells were held at −90 mV and stepped to potentials of −80 to +60 mV in 5-mV increments for 50-ms every 5-s.

Voltage-dependent steady-state inactivation was measured with a series of 500-ms pre-pulses (−120 to 20 mV in 5-mV increments), followed by a 50-ms depolarization to 20 mV (rNav1.8) or −10 mV (rNav1.4 and hNav1.7) to assess the available non-inactivated currents, and the repetition interval was 5-s.

For electrophysiology experiments, the stock solution of rpTx1 was diluted with fresh pipette solution to a concentration in pipette when intracellular application. During extracellular application, effects of rpTx1 on Nav current in heterologous expression cell lines were assessed following incubation with varying concentrations of rpTx1 diluted in extracellular solution for 5–30 min at a holding potential. Concretely, the stock solution of rpTx1 was diluted with fresh bath solution to a concentration of tenfold of the interested concentration, 30 μL of the concentrated peptide was diluted into the recording chamber (containing 270 μL bath solution) far from the recording pipet (the recording cell), and was mixed by repeatedly pipetting to achieve the specified final concentration.

## Cellular uptake study

HEK293T cells were seeded in thin glass-bottomed 35 mm petri dishes at a density of $1.0 \times 10^4$ cells per dish and incubated for 24 h at 37 °C in a 5% $CO_2$. Next, the media were removed and the cells were washed twice with PBS, followed by an incubation with rpTx1-FITC (5 μM in PBS), rpTx1-8K/A-FITC (5 μM in PBS) or FITC-rpTx1-Ntermi (5 μM in PBS) for 30 min at 25 °C. After incubation, the cells were washed three times for 5 min in PBS and then fixed in 4% PFA in PBS for 10 min. Afterward, the cells were stained with DiD (10 μM in PBS) for 30 min at 25 °C. Subsequently, the cells were washed three times in PBS, stained with DAPI (5 μg $mL^{-1}$ in PBS) for 10 min, and finally imaged using a Leica SP8 confocal microscopy (Leica, Germany).

## Surface plasmon resonance (SPR) assay

For the analysis of rpTx1 binding to the IFMT and mutant peptides, SPR experiments were carried out on a Biacore X100 (GE Healthcare, Uppsala, Sweden) with active temperature control at 25 °C, according to the manufacturer's protocols. 100 μg the IFMT peptide or rpTx1 was immobilized by amine coupling onto channel 2 of a CM5 sensor chip (GE Healthcare), and channel 1 was used as the reference flow cell. rpTx1 or the IFMT peptides were diluted in a linear gradient with running buffer (10 mM HEPES, 150 mM NaCl, 3 mM EDTA, 0.05% (v/v) P20, pH 7.4) and made to flow across the immobilized IFMT peptide or rpTx1 for 200 s or 100 s at a flow rate of 10 μL min$^{-1}$ (association), followed by dissociation in the running buffer for 300 s or 50 s, respectively. The resulting data were fitted to a 1:1 binding model using Biacore Evaluation Software (GE Healthcare).

## Western blot analysis

Whole-cell extracts were prepared in RIPA lysis buffer (Sangon), and protein concentration was measured using the BCA Protein Assay Kit (Sangon). Equivalent total proteins from different samples were electrophoresed through 8–12% sodium dodecyl sulfate-polyacrylamide gels, and the proteins were then electro-transferred onto a PVDF membrane. Membranes were blocked with 5% BSA in tris-buffered saline with Tween (TBST) for 1 h at room temperature and incubated with primary antibodies against GFP (rabbit, 1:10,000, Proteintech, 50430-2-AP) for 12 h at 4 °C. Membranes were washed three times with TBST and then incubated with horseradish peroxidase (HRP)-conjugated secondary antibodies for 1 h at room temperature.

## The MD system preparation and equilibration

The solved structures of human Nav1.8 (PDB: 7WEL) were retrieved from the RCSB PDB database (Huang et al, 2022). The Protein Preparation Wizard in Schrödinger 2018 was utilized to remove the irrelevant components, add missing residues and fix bond orders. The structures were then assigned to CHARMM-GUI (Jo et al, 2008) in order to add membrane molecules. The force field parameters for the protein and lipids were generated using the *Antechamber* module in AMBER20 (Case et al, 2005), with the FF14SB force field (Maier et al, 2015) for the protein and the Lipid17 force field for the lipids. The protein, POPC molecules, water and counter ions were assembled using the *tleap* module, and the explicit solvent model (TIP3P) was used for water molecules.

The system was minimized by the *pmemd* program in AMBER20 (Case et al, 2005). Initially, a restraint force of 10.0 kcal·mol$^{-1}$·Å$^{-1}$ was applied to the protein and lipids. The system underwent 5000 cycles of steepest descent minimization followed by 5000 cycles of conjugate gradient minimization to optimize the solvent and ions. Subsequently, the restraint forces on the lipids and protein side chains were removed, and the system underwent an additional 5000 cycles of steepest descent minimization and 5000 cycles of conjugate gradient minimization. Finally, all restraint forces were removed, and the system underwent an additional 5000 cycles of steepest descent minimization and 5000 cycles of conjugate gradient minimization. Next, the heating and equilibration were executed by the GROMACS 2018 package

(Abraham et al, 2015). The minimized system was heated from 0 to 310 K over a period of 2000 ps and then equilibrated for total 15 ns with the Langevin thermostat and Nose-Hoover Langevin pressure control in the NPT (P = 1 atm and T = 310 K) ensemble.

## The metadynamics simulation

The metadynamics sampling approach was employed to produce different binding states between the IFMT motif and the binding pocket. The simulations were executed by GROMACS 2018 with the PLUMED2.5 plugin (Abraham et al, 2015; Tribello et al, 2014). Two collective variables (CVs) were defined to determine the sampling space. CV1 measured the distance between the center-of-mass of atom group A (consisting of the I-F-M residues in the IFMT motif) and atom group B (composed of residues V1270, A1273, L1274, F1421, A1606, N1609, I1610, N1715, I1718, A1719 that form the IFMT binding site). CV2 was set as the contact Q value between atom-pairs within the I-F-M residues of the IFMT motif and the IFMT binding site residues, and only those atom-pairs with native contact (threshold set at 5.0 Å) (Best et al, 2013). The contact Q value for a given conformation X sampled during the metadynamics simulation was calculated as follows:

$$Q(X) = \sum_{i=1}^{m} \frac{1}{1 + e^{\beta(r_i - \lambda r_0)}}$$

where $\beta$ and $\lambda$ were set to 50.0 nm$^{-1}$ and 1.8, respectively, $r_i$ is the center-of-mass distance between $i$th atom-pair, and $r_0$ was set to 5.0 Å. The parameter $m$ is the total number of the native contact atom-pairs in each system, and a total of 84 native pairs were identified for hNav1.8 by our in-house script. The Gaussian widths for CV1 and CV2 were set to 0.15 and 4.0, respectively, with a Gaussian height of 1.0 kJ/mol. The sampling duration extend for 1000 ns. The generation of representative conformations and residue contact analysis were conducted using our in-house scripts.

## The docking-based prediction of rpTx1 binding

Multiple computer-aided techniques were employed to predict the binding patterns between rpTx1 and the IFMT motif of different Nav subtypes. Initially, AlphaFold2 was used to predict the high-confidence three-dimensional structure of rpTx1 (Jumper et al, 2021; Varadi et al, 2022). The representative conformations generated by metadynamics were then assigned to protein-protein docking with rpTx1, with the IFMT motif in different states. The Protein-Protein Docking module in Schrodinger package was utilized to predict the binding complex of rpTx1 and the Nav channel. To generate reasonable docking configurations, a constraint was imposed to restrict the distance between the isoleucine residue of the IFMT motif and rpTx1-T66 within 5.5 Å, and the docking solutions with the confidence scores more than 0.5 were retained for further analysis. Conventional MD simulations were executed to assess the stability of the generated complex of the Nav channel and rpTx1. The minimization, equilibration, and production were carried out by the *pmemd* program in AMBER20 (Salomon-Ferrer et al, 2013). The minimization protocol remained unchanged. Subsequently, each minimized system was gradually heated from 10 to 310 K over 100 ps and equilibrated for 3.0 ns in the NPT ensemble (T = 310 K and P = 1 bar). For each

complex, a 200 ns production run was conducted in the NPT ensemble. The temperature was controlled by the Langevin temperature equilibration scheme with a collision frequency of $1.0 \, \text{ps}^{-1}$. The particle mesh Ewald (PME) (Darden et al, 1993) algorithm was used to handle the long-range electrostatic interactions under the periodic boundary condition, and a cut off of $9.0 \, \text{Å}$ was used for the real-space interactions. The SHAKE algorithm was used to constrain all covalent bonds involving hydrogen atoms, and the time step was set to 2 fs (Ryckaert et al, 1977). Finally, the structural characteristics were analyzed by the *cpptraj* module in AMBER20 (Salomon-Ferrer et al, 2013).

## Data analysis

Data were analyzed with PatchMaster v2x73 (HEKA Elektronik), IgorPro6 (Version 6.1.0.9, WaveMetrics, Lake Oswego, OR, USA), Prism7 (Version 7.00, GraphPad Software), Leica Application Suite X (Version 3.4.2.18368, Leica, Germany), PPSQ Analysis (Version 2.21, Shimadzu, Kyoto, Japan), Biacore Evaluation Software (GE Healthcare) and Office Excel 2010 (Version 14.0.4760.1000, Microsoft, USA). All values are shown as mean ± S.E.M. and n represents the number of cells or animals examined. One-way ANOVA and two-way ANOVA were used to assess the difference between multiple groups. All data were analyzed by the Shapiro-Wilk or D'Agostino & Pearson normality test before t-test analysis. If the data were normally distributed, a parametric t-test was used. Otherwise, a non-parametric t-test was used. In figure legends, statistical method to a specific experiment was mentioned, and p values were also shown. Statistical significance was accepted at a level of $p < 0.05$. Statistical analyses were performed with Prism 7 (Version 7.00, GraphPad) software.

The G–V curves were obtained by calculating the conductance (G) at each voltage (V) using the equation $G = I/(V - V_{rev})$, with $V_{rev}$ being the reversal potential determined for each cell individually. G–V curves were fitted using a Boltzmann equation: $y = 1/(1 + \exp[(V_{1/2} - V)/\kappa])$ in which $V_{1/2}$, V, and κ represent midpoint voltage of kinetics, test potential, and slope factor, respectively.

For steady-state inactivation curves, peak inward currents at the test pulse were normalized to the maximal inward current and fit with Boltzmann functions: $I/I_{max} = A + (1 - A)/\{1 + \exp[(V - V_{1/2})/\kappa]\}$, where V represents the inactivating pre-pulse potential, $V_{1/2}$ is the midpoint of the steady-state fast-inactivation, A is the minimal channel availability, and κ is the slope factor.

Concentration–response curves of toxins were fitted using the following Hill logistic equation: $y = f_{max} - (f_{max} - f_{min})/(1 + (x/EC_{50})^n)$, where $f_{max}$ and $f_{min}$ represent the maximum and minimum response of channel to toxins, $x$ represents toxins concentration and n is an empirical Hill coefficient.

To perform double-mutant cycle analysis, the $EC_{50}$ values of the four channel-toxin interactions (WT channel, WT toxin: $EC_{50\_1}$; mutant channel, WT toxin: $EC_{50\_2}$; WT channel, mutant toxin: $EC_{50\_3}$; mutant channel, mutant toxin: $EC_{50\_4}$) were determined separately. The interaction between residues was determined as $\text{Ln}\Omega$, which is calculated value without units of energy, since multiple toxins, acting with positive cooperativity, exhibit the slope factors for their concentration–response curves that are greater than 1, thus inhibiting the channels. Therefore, the values of $\text{Ln}\Omega$ in this study should not be compared directly to those in the cited literature. $\text{Ln}\Omega = \text{Ln}\left(\frac{EC_{50\_1} * EC_{50\_4}}{EC_{50\_2} * EC_{50\_3}}\right)$.

Association then dissociation curves of IFMT motif with its receptor site were calculated as follows:

$$R_t = \frac{R_{max} * C}{K_D + C}\left(1 - \frac{1}{e^{\left(K_{on} * C + K_{off}\right) * t}}\right) \text{ (association)}$$

$$R_t = R_0 * e^{-K_{off} * t} \text{ (dissociation)}$$

$$K_D = K_{off}/K_{on}$$

Where $K_{on}$ and $K_{off}$ represent the association and dissociation in inverse time units, respectively; $t$ is the time; $R_{max}$ represents maximum binding at equilibrium with maximum concentration of ligand; Since the binding of the IFMT motif (ligand) to its receptor site conforms to an absolute 1:1 ratio in this study, we set the ligand concentration of rNav1.4, hNav1.5, and rNav1.8 to a constant C.

## Data availability

This study includes no data deposited in external repositories.

The source data of this paper are collected in the following database record: biostudies:S-SCDT-10_1038-S44318-025-00438-9.

## Peer review information

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

## Acknowledgements

We would like to thank Dr. Lei Xu (Institute of Bioinformatics and Medical Engineering, School of Electrical and Information Engineering, Jiangsu University of Technology, Changzhou, China) for his assistance with the software and computational infrastructure, as well as the valuable feedback on docking simulations and data analysis. This work was supported by funding from the National Natural Science Foundation of China (Grant Numbers 32371322 to XZ, 32271329 and 32071262 to ZHL, 82204280 to HYC, and 31872718 to SPL); the Science and Technology Innovation Program of Hunan Province (Grant Numbers 2021RC3092 to XZ, and 2020RC4023 to ZHL); the National Key R&D Program of China (Grant Numbers 2023YFF1304900 to ZHL); the Research Foundation of the Education Department of Hunan Province (Grant Numbers 24A0059 to XZ, and 24B0052 to MZC); the Scientific Research Program of FuRong Laboratory (Grant Numbers 2023SK2096 to ZHL).

## Author contributions

**Xi Zhou**: Conceptualization; Data curation; Formal analysis; Supervision; Funding acquisition; Validation; Investigation; Writing—original draft; Project administration; Writing—review and editing. **Haiyi Chen**: Data curation; Funding acquisition; Investigation; Writing—original draft; Writing—review and editing. **Shuijiao Peng**: Data curation; Investigation. **Yuxin Si**: Data curation; Investigation. **Gaoang Wang**: Investigation. **Li Yang**: Investigation. **Qing Zhou**: Investigation. **Minjuan Lu**: Investigation. **Qiaoling Xie**: Investigation. **Xi He**: Investigation. **Meijing Wu**: Investigation. **Xin Xiao**: Investigation. **Xiaoqing Luo**: Investigation. **Xujun Feng**: Investigation. **Wenxing Wang**: Investigation. **Sen Luo**: Investigation. **Yaqi Li**: Investigation. **Jiaxin Qin**: Investigation. **Minzhi Chen**: Investigation. **Qianqian Zhang**: Investigation. **Weijun Hu**: Investigation. **Songping Liang**: Funding acquisition; Validation. **Tingjun Hou**: Supervision; Funding acquisition; Project administration; Writing—review and editing. **Zhonghua Liu**: Conceptualization; Funding acquisition; Project administration; Writing—review and editing.

Source data underlying figure panels in this paper may have individual authorship assigned. Where available, figure panel/source data authorship is listed in the following database record: biostudies:S-SCDT-10_1038-S44318-025-00438-9.

## Disclosure and competing interests statement

The authors declare no competing interests.

# Expanded View Figures

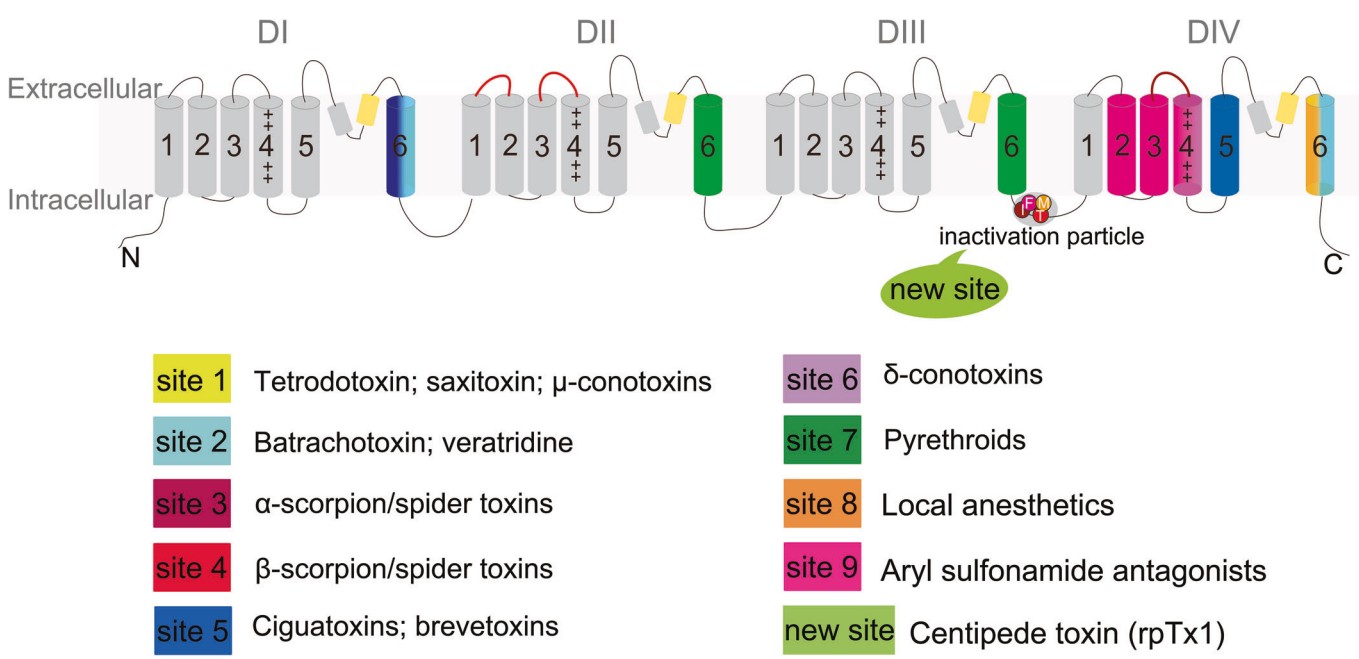

**Figure EV1.  Neurotoxin/drug receptor sites on Nav channels.**

The IFMT motif is a new neurotoxin binding site identified in this study (new site). Adapted from Catterall et al (2007)[21] and Klint et al (2012)[14].

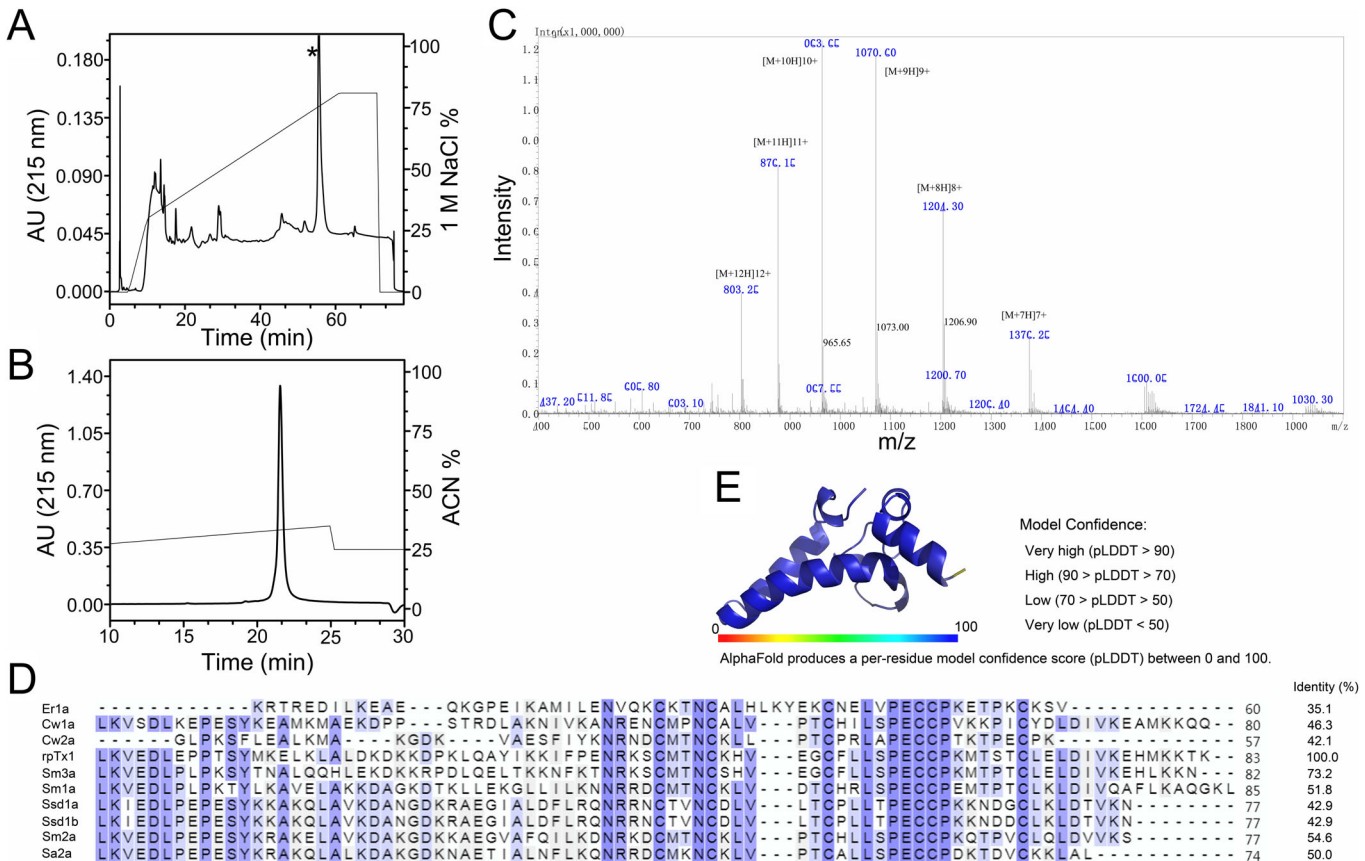

**Figure EV2. Purification and characterization of rpTx1.**

(A) RpTx1 was purified to homogeneity by cation-exchange HPLC. (B) Further purification and desalting of the active fraction from (A) by RP-HPLC. (C) ESI mass spectra of the purified rpTx1. (D) Sequence alignment of rpTx1 with several centipede toxins. (E) High or very high confidence scores are yielded for the rpTx1 structure predicted by Alphafold2.

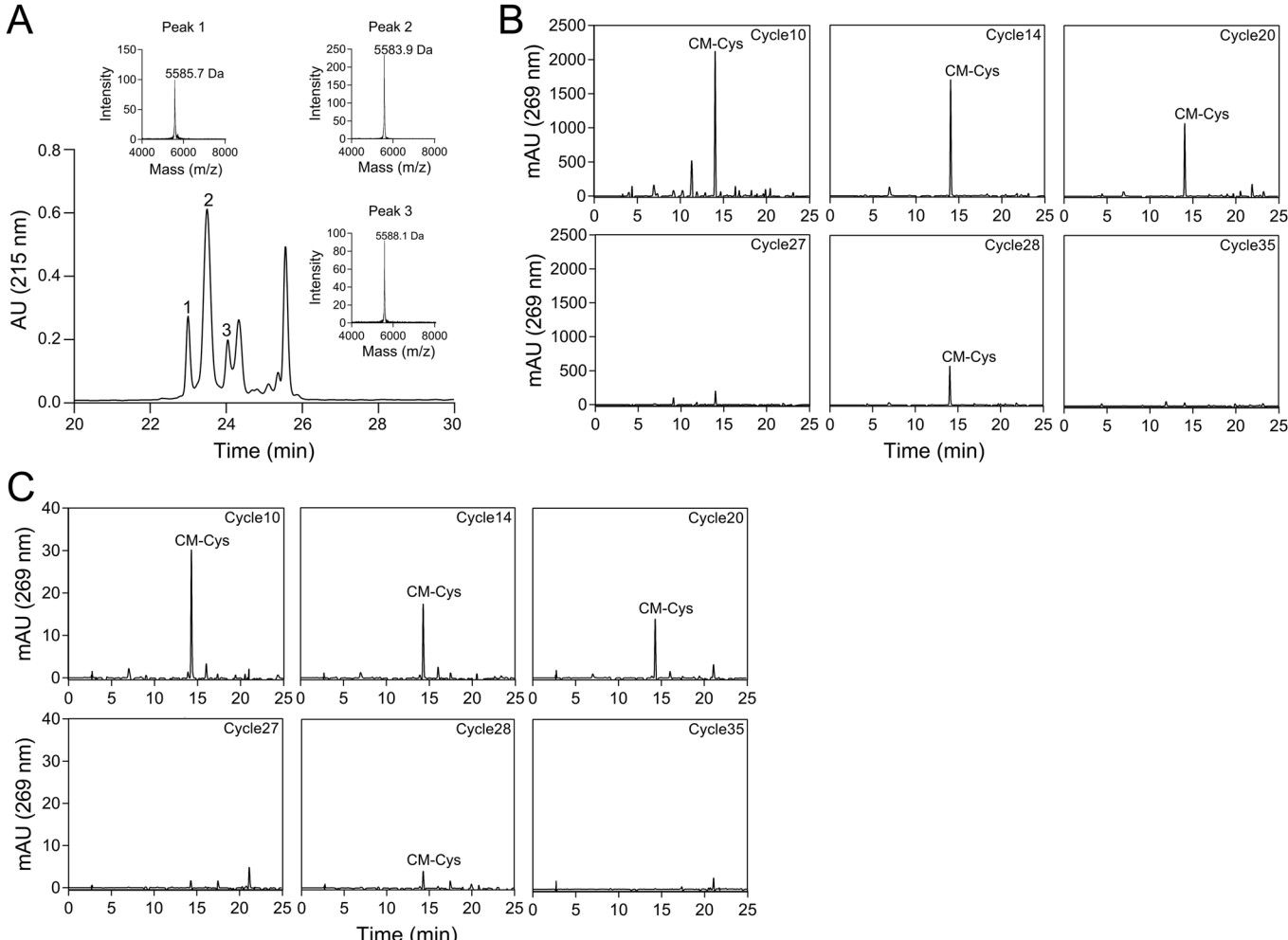

**Figure EV3. The determination of disulfide bonds of rpTx1.**

(A) RP-HPLC chromatogram of the partially reduced products of the truncated rpTx1 (sequence in the section of method), and the MALDI-TOF MS analysis of fractions labeled (insets). These results indicate that the peak 1 is a peptide with one disulfide bond reduced, peak 2 is nonreduced peptide, and peak 3 is a two disulfide bonds reduced peptide. (B, C) Edman degradation sequencing of the partially reduced peptides after their free sulfydryl groups were alkylated by iodoacetamide. Cysteine residues occur at cycle 10, 14, 20, 27, 28, and 35, respectively, indicating on the upper right of panel. (B) The cysteine residue cycles of the alkylated peak 3, a rpTx1 analogue with one disulfide bond, and signals of the alkylated Cys residues (PTH-CM-Cys) are observed in cycle 10, 14, 20 and 28, but not in cycle 27 and 35, indicating that the two disulfide bonds formed by cystine10, 14, 20 and 28 were reduced by TCEP, while the remained disulfide bond Cys27–Cys35 was kept intact. Additionally, we analyzed the decay rates of the four PTH-CM-Cys signals in the Peak 3 Edman degradation sequencing, showing that compared to cycle 10, the signal yield was attenuated by 19% for cycle 14, 50% for cycle 20, and 73% for cycle 28. This is likely the normal signal decay occurring during sequencing. (C) Next, we continued to determine the second disulfide bond of rpTx1 by the sequencing of alkylated Peak 1, a rpTx1 analogue with two disulfide bonds. Theoretically, PTH-CM-Cys signals should only appear in two cycles because this peak has only one disulfide bond reduced and alkylated. However, PTH-CM-Cys signals are observed in the chromatogram at the cycle 10, 14, 20, and 28, suggesting that this peak should be a mixture of two fractions, each containing two disulfide bonds, one of which is Cys27–Cys35, and the other one should be different. We attempted to infer the disulfide bond reduced in the two fractions based on the signal intensities of the four PTH-CM-Cys which should correlate with the relative abundance of the two fractions in the mixture and the normal signal decay during sequencing. compared to cycle 10, the signal yield was attenuated by 42% for cycle 14, 54% for cycle 20, and 87% for cycle 28, which shows different pattern from that shown in (B). Specifically, compared to Cycle 10, the PTH-CM-Cys signal decays in Cycles 14 and 28 are significantly greater than the normal decay, while the signal decay in Cycle 20 is similar to the normal one. This suggests that the PTH-CM-Cys signals in Cycle 10 and Cycles 14 and 28 originate from two different fractions, with the fraction containing the Cycle 10 PTH-CM-Cys signal being more abundant than the fraction containing the PTH-CM-Cys signals from Cycles 14 and 28, while the PTH-CM-Cys signals in Cycle 10 and Cycle 20 likely originate from the same fraction. Based on these results, it can be inferred that Cys10 and Cys20, as well as Cys14 and Cys28, respectively, form disulfide bonds.

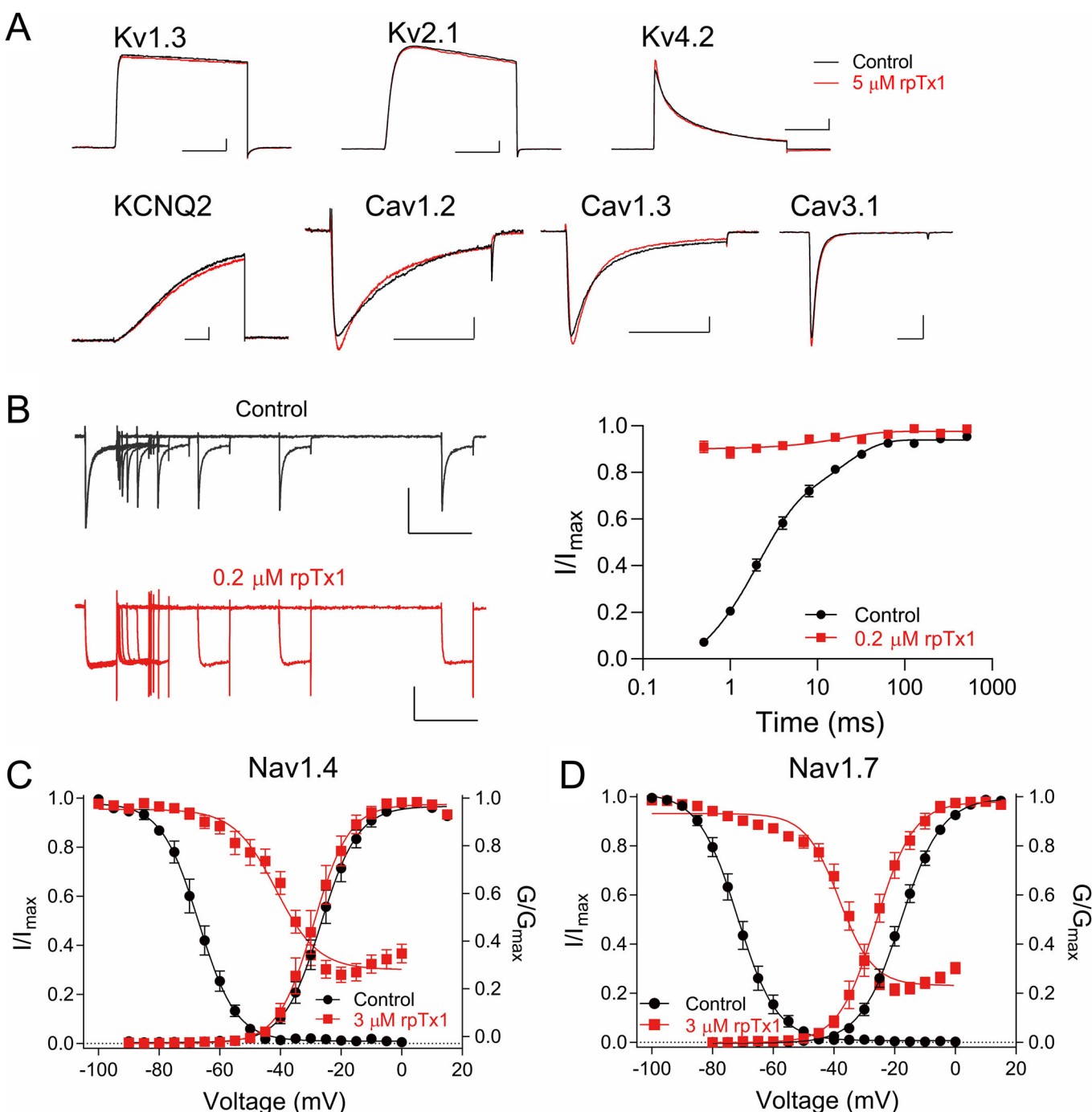

Figure EV4.  The effects of rpTx1 on other ion channels, on the fast inactivation recovery of rNav1.8, and on the steady-state activation and inactivation of rNav1.4 and hNav1.7.

(A) RpTx1 has no effect on the currents of voltage-gated potassium channels (mKv1.3, rKv2.1, rKv4.2, and hKCNQ2) or voltage-gated calcium channels (mCav1.2, rCav1.3, and rCav3.1) ($n = 3$–5 per group). Scale bar: 100 pA/pF, 100 ms (upper panel), and 20 pA/pF, 50 ms (lower panel). (B) Left: representative inactivation recovery current traces of rNav1.8 channels in the absence or presence of 0.2 μM rpTx1. Right: time course of recovery from the fast inactivation of rNav1.8 in the absence ($n = 9$) or presence ($n = 5$) of 0.2 μM rpTx1. Scale bar: 1 nA, 100 ms. (C, D) The effect of rpTx1 on the steady-state activation (rNav1.4: $n = 9$ for control, $n = 8$ for rpTx1; hNav1.7: $n = 12$ for control, $n = 7$ for rpTx1) and inactivation (rNav1.4: $n = 6$ for control, $n = 8$ for rpTx1; hNav1.7: $n = 14$ for control, $n = 7$ for rpTx1) of rNav1.4 (B) and hNav1.7 (C). Data are presented as mean ± S.E.M.

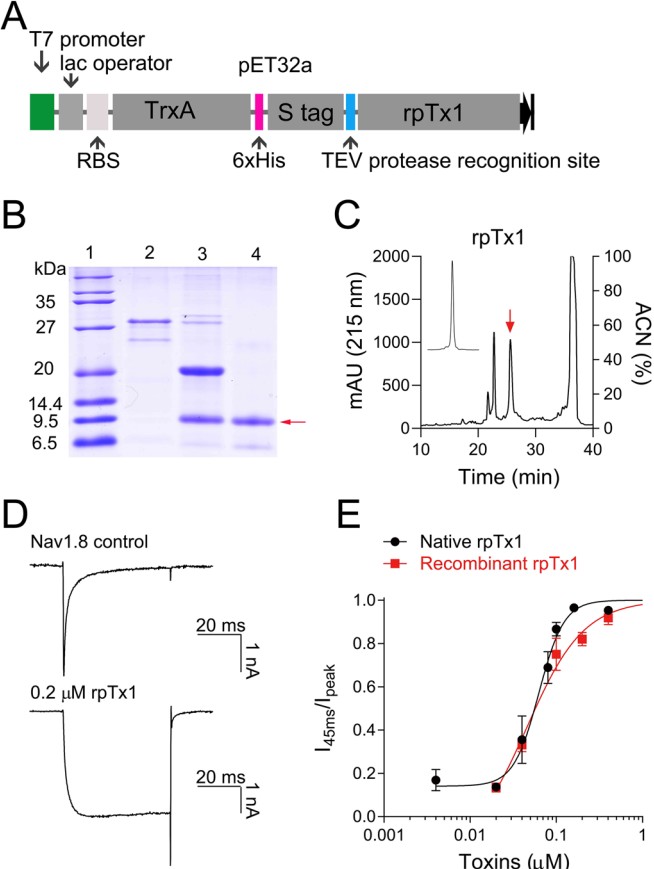

**Figure EV5. Recombinant production of rpTx1.**

(A) Architecture of the pET32a vector used for recombinant expression of rpTx1. The coding region includes a TrxA protein, a 6×His tag, a S-tag, a TEV protease recognition site, and a codon-optimized gene encoding rpTx1. (B) Tricine-SDS-PAGE analysis of rpTx1 expressed in BL21 (DE3). Lane 1, molecular weight markers; lane 2, elution fraction with 250 mM imidazole; lane 3, fusion protein after TEV protease cleavage; lane 4, eluted rpTx1 from RP-HPLC. (C) RP-HPLC characterization of recombinant rpTx1. The red arrow-labeled peak indicates recombinant rpTx1. Recombinant rpTx1 was further purified to homogeneity by analytical RP-HPLC (inset). (D) Representative traces of current traces recorded from ND7/23 cells expressing rNav1.8 in the absence or presence of 0.2 μM recombinant rpTx1 in intracellular solution. (E) The concentration–response curves of recombinant rpTx1 and native rpTx1 inhibiting the fast inactivation of rNav1.8 ($n = 3$–10 per concentration). Data are presented as mean ± S.E.M.

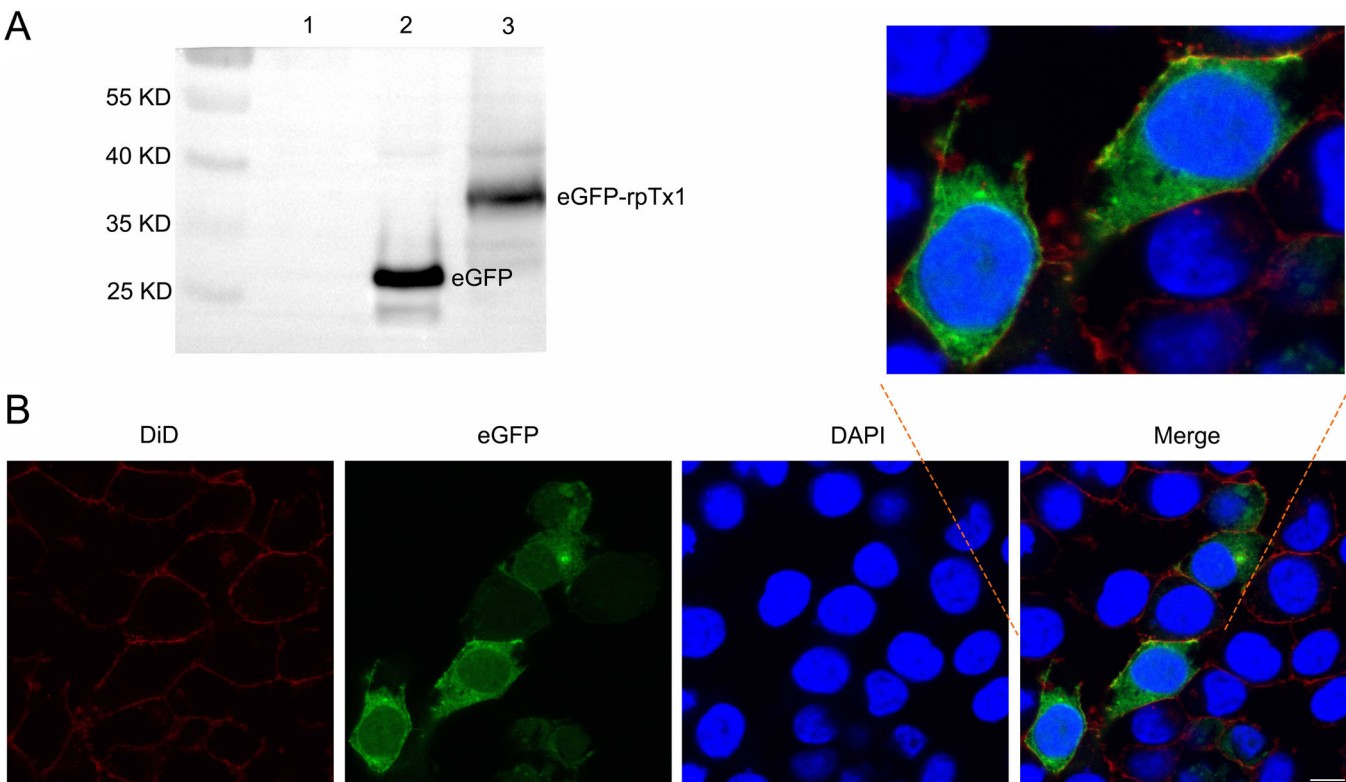

**Figure EV6.   The eGFP-rpTx1 fusion protein maintained rpTx1 on the intracellular side of cells.**

(**A**) Representative western blotting images of recombinant eGFP in cell lysates from HEK293T cells (line 1), HEK293T cells expressing eGFP (line 2), and HEK293T cells expressing eGFP-rpTx1 (line 3), respectively ($n = 3$). (**B**) Confocal images show that eGFP-rpTx1 is primarily distributed within the cytoplasm. DiD and DAPI were used as membranous and nuclei markers, respectively. Scale bar: 10 μm.

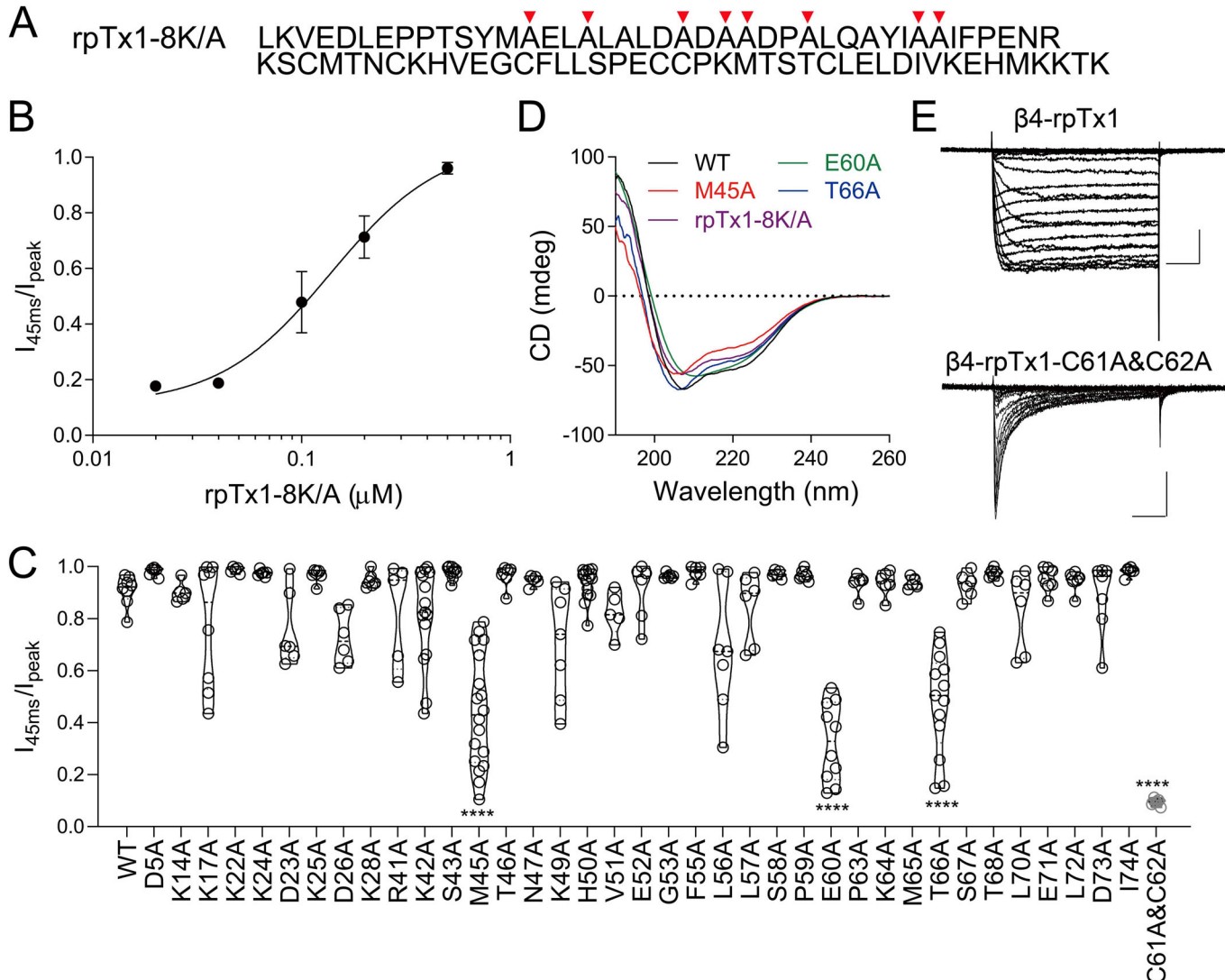

**Figure EV7. The effect of rpTx1 mutants on the fast inactivation of rNav1.8.**

(A) The sequence of rpTx1-8K/A with mutation sites labeled by red arrows. (B) Concentration-dependent curves show the effect of the intracellular application of rpTx1-8K/A on rNav1.8 ($n = 4$–11 per concentration). (C) Potency of 0.1 µM WT rpTx1 and mutants measured on rNav1.8. Note that M45A, E60A, T66A and C61A&C62A mutations remarkably reduced toxin's availability on the channel (one-way ANOVA with post hoc analysis using Dunnett's multiple comparisons test, $n = 5$–18). (D) CD spectra of WT and rpTx1 mutants. (E) Representative traces of current families were recorded from ND7/23 cells expressing rNav1.8 co-expressed with 3.5 µg β4-rpTx1 ($n = 5$) or β4-rpTx1-C61A&C62A ($n = 6$). Scale bar: 0.5 nA, 10 ms. Data are presented as mean ± S.E.M. ****$p < 0.0001$.

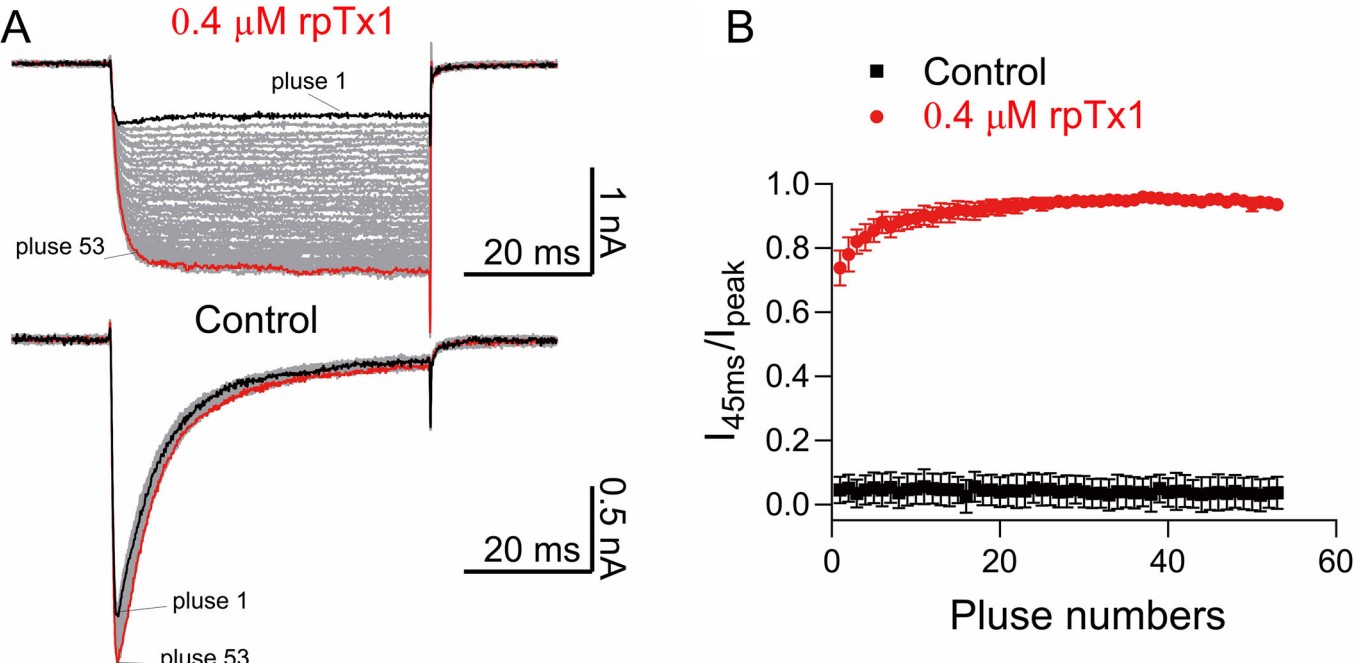

**Figure EV8. RpTx1 rapidly inhibits the fast inactivation of rNav1.8.**

(A) Representative current traces of repetitive test pulse at 20 mV from the holding potential at −90 mV at 0.2 Hz frequency in the presence (upper) or absence (lower) of 0.4 µM rpTx1 in pipette. Patch-clamp recordings were performed immediately after establishing a whole-cell configuration on ND7/23 cells expressing rNav1.8. (B) Time-course of inhibition of rNav1.8's fast inactivation by 0.4 µM rpTx1 ($n = 6$ for rpTx1, $n = 5$ for control).

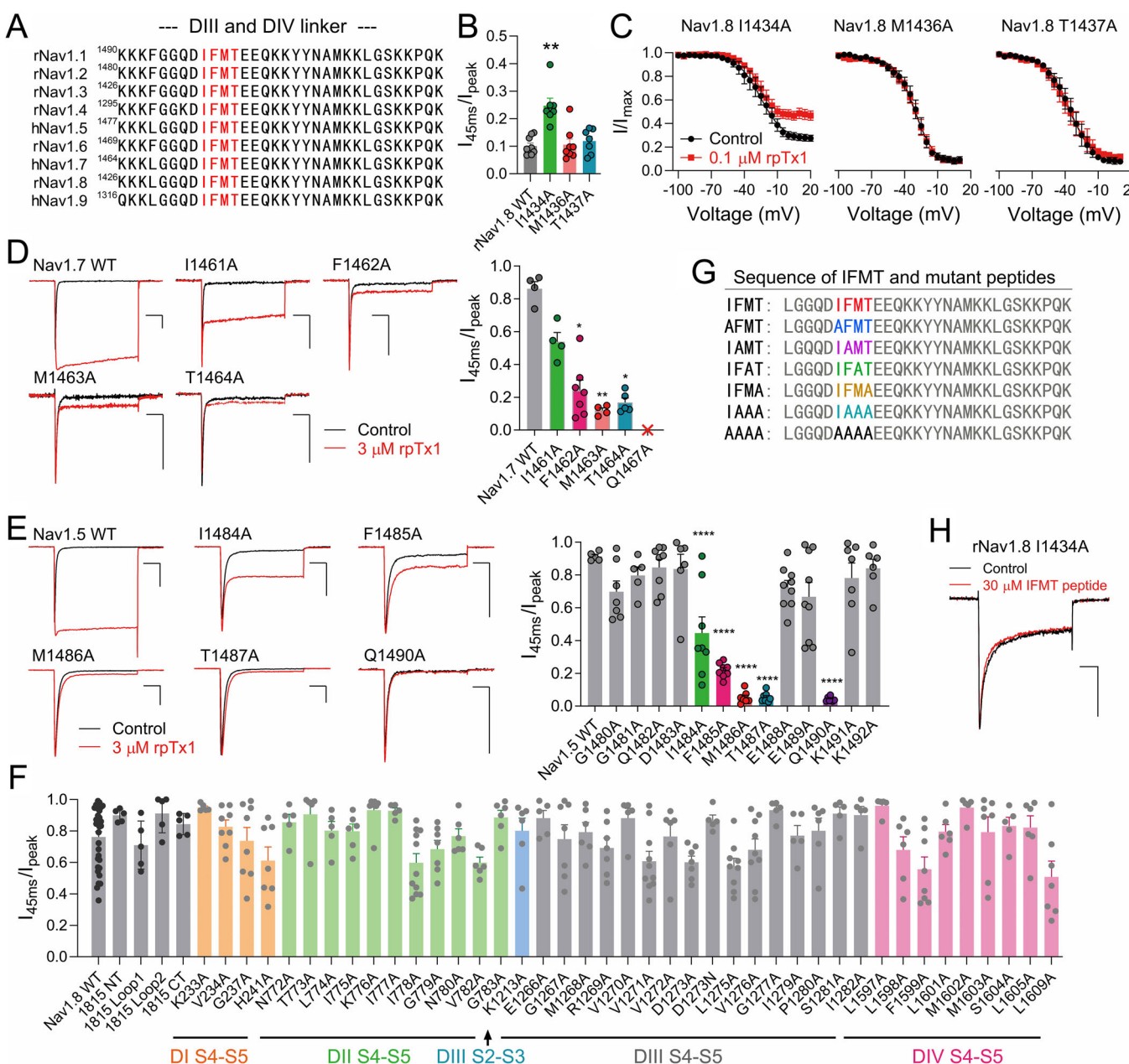

**Figure EV9.  The IFMT motif of Nav channels is the key region for the action of rpTx1.**

(A) Sequence alignments corresponding to the fast inactivation gate region (DIII–DIV linker) of Nav channel subtypes. The red highlighted sequences show the IFMT motif. (B) The scatter dot plot shows the effect of rNav1.8 mutants I1434A ($n = 7$), M1436A ($n = 8$), and T1437A ($n = 7$) on the fast inactivation of the channel (Non-parametric test using Dunn's multiple comparisons test). (C) The effect of rpTx1 in the pipette on the steady-state inactivation of rNav1.8 mutants I1434A ($n = 9$ for control, $n = 8$ for rpTx1), M1436A ($n = 5$ for control, $n = 6$ for rpTx1) and T1437A ($n = 5$ for control, $n = 6$ for rpTx1). (D) Representative current traces from WT and mutant hNav1.7 channels in the absence or presence of 3 μM rpTx1 in pipette. Scatter dot plot show the effect of 3 μM rpTx1 on the persistent currents ($I_{45\,ms}/I_{peak}$) of WT and mutant hNav1.7 channels (Non-parametric test using Dunn's multiple comparisons test, $n = 3$–7). Scale bar: 1 nA, 10 ms. (E) Representative current traces from WT and mutant hNav1.5 channels in the absence or presence of 3 μM rpTx1in pipette. Scatter dot plot show the effect of 3 μM rpTx1 on the persistent currents ($I_{45\,ms}/I_{peak}$) of WT and mutant hNav1.5 channels (one-way ANOVA with post hoc analysis using Dunnett's multiple comparisons test, $n = 5$–9). Scale bar: 2 nA, 10 ms. (F) Scatter dot plot show the effect of 0.1 μM rpTx1in pipette on the persistent currents ($I_{45\,ms}/I_{peak}$) of WT and mutant rNav1.8 channels (Non-parametric test using Dunn's multiple comparisons test, $n = 5$–32). (G) Sequences of the IFMT peptide and its 6 mutants. The mutated amino acid residues are highlighted in colors. (H) Representative current traces showing the effect of 30 μM IFMT peptide on rNav1.8 I1434A upon intracellular application ($n = 4$). Scale bar: 1 nA, 10 ms. Data are presented as mean ± S.E.M. **$p < 0.01$, ***$p < 0.001$, ****$p < 0.0001$.

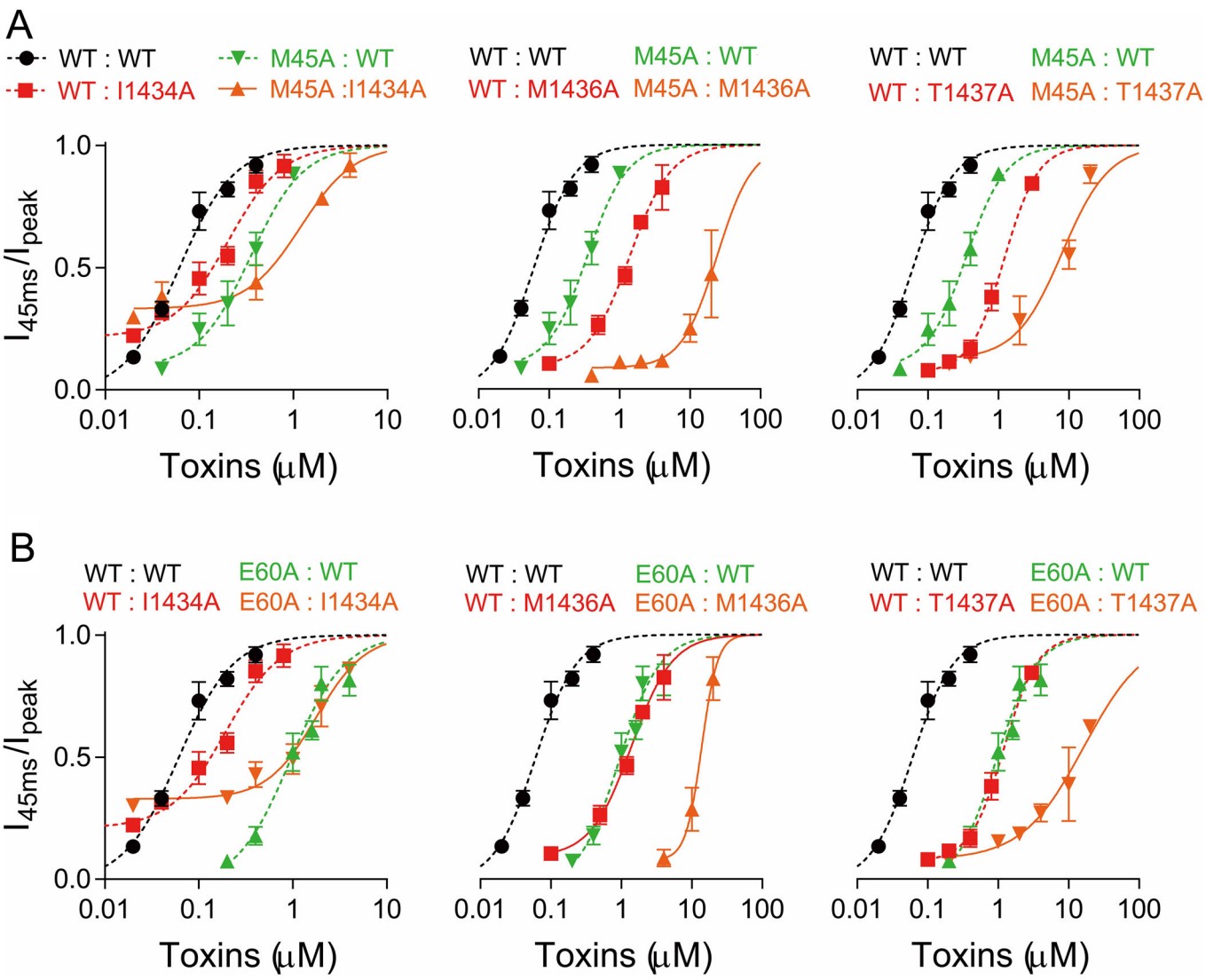

**Figure EV10.** **Mutant cycle analysis for pairwise coupling between rpTx1 and rNav1.8.**

Concentration–response curves for determining the interaction Ln(Ω) values between the rpTx1 M45 and rNav1.8 I1434 or M1436 or T1437 pair (**A**), the rpTx1 E60 and rNav1.8 I1434 or M1436 or T1437 pair (**B**). $n = 3$–17 per concentration. Data are presented as mean ± S.E.M.

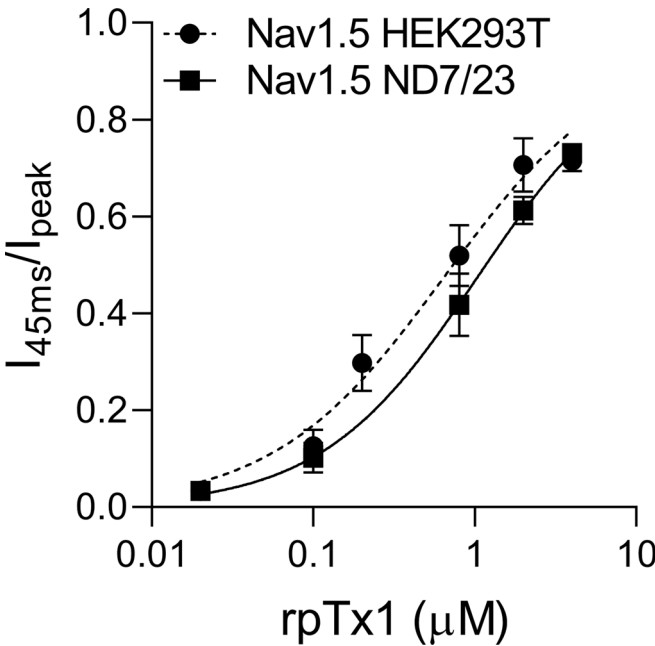

**Figure EV11.   The effect of rpTx1 on the activity of hNav1.5 expressed in different cell types.**

The concentration-dependent curves show the effect of rpTx1 in pipette on hNav1.5 expressed in either HEK293T or ND7/23 cells ($n = 3$–8 per concentration). Data are presented as mean ± S.E.M.

