## [Peer Review File · The EMBO Journal]

Inherent fast inactivation particle of Nav channels as a new binding site for a neurotoxin

Zhonghua Liu, Xi Zhou, Haiyi Chen, Shuijiao Peng, Yuxin Si, Gaoang Wang, Li Yang, Qing Zhou, Minjuan Lu, Meijing Wu, Xin Xiao, Xiaoqing Luo, Xujun Feng, Sen Luo, Yaqi Li, Jiaxin Qin, Minzhi Chen, qianqian Zhang, Weijun Hu, Songping Liang, Tingjun Hou, Qiaoling Xie, Xi He, and Wenxing Wang

Corresponding authors: Zhonghua Liu (liuzh@hunnu.edu.cn) , Tingjun Hou (tingjunhou@zju.edu.cn), Xi Zhou (xizh@hunnu.edu.cn)

Review Timeline:

Submission Date:	30th Sep 24
Editorial Decision:	28th Oct 24
Revision Received:	30th Jan 25
Editorial Decision:	26th Feb 25
Revision Received:	1st Mar 25
Accepted:	4th Mar 25

Editor: Yehu Moran

Transaction Report:

Dear Prof. Liu,

Thank you for submitting your manuscript for consideration by the EMBO Journal. It has now been seen by three referees whose comments are shown below.

Should you be able to address these criticisms in full, we could consider a revised manuscript. I should remind you that it is EMBO Journal policy to allow a single round of revision only and that, therefore, acceptance or rejection of the manuscript will depend on the completeness of your responses in this revised version. I do realize that addressing all the referees' criticisms will require a lot of additional time and effort and be technically challenging. I would therefore understand if you wish to publish the manuscript more rapidly elsewhere, in which case please let us know so we can withdraw it from our system.

Please note I also asked the referees to read each other's reports and try to suggest a more focused work plan that they believe can be achievable in 3 months and address the major concerns. Two of them provided such additional comments. I provide these below the initial independent reports.

I strongly suggest that you consider the feasibility of fitting all these experiments in the time frame of 3 months and if you wish to go for a revision, send me in advance an email where you detail your revision plan (with emphasis on additional experiments and how they address the referees' concerns) so we are aware of it. If something that was suggested is not doable due to any technical limitations, you should explain in detail why it is not doable.

If you decide to thoroughly revise the manuscript for the EMBO Journal, please include a detailed point-by-point response to the referees' comments. Please bear in mind that this will form part of the Review Process File, and will therefore be available online to the community. For more details on our Transparent Editorial Process, please visit our website: <https://www.embo.org/embo-press>

Thank you for the opportunity to consider your work for publication. I look forward to your revision.

Yours sincerely,

Yehu Moran
Academic Editor
The EMBO Journal

- a point-by-point response to the referees' comments, with a detailed description of the changes made (as a word file).
- a word file of the manuscript text
- individual production quality figure files (one file per figure)
- a complete author checklist, which you can download from our author guidelines (<https://www.embopress.org/page/journal/14602075/authorguide>).
- Expanded View files (replacing Supplementary Information)

The revision must be submitted online within 90 days; please click on the link below to submit the revision online before 26th Jan 2025.

Referee #1:

Although description of a novel mechanism of action of peptide toxins at sodium channels could potentially be interesting, there are some questions regarding the validity of the conclusions, as well as some minor comments generally.

The definition of the toxin binding sites is meanwhile outdated - there are several other publications that have laid claim to a site "10".

Was the peptide synthesised in one step?

How was the peptide discovered in the first place? If there is no activity at 5 minutes it seems a fortuitous coincidence that the peptide was found at all.

It is not clear at all why fusion to the C-terminus of eGFP should tether the peptide; nor is there any evidence that it is tethered. Similarly, the strategy used to fuse the peptide to b1 is not clear, nor is there any evidence of the peptide being "maintained" on the intracellular side of cells. The interactions of b1, b2 and b4 with the sodium channel are quite different, and it seems implausible that the peptide would be able to reach a common binding site without a cleverly designed linker. Please elaborate on the design strategy.

Please show a time course of peptide activity following extracellular application.

It is also not clear why treatment with 5 μ M recombinant peptide (for how long?) only caused a response in a small subset of cells (Fig 2c). This is in contrast to the near uniform permeability implied by the experiment with FITC-labelled peptide. This is also in contrast to the effect observed after 30 minutes with NaV1.7 (unless a responding cell was cherry-picked).

The labelling is also confusing - is rpTx1 the same as rpTx1-N? presumably it is the N-terminal half, what was the exact sequence of this and the C-terminal peptide?

While identifying residues that contribute to activity is interesting, a 6-22 fold reduction in potency could hardly be termed "critical".

Importantly, the view that the IFM motif binds to a region in the pore is a now outdated view, with the binding site having been identified to lie adjacent to the pore. This makes activity of both b1 and b2-fused peptide even more surprising as a longer linker would be expected to be required.

The concentration-response curves shown in Fig 3 C are concerning. For example, Fig. 3A shows that the I1434A mutant has decreased fast inactivation (consistent with this residue being part of the IFM cluster important for fast inactivation). Indeed, this mutant has a persistent current shown as ~ 0.5 of peak. (As a side note, what is missing is quantification of the effect of these mutations on persistent current.) Now, a very low concentration of peptide should accordingly be similar to control, i.e. ~ 0.5 I45/Ipeak - but what is shown in the CRC is ~ 0.2 . There are two possibilities, either the representative current trace isn't very representative, or the inactivation particle of NaV1.8 actually does not contribute much to inactivation.

As an additional control - there are some very good antibodies against the inactivation particle. What is the effect of this antibody (pan-NaV, e.g. from Sigma Aldrich) on persistent current? And does the antibody still bind in presence of rpTx1?

The schematic in Fig 6a is very confusing; but if the hypothesis is that rpTx1 prevents transition from the open to inactivated state then some kinetic electrophysiological studies should be conducted that can address these questions - eg recovery from inactivation. The voltage protocol used in Fig 6 are not clear.

In terms of proposed mechanism - although it could make sense that the apparent potency of rpTx1 is increased at NaV1.8 because the inactivation particle binds less strongly to the receptor site. However, by that logic, the inactivation particle of NaV1.9 would bind even less strongly to the receptor site, and the peptide should be even more potent.

Was Fig 1e) obtained following extra- or intracellular peptide application?

Please provide more information on FITC labelling: what is the reaction, what was the efficiency, where was FITC attached?

There is no information on the hNaV1.9 clone. It is also notable that no other lab in the world has successfully managed to express hNaV1.9; including some of world's leaders on NaV channels. Can you elaborate on why this clone expresses, and whether or not anyone else is able to replicate your findings? The lack of reproducibility of this astounding result, particularly the current size, is somewhat concerning.

The manuscript will need to be proof-read for language carefully as there are several errors throughout.

Referee #2:

Voltage-gated sodium channels (Navs) are critical players in neurotransmission. Animal toxins have long been used to understand Nav structure and function. Fast inactivation is a hallmark of the kinetics of mammalian Navs and is critical for repetitive firing of action potentials. The short intracellular linker between DIII and DIV (DIII-DIV linker) is a key structural element

for fast inactivation with a hydrophobic motif Ile-Phe-Met-Thr (the IFMT motif), located in the N-terminal of this linker, known as the "inactivation particle". This study identified a novel centipede toxin, rpTx1 from the Chinese red-headed centipede *Scolopendra subspinipes mutilans*, that appears to penetrate the cell membrane (via an N-terminal alpha helical domain) and specifically binds to the inactivation particle to inhibit the fast inactivation of Nav channels (via a C-terminal toxin domain). As such, this toxin represents a novel tool to study Nav channel inactivation and represents a beautiful example of the diverse mechanisms evolved by venomous animals to subdue their prey and defend against predators. This is a very exciting and relevant study that will be of broad interest to the EMBO J readership and the scientific community. However, I have several major concerns that need to be addressed before accepting this work for publication. Briefly, the result section lacks details and some of the claims are not well supported by rigorous data. This includes the claim that rpTx1 crosses the cell membrane. The electrophysiological data strongly suggests that this is indeed the case but the authors fail to provide compelling data using an orthogonal approach. Since cell penetration is one of the key findings of this study, there has to be strong unambiguous data supporting this finding. The FITC-labeled HEK cell penetration data is weak and not convincing. Additional reasons for why there needs to be strong supporting data is that the discovered toxin could represent an unprocessed precursor of the actual mature toxin that only consists of the disulfide rich domain. There is an obvious dibasic cleavage site (RK) separating the two domains, suggesting that the toxin may be further processed in the venom. Furthermore, the toxin appears to only constitute a very small fraction of the HPLC trace and the authors do not discuss how the venom was extracted (was the entire gland lysed?) and whether they also found or at least looked for a version of the toxin lacking the N-terminus. The electrophysiological data on the action of the toxin on the intracellular domain appears convincing but toxins can get inside cells by the action of other toxins present in the venom cocktail. It is also somewhat surprising that such a rather big toxin can diffuse into the cell via linkage to a helical peptide. Given all these observations, there needs to be strong data supporting the claim that rpTx1 can indeed efficiently penetrate the cell membrane. There are various methods to quantitatively measure cell penetration and the authors need to show at least one of these methods in a robust manner. A more detailed explanations of this and my other concerns is provided below.

Major concerns essential to be addressed to support the conclusions:

The results lack a lot of the information needed to evaluate the claims made by the authors. While some of the information is provided in the method and SI sections it is difficult for the reader to find this information and it should therefore be included in the result section alongside the claims.

Specific comments:

1. The language should be changed in a couple of places to avoid misleading interpretations of the significance of the results. Currently, the following two sentences could suggest that the toxin identified a previously unknown site on the channel:

Lines 25-27: "This study revealed the inherent inactivation particle of Nav channels as a new neurotoxin binding site, utilizing a novel centipede peptide toxin called rpTx1."

Change to: Utilizing a novel centipede peptide toxin called rpTx1, this study revealed that the inherent inactivation particle of Nav channels represent a binding site for a neurotoxin.

Lines 395-397: "we identified the inherent inactivation particle of Nav channels as a new receptor site for a centipede peptide toxin that can modify Nav channel activity."

Change to: we identified that the centipede toxin modifies Nav channel activity via binding to the inactivation particle.

2. Similarly, the title needs to be changed to clarify that the study identified a single toxin with this mode of action. Suggesting the existence of other toxins is fine but does not warrant the use of plural in the title. Change "neurotoxins" to "a neurotoxin"

3. The result section lacks important details and, in some places, experiments lack rigor and results/conclusions are questionable.

For example, a major concern is the lack of information on how the new toxin relates to previously described, similar sequences. The authors state that BLAST did not identify similar sequences but some similar sequences are shown in Figure S2d but the source of these sequences is not provided. Were any of these characterized before? This needs to be better described for transparency.

Another example are the results for the FITC-labeled cell penetration assay. These are not well described and somewhat confusing. Since cell-penetration is an essential claim of the paper the FITC data needs to be more thoroughly analyzed or repeated with higher confidence. Only very few cells were analyzed and data is not quantitative and completely lacks any statistical analysis.

Why is there no quantitative comparison between the data for rpTx1, rpTx1-8K/A, and rpTx-N? If anything, currently it seems that rpTx1 had better cell penetration than rpTx-N and the image for rpTx-N looks like the experiment may not have worked. Why were such few cells analyzed and how could such a small cell number possibly give statistically significant results (i.e., 8/22 cells for 5uM rpTx1 and 2/15 cells for 1uM)? Perhaps this is why there is no statistical analysis here.

4. The identification of potentially similar two-domain toxins as presented in the discussion and Figure 11 is intriguing but very immature and not supported by any data. This part should be removed from the paper given the complete lack of evidence of dual action in any of the peptides shown in Figure 11. For example, it has long been known that cone snail toxins are translated as propeptides that contain an N-terminal pro-region. Several roles have been attributed to this region (assistance in folding, binding of modifying enzymes, and binding of secretory trafficking molecules). Using alphafold these proregions that can contain positive charges are often predicted to be helical which could suggest that these may penetrate cells, although all known cone snail toxins have extracellular targets. This is not to say that the mechanism described for rpTx1 does not exist in cone snail toxins and other toxins but it is to say that seemingly having two domains like rpTx1 on the predicted propeptide region of a toxin is NOT suggestive of the same mode of action as seen for rpTx1. Again, in the absence of any evidence, figure 11 needs to be removed.

Instead, the authors could suggest that similar sequences to rpTx1 from related centipede species, as shown in figure S2d, may act like rpTx1 and that other toxins with similar mechanisms may also exist in other animal venoms.

Minor concerns that should be addressed:

1. General: Insert space before citations.

2. Introduction: When discussing the usefulness of toxins that target intracellular sites, it would be useful to list a few examples of other toxins that target other ion channels at intracellular sites. These are discussed in the discussion section but could already be mentioned in the introduction.

3. Results: It is difficult to see what HPLC fraction the asterisk in Figure 1a belongs to. Clarify this by using an arrow or something else that more clearly shows the active fraction.

4. Results: In the caption of Figure 1a include information on how much venom was loaded on this HPLC run and how much of this was estimated to be rpTx1. Provide information on whether the venom was "milked" or extracted from homogenized glands? Also provide information on whether the toxin was found in its "truncated version". There's a dibasic cleavage site between the two domains and it would not be surprising but highly relevant if the C terminus "toxin"-like part was found without the N-terminal "carrier".

5. Results: 5'RACE requires information on related DNA sequences for at least one end. What sequences were used as the basis of primer design (degenerative primer)? This information should be provided in the results. What portion of the sequence was confidently identified by Edman sequencing? Presumably not all residues and not the entire toxin was sequenced by Edman. Similarly, RACE may not have provided the entire toxin region. This information should be provided.

6. Results: The rationale for naming the toxin is unclear ("We named it rpTx1 (rational nomenclature: δ -SLPTX10-Ssm1a(King, Gentz et al., 2008)).") and should be better described. Is this based on other centipede peptides and/or on the activity at sodium channels etc.?

7. Results: If BLAST did not identify any similar sequences, how were the other sequences (shown in Figure S2d) identified? This section is unclear and needs further clarification. The sequence of rpTx1 should be included in the alignment.

8. Results: Provide alphafold confidence values rather than just saying a high confidence structure was predicted or show a figure with the color code of alphafold confidence value in the Supporting information. There are typically regions of high and low confidence in alphafold predicted structures. This should be clarified.

9. Results: The authors state that the disulfide connectivity was "derived from the experimental data and predicted structure was...". Please describe what experimental data was obtained on the disulfide connectivity and whether this was done on the venom peptide or the recombinant peptide.

10. Results: For EC50 values, state what {plus minus} refers to when showing the results in the result section (SEM, SDEV, 95% confidence?).

11. Supporting Information: Show the HPLC trace of the final, purified product (Figure S5c presumably only shows the pre-purified toxin).

12. Supporting Information: Show MS spectra before deconvolution, including all m/z precursor ions. The deconvoluted spectra is somewhat meaningless as the authors just labeled the peak with the deconvoluted mass.

13. Supporting Information: Spelling error in Figure S8e ("Sequence")

Referee #3:

In this manuscript, the authors report the discovery of a unique peptide toxin, rpTx1, derived from centipede venom. They utilized a comprehensive, multi-method approach involving electrophysiology, fluorescence imaging, mutagenesis, surface plasmon resonance, mutant cycle analysis, and molecular dynamics (MD) simulations to reveal the molecular interactions between the toxin and Nav channels. Notably, the authors identified the intrinsic inactivation particle of Nav channels as a novel receptor site for this centipede-derived peptide toxin, thereby demonstrating its capacity to modulate Nav channel activity. The functional assays are extensive and adequately illustrate the binding and modulatory mechanism.

Nonetheless, several aspects require further clarification or adjustment:

1. The toxin rpTx1 contains two functional domains: an N terminal domain for cell penetration and a C terminal domain for modulating channel activity. The C terminal domain has three disulfide bonds. What's the function of these disulfide bonds, considering they would be broken when the toxin enters the cytosol? The authors also attempted to co-express the toxin with Nav channels and obtained similar electrophysiological results. Perhaps they could mutate the Cys residues of the toxin when co-expressing with Nav channels.
2. In Figure 2, control current traces are not shown alongside the low-concentration toxin traces. Please add control traces for comparison or provide an explanation for their absence.
3. In Figure 3, the IFMT-related mutants abolished the effect of rpTx1 on the Nav1.8 persistent current. Given that rpTx1 also induces a shift in the voltage-dependent inactivation of Nav1.8, do these IFMT-related mutants similarly affect the inactivation V_{half} shift caused by the toxin? Please provide further details on this point.
4. For the MD simulation section, additional data should be presented. Specifically, it would be beneficial to include the interaction frequency or distance change of T66-T1436 (1437) and T66-I1433 (1434) over the 200 ns simulation period. For the metadynamics simulations, it would be helpful to explain a bit how these five structures (B1, B2, T1, U1 and U2) were generated and if they are at their free energy minima?
5. Functional assays indicate specific interactions within T66-I1434 and T66-T1437. However, these interactions appear absent in the initially predicted structure. Could you please provide a reasonable explanation for this discrepancy? Additionally, the distances of the specific interactions T66-T1436 (1437) and T66-I1433 (1434) are 5.5 Å or 6.8 Å which may be too far to form the strong interaction between the toxin and the Nav1.8. Further elaboration on these points is recommended.
6. Please re-check the residue number of IFMT in Figure 4, Figure 5 and the manuscript. Please verify and correct as necessary.

ADDITIONAL SPECIFIC COMMENTS BY REFEREE #1

For me the most concerning result is the "inhibition" of extracellular peptide shown after incubation for 1 hour. This is shown in Fig 2c (and was also noted by Reviewer #2), where despite the high concentration and long incubation period, only 13.3 - 36.4% of cells showed significant inhibition of inactivation.

Combined with lack of convincing data showing penetration of the peptide intracellularly, the proposed mechanism of action is not strongly supported by the experimental data. This data is also at odds with the (single current trace) shown in Fig 1c that implies inhibition of inactivation after 30 min incubation.

At a minimum I would require additional experiments showing this effect; and more experimental detail in general.

ADDITIONAL SPECIFIC COMMENTS BY REFEREE #3

After thoroughly reviewing the comments provided by Referee #1 and Referee #2, I would like to offer my perspective and contribute to the discussion regarding the minimal additional experiments and analyses required to support the authors' claims within a feasible time frame of approximately three months.

Recommendations for Minimal Additional Experiments:

Based on the comments from all three reviewers, I believe the following minimal set of experiments and analyses would be necessary to support the authors' conclusions while being realistically achievable within three months:

1. **Venom Processing:** Investigate the possibility that rpTx1 may be a precursor peptide, including investigating whether a truncated version of rpTx1 exists in venom and an exploration of potential cleavage sites. This is critical, as Reviewer 2 noted that the N-terminal domain may be cleaved in the mature toxin.
2. **Electrophysiological Assays:** Additional electrophysiological measurements to include control current traces in Figure 2 and to explore the V_{half} shift in the IFMT-related mutants (point 3 of my review) are necessary. This will clarify the functional impact of the toxin on voltage-dependent inactivation and help address the reviewers' concerns about the thoroughness of the functional assays. Meanwhile, perform detailed electrophysiological studies, including: Time course of peptide activity following extracellular application; Concentration-response experiments with proper controls, ensuring that the differences between mutants (such as I1434A) and wild-type are statistically validated; Quantification of persistent currents to better characterize the peptide's effects on Nav channels; Performing kinetic studies such as "recovery from inactivation" to clarify the peptide's effects on sodium channels.
3. **Cell Penetration Quantification:** As raised by reviewer 1 and 2, provide robust, quantitative evidence of rpTx1 cell penetration, using alternative methods to FITC labeling. This could include quantitative flow cytometry or biochemical uptake assays. These

are essential, as cell penetration is a critical claim of the study.

4. MD Simulation Data: Expanding the molecular dynamics data to include the interaction frequencies or distance changes between specific residues (point 4 of my review) will provide deeper insight into the toxin-channel interactions, addressing concerns from all reviewers about the validity of these computational models. Additionally, clarifying the structural energy minima for the meta dynamics simulations (point 4 of my review) will be helpful in interpreting the simulation results in relation to the experimental data. More robust MD simulation data and additional electrophysiological assays could offer deeper insights into alternative toxin mechanisms or support further hypotheses regarding potential dual actions of the toxin.

5. Clarification of Discrepancies in Interaction Distances: Providing further elaboration on the discrepancies in interaction distances between T66-T1436 (1437) and T66-I1433 (1434) (point 5 of my review) will strengthen the manuscript's mechanistic claims. Although this may require some additional analysis or computational refinement, it is a necessary step to resolve concerns about the structural data.

Referee #1:

Although description of a novel mechanism of action of peptide toxins at sodium channels could potentially be interesting, there are some questions regarding the validity of the conclusions, as well as some minor comments generally.

The definition of the toxin binding sites is meanwhile outdated - there are several other publications that have laid claim to a site "10".

Was the peptide synthesised in one step?

How was the peptide discovered in the first place? If there is no activity at 5 minutes it seems a fortuitous coincidence that the peptide was found at all.

It is not clear at all why fusion to the C-terminus of eGFP should tether the peptide; nor is there any evidence that it is tethered. Similarly, the strategy used to fuse the peptide to b1 is not clear, nor is there any evidence of the peptide being "maintained" on the intracellular side of cells. The interactions of b1, b2 and b4 with the sodium channel are quite different, and it seems implausible that the peptide would be able to reach a common binding site without a cleverly designed linker. Please elaborate on the design strategy.

Please show a time course of peptide activity following extracellular application.

It is also not clear why treatment with 5 μ M recombinant peptide (for how long?) only caused a response in a small subset of cells (Fig 2c). This is in contrast to the near uniform permeability implied by the experiment with FITC-labelled peptide. This is also in contrast to the effect observed after 30 minutes with NaV1.7 (unless a responding cell was cherry-picked).

The labelling is also confusing - is rpTx1 the same as rpTx1-N? presumably it is the N-terminal half, what was the exact sequence of this and the C-terminal peptide?

While identifying residues that contribute to activity is interesting, a 6-22 fold reduction in potency could hardly be termed "critical".

Importantly, the view that the IFM motif binds to a region in the pore is a now outdated view, with the binding site having been identified to lie adjacent to the pore. This makes activity of both b1 and b2-fused peptide even more surprising as a longer linker would be expected to be required.

The concentration-response curves shown in Fig 3 C are concerning. For example, Fig. 3A shows that the I1434A mutant has decreased fast inactivation (consistent with this residue being part of the IFM cluster important for fast inactivation). Indeed, this mutant has a persistent current shown as ~ 0.5 of peak. (As a side note, what is missing is quantification of the effect of these mutations on persistent current.) Now, a very low concentration of peptide should accordingly be similar to control, i.e. $\sim 0.5 I_{45}/I_{peak}$ - but what is shown in the CRC is ~ 0.2 . There are two possibilities, either the representative current trace isn't very representative, or the inactivation particle of NaV1.8 actually does not contribute much to inactivation.

As an additional control - there are some very good antibodies against the inactivation particle. What is the effect of this antibody (pan-NaV, e.g. from Sigma Aldrich) on persistent current? And does the antibody still bind in presence of rpTx1?

The schematic in Fig 6a is very confusing; but if the hypothesis is that rpTx1 prevents transition from the open to inactivated state then some kinetic electrophysiological studies should be conducted that can address these questions - eg recovery from inactivation. The voltage protocol used in Fig 6 are not clear.

In terms of proposed mechanism - although it could make sense that the apparent potency of rpTx1 is increased at NaV1.8 because the inactivation particle binds less strongly to the receptor site. However, by that logic, the inactivation particle of NaV1.9 would bind even less strongly to the receptor site, and the peptide should be even more potent.

Was Fig 1e) obtained following extra- or intracellular peptide application?

Please provide more information on FITC labelling: what is the reaction, what was the efficiency, where was FITC attached?

There is no information on the hNaV1.9 clone. It is also notable that no other lab in the world has successfully managed to express hNaV1.9; including some of world's leaders on Nav channels. Can you elaborate on why this clone expresses, and whether or not anyone else is able to replicate your findings? The lack of reproducibility of this astounding result, particularly the current size, is somewhat concerning.

The manuscript will need to be proof-read for language carefully as there are several errors throughout.

Referee #2:

Voltage-gated sodium channels (Navs) are critical players in neurotransmission. Animal toxins have long been used to understand Nav structure and function. Fast inactivation is a hallmark of the kinetics of mammalian Navs and is critical for repetitive firing of action potentials. The short intracellular linker between DIII and DIV (DIII-DIV linker) is a key structural element for fast inactivation with a hydrophobic motif Ile-Phe-Met-Thr (the IFMT motif), located in the N-terminal of this linker, known as the "inactivation particle". This study identified a novel centipede toxin, rpTx1 from the Chinese red-headed centipede *Scolopendra subspinipes mutilans*, that appears to penetrate the cell membrane (via an N-terminal alpha helical domain) and specifically binds to the inactivation particle to inhibit the fast inactivation of Nav channels (via a C-terminal toxin domain). As such, this toxin represents a novel tool to study Nav channel inactivation and represents a beautiful example of the diverse mechanisms evolved by venomous animals to subdue their prey and defend against predators. This is a very exciting and relevant study that will be of broad interest to the EMBO J readership and the scientific community.

However, I have several major concerns that need to be addressed before accepting this work for publication. Briefly, the result section lacks details and some of the claims are not well supported by rigorous data. This includes the claim that rpTx1 crosses the cell membrane. The electrophysiological data strongly suggests that this is indeed the case but the authors fail to provide compelling data using an orthogonal approach. Since cell penetration is one of the key findings of this study, there has to be strong unambiguous data supporting this finding. The FITC-labeled HEK cell penetration data is weak and not convincing. Additional reasons for why there needs to be strong supporting data is that the discovered toxin could represent an unprocessed precursor of the actual mature toxin that only consists of the disulfide rich domain. There is an obvious dibasic cleavage site (RK) separating the two domains, suggesting that the toxin may be further processed in the venom. Furthermore, the toxin appears to only constitute a very small fraction of the HPLC trace and the authors do not discuss how the venom was extracted (was the

entire gland lysed?) and whether they also found or at least looked for a version of the toxin lacking the N-terminus. The electrophysiological data on the action of the toxin on the intracellular domain appears convincing but toxins can get inside cells by the action of other toxins present in the venom cocktail. It is also somewhat surprising that such a rather big toxin can diffuse into the cell via linkage to a helical peptide. Given all these observations, there needs to be strong data supporting the claim that trpTx1 can indeed efficiently penetrate the cell membrane. There are various methods to quantitatively measure cell penetration and the authors need to show at least one of these methods in a robust manner.

A more detailed explanations of this and my other concerns is provided below.

Major concerns essential to be addressed to support the conclusions:

The results lack a lot of the information needed to evaluate the claims made by the authors. While some of the information is provided in the method and SI sections it is difficult for the reader to find this information and it should therefore be included in the result section alongside the claims.

Specific comments:

1. The language should be changed in a couple of places to avoid misleading interpretations of the significance of the results. Currently, the following two sentences could suggest that the toxin identified a previously unknown site on the channel:

Lines 25-27: "This study revealed the inherent inactivation particle of Nav channels as a new neurotoxin binding site, utilizing a novel centipede peptide toxin called rpTx1."

Change to: Utilizing a novel centipede peptide toxin called rpTx1, this study revealed that the inherent inactivation particle of Nav channels represent a binding site for a neurotoxin.

Lines 395-397: "we identified the inherent inactivation particle of Nav channels as a new receptor site for a centipede peptide toxin that can modify Nav channel activity."

Change to: we identified that the centipede toxin modifies Nav channel activity via binding to the inactivation particle.

2. Similarly, the title needs to be changed to clarify that the study identified a single toxin with this mode of action. Suggesting the existence of other toxins is fine but does not warrant the use of plural in the title. Change "neurotoxins" to "a neurotoxin"

3. The result section lacks important details and, in some places, experiments lack rigor and results/conclusions are questionable.

For example, a major concern is the lack of information on how the new toxin relates to previously described, similar sequences. The authors state that BLAST did not identify similar sequences but some similar sequences are shown in Figure S2d but the source of these sequences is not provided. Were any of these characterized before? This needs to be better described for transparency.

Another example are the results for the FITC-labeled cell penetration assay. These are not well described and somewhat confusing. Since cell-penetration is an essential claim of the paper the FITC data needs to be more thoroughly analyzed or repeated with higher confidence. Only very few cells were analyzed and data is not quantitative and completely lacks any statistical analysis.

Why is there no quantitative comparison between the data for rpTx1, rpTx1-8K/A, and rpTx-N? If anything, currently it seems that rpTx1 had better cell penetration than rpTx-N and the image for rpTx-N looks like the experiment may not have worked.

Why were such few cells analyzed and how could such a small cell number possibly give statistically significant results (i.e., 8/22 cells for 5uM rpTx1 and 2/15 cells for 1uM)? Perhaps this is why there is no statistical analysis here.

4. The identification of potentially similar two-domain toxins as presented in the discussion and Figure 11 is intriguing but very immature and not supported by any data. This part should be removed from the paper given the complete lack of evidence of dual action in any of the peptides shown in Figure 11. For example, it has long been known that cone snail toxins are translated as propeptides that contain an N-terminal pro-region. Several roles have been attributed to this region (assistance in folding, binding of modifying enzymes, and binding of secretory trafficking molecules). Using alphafold these proregions that can contain positive charges are often predicted to be helical which could suggest that these may penetrate cells, although all known cone snail toxins have extracellular targets. This is not to say that the mechanism described for rpTx1 does not exist in cone snail toxins and other toxins but it is to say that seemingly having two domains like rpTx1 on the predicted propeptide region of a toxin is NOT suggestive of the same mode of action as seen for rpTx1. Again, in the absence of any evidence, figure 11 needs to be removed.

Instead, the authors could suggest that similar sequences to rpTx1 from related centipede species, as shown in figure S2d, may act like rpTx1 and that other toxins with similar mechanisms may also exist in other animal venoms.

Minor concerns that should be addressed:

1. General: Insert space before citations.
2. Introduction: When discussing the usefulness of toxins that target intracellular sites, it would be useful to list a few examples of other toxins that target other ion channels at intracellular sites. These are discussed in the discussion section but could already be mentioned in the introduction.
3. Results: It is difficult to see what HPLC fraction the asterisk in Figure 1a belongs to. Clarify this by using an arrow or something else that more clearly shows the active fraction.
4. Results: In the caption of Figure 1a include information on how much venom was loaded on this HPLC run and how much of this was estimated to be rpTx1. Provide information on whether the venom was "milked" or extracted from homogenized glands? Also provide information on whether the toxin was found in its "truncated version". There's a dibasic cleavage site between the two domains and it would not be surprising but highly relevant if the C terminus "toxin"-like part was

found without the N-terminal "carrier".

5. Results: 5'RACE requires information on related DNA sequences for at least one end. What sequences were used as the basis of primer design (degenerative primer)? This information should be provided in the results. What portion of the sequence was confidently identified by Edman sequencing? Presumably not all residues and not the entire toxin was sequenced by Edman. Similarly, RACE may not have provided the entire toxin region. This information should be provided.

6. Results: The rationale for naming the toxin is unclear ("We named it rpTx1 (rational nomenclature: δ -SLPTX10-Ssm1a(King, Gentz et al., 2008)).") and should be better described. Is this based on other centipede peptides and/or on the activity at sodium channels etc.?

7. Results: If BLAST did not identify any similar sequences, how were the other sequences (shown in Figure S2d) identified? This section is unclear and needs further clarification. The sequence of rpTx1 should be included in the alignment.

8. Results: Provide alphafold confidence values rather than just saying a high confidence structure was predicted or show a figure with the color code of alphafold confidence value in the Supporting information. There are typically regions of high and low confidence in alphafold predicted structures. This should be clarified.

9. Results: The authors state that the disulfide connectivity was "derived from the experimental data and predicted structure was...". Please describe what experimental data was obtained on the disulfide connectivity and whether this was done on the venom peptide or the recombinant peptide.

10. Results: For EC50 values, state what {plus minus} refers to when showing the results in the result section (SEM, SDEV, 95% confidence?).

11. Supporting Information: Show the HPLC trace of the final, purified product (Figure S5c presumably only shows the pre-purified toxin).

12. Supporting Information: Show MS spectra before deconvolution, including all m/z precursor ions. The deconvoluted spectra is somewhat meaningless as the authors just labeled the peak with the deconvoluted mass.

13. Supporting Information: Spelling error in Figure S8e ("Sequence")

Referee #3:

In this manuscript, the authors report the discovery of a unique peptide toxin, rpTx1, derived from centipede venom. They utilized a comprehensive, multi-method approach involving

electrophysiology, fluorescence imaging, mutagenesis, surface plasmon resonance, mutant cycle analysis, and molecular dynamics (MD) simulations to reveal the molecular interactions between the toxin and Nav channels. Notably, the authors identified the intrinsic inactivation particle of Nav channels as a novel receptor site for this centipede-derived peptide toxin, thereby demonstrating its capacity to modulate Nav channel activity. The functional assays are extensive and adequately illustrate the binding and modulatory mechanism.

Nonetheless, several aspects require further clarification or adjustment:

1. The toxin rpTx1 contains two functional domains: an N terminal domain for cell penetration and a C terminal domain for modulating channel activity. The C terminal domain has three disulfide bonds. What's the function of these disulfide bonds, considering they would be broken when the toxin enters the cytosol? The authors also attempted to co-express the toxin with Nav channels and obtained similar electrophysiological results. Perhaps they could mutate the Cys residues of the toxin when co-expressing with Nav channels.

2. In Figure 2, control current traces are not shown alongside the low-concentration toxin traces. Please add control traces for comparison or provide an explanation for their absence.

3. In Figure 3, the IFMT-related mutants abolished the effect of rpTx1 on the Nav1.8 persistent current. Given that rpTx1 also induces a shift in the voltage-dependent inactivation of Nav1.8, do these IFMT-related mutants similarly affect the inactivation V_{half} shift caused by the toxin? Please provide further details on this point.

4. For the MD simulation section, additional data should be presented. Specifically, it would be beneficial to include the interaction frequency or distance change of T66-T1436 (1437) and T66-I1433 (1434) over the 200 ns simulation period. For the metadynamics simulations, it would be helpful to explain a bit how these five structures (B1, B2, T1, U1 and U2) were generated and if they are at their free energy minima?

5. Functional assays indicate specific interactions within T66-I1434 and T66-T1437. However, these interactions appear absent in the initially predicted structure. Could you please provide a reasonable explanation for this discrepancy? Additionally, the distances of the specific interactions T66-T1436 (1437) and T66-I1433 (1434) are 5.5 Å or 6.8 Å which may be too far to form the strong interaction between the toxin and the Nav1.8. Further elaboration on these points is recommended.

6. Please re-check the residue number of IFMT in Figure 4, Figure 5 and the manuscript. Please verify and correct as necessary.

ADDITIONAL SPECIFIC COMMENTS BY REFEREE #1

For me the most concerning result is the "inhibition" of extracellular peptide shown after incubation for 1 hour. This is shown in Fig 2c (and was also noted by Reviewer #2), where despite the high concentration and long incubation period, only 13.3 - 36.4% of cells showed significant inhibition of inactivation.

Combined with lack of convincing data showing penetration of the peptide intracellularly, the proposed mechanism of action is not strongly supported by the experimental data. This data is also at odds with the (single current trace) shown in Fig 1c that implies inhibition of inactivation after 30 min incubation.

At a minimum I would require additional experiments showing this effect; and more experimental detail in general.

ADDITIONAL SPECIFIC COMMENTS BY REFEREE #3

After thoroughly reviewing the comments provided by Referee #1 and Referee #2, I would like to offer my perspective and contribute to the discussion regarding the minimal additional experiments and analyses required to support the authors' claims within a feasible time frame of approximately three months.

Recommendations for Minimal Additional Experiments:

Based on the comments from all three reviewers, I believe the following minimal set of experiments and analyses would be necessary to support the authors' conclusions while being realistically achievable within three months:

1. **Venom Processing:** Investigate the possibility that rpTx1 may be a precursor peptide, including investigating whether a truncated version of rpTx1 exists in venom and an exploration of potential cleavage sites. This is critical, as Reviewer 2 noted that the N-terminal domain may be cleaved in the mature toxin.
2. **Electrophysiological Assays:** Additional electrophysiological measurements to include control current traces in Figure 2 and to explore the V_{half} shift in the IFMT-related mutants (point 3 of my review) are necessary. This will clarify the functional impact of the toxin on voltage-dependent inactivation and help address the reviewers' concerns about the thoroughness of the functional assays. Meanwhile, perform detailed electrophysiological studies, including: Time course of peptide activity following extracellular application; Concentration-response experiments with proper controls, ensuring that the differences between mutants (such as I1434A) and wild-type are statistically validated; Quantification of persistent currents to better characterize the peptide's effects on Nav channels; Performing kinetic studies such as "recovery from inactivation" to clarify the peptide's effects on sodium channels.
3. **Cell Penetration Quantification:** As raised by reviewer 1 and 2, provide robust, quantitative evidence of rpTx1 cell penetration, using alternative methods to FITC labeling. This could include quantitative flow cytometry or biochemical uptake assays. These are essential, as cell penetration is a critical claim of the study.
4. **MD Simulation Data:** Expanding the molecular dynamics data to include the interaction frequencies or distance changes between specific residues (point 4 of my review) will provide deeper insight into the toxin-channel interactions, addressing concerns from all reviewers about the validity of these computational models. Additionally, clarifying the structural energy minima for the meta dynamics simulations (point 4 of my review) will be helpful in interpreting the simulation results in relation to the experimental data. More robust MD simulation data and additional electrophysiological assays could offer deeper insights into alternative toxin

mechanisms or support further hypotheses regarding potential dual actions of the toxin.

5. Clarification of Discrepancies in Interaction Distances: Providing further elaboration on the discrepancies in interaction distances between T66-T1436 (1437) and T66-I1433 (1434) (point 5 of my review) will strengthen the manuscript's mechanistic claims. Although this may require some additional analysis or computational refinement, it is a necessary step to resolve concerns about the structural data.

Inherent fast inactivation particle of Nav channels as a new binding site for a neurotoxin

Xi Zhou^{1,2,4,#,*}, Haiyi Chen^{3,5#}, Shuijiao Peng^{1,2,4}, Yuxin Si^{1,2,4}, Gaoang Wang³, Li Yang^{1,2,4}, Qing Zhou^{1,2,4}, Minjuan Lu^{1,2,4}, Qiangling Xie^{1,2,4}, Xi He^{1,2,4}, Meijing Wu^{1,2,4}, Xin Xiao^{1,2,4}, Xiaoqing Luo^{1,2,4}, Xujun Feng^{1,2,4}, Wenxing Wang^{1,2,4}, Sen Luo^{1,2,4}, Yaqi Li^{1,2,4}, Jiaxin Qin^{1,2,4}, Minzhi Chen^{1,2,4}, Qianqian Zhang^{1,2,4}, Weijun Hu^{1,2,4}, Songping Liang^{1,2,4}, Tingjun Hou^{3,*}, Zhonghua Liu^{1,2,4,*}

Our revisions are described below in a point-by-point manner. For easier reading, the reviewers' comments are marked in black, responses are marked in red, figure legend are marked in blue. Questions are copy-pasted from the EMBO J Editor's decision correspondence. Changes in the main text of the revised manuscript are highlighted in red.

Referee #1:

Although description of a novel mechanism of action of peptide toxins at sodium channels could potentially be interesting, there are some questions regarding the validity of the conclusions, as well as some minor comments generally.

Response: Thank you very much for recognizing the value of our work. We have carefully revised the manuscript according to your comments and those of the other reviewers. Point-by-point responses to the comments are listed below.

The definition of the toxin binding sites is meanwhile outdated - there are several other publications that have laid claim to a site "10".

Response: Thank you for pointing out this comment. We have changed "10" to "new" in the revised manuscript, that is, we defined the inactivation gate as "a new bind site" instead of "site 10".

Was the peptide synthesised in one step?

Response: Thanks. In this study, firstly, rpTx1 was isolated and purified from centipede venom, and after determining its amino acid sequence, the peptide rpTx1 and its mutants were obtained through prokaryotic expression in *E. coli* BL21 (DE3). Meanwhile, the IFMT peptides, the KIFMK peptide, and rpTx1-Ntermi were synthesized in a single step through solid-phase chemical synthesis.

How was the peptide discovered in the first place? If there is no activity at 5 minutes it seems a fortuitous coincidence that the peptide was found at all.

Response: Your assumption is reasonable, and there might indeed be an element of coincidence. However, in reality, our screening process was guided by a clear objective. We aimed to identify peptide toxins with novel mechanisms, and therefore, we paid special attention to certain unusual phenomena during the screening process. In this case, during our screening for modulators of Nav channels using patch clamp recording, we observed that upon adding a RP-HPLC fraction from the centipede venom, some cells exhibited an obvious decrease in seal resistance, we were interested with what would happen further if the recording continued. We found that the seal resistance recovered to its initial level within approximately 2 minutes, and after recording the current for more than 5 minutes, the recorded Nav channel currents exhibited delayed inactivation. Therefore, we thought this fraction should have the activity of slowing the fast inactivation of Nav channels. Moreover, this represented a relatively slow mode of action, which differed from other fast-acting toxins. We speculated that the toxin contained in this fraction might have a mechanism distinct from currently known Nav channel toxins, that is, the peptide toxin might (1) be entering the cells to affect the fast inactivation of Nav channels, rather than acting on the extracellular region; or (2) act on specific proteins which can modulate the activity of Nav channels.

It is not clear at all why fusion to the C-terminus of eGFP should tether the peptide; nor is there any evidence that it is tethered. Similarly, the strategy used to fuse the peptide to b1 is not clear, nor is there any evidence of the peptide being "maintained" on the intracellular side of cells. The interactions of b1, b2 and b4 with the sodium channel are quite different, and it seems implausible

that the peptide would be able to reach a common binding site without a cleverly designed linker. Please elaborate on the design strategy.

Response: Thank you for your valuable comments. The aim of constructing the eGFP-rpTx1 and beta1/2/4-rpTx1 fusion proteins is to demonstrate the intracellular functionality of rpTx1. Both eGFP-rpTx1 and beta1/2/4-rpTx1 are constructed using an identical method, which involves attaching rpTx1 to the C-terminus of either eGFP or beta1/2/4 via a linker (RILENLYFQG). To confirm the expression and intracellular localization of the fusion proteins, we performed Western blotting (WB) and confocal microscopy experiments. As shown in Figure R1, eGFP-rpTx1 is strongly expressed in ND7/23 cells. The molecular weight of eGFP-rpTx1 is significantly higher than that of eGFP by approximately 10 kDa, which is consistent with the theoretical molecular weight of eGFP-rpTx1; Moreover, in the cell lysate expressing the rpTx1-eGFP fusion protein, only the band of the fusion protein was detected, while the eGFP band itself was absent. This suggests that rpTx1 in eGFP-rpTx1 forms a complete fusion protein with eGFP. The confocal microscopy results further indicate that eGFP-rpTx1 is primarily localized within the cytoplasm on the inner side of the plasma membrane. Given that eGFP-rpTx1 and beta1/2/4-rpTx1 share the same linkage strategy, we hypothesize that beta1/2/4-rpTx1 may exhibit similar results. We have added these results in the revised manuscript (**Revised Supplementary Fig. 6**).

Actually, the interactions of beta1, 2 and b4 with Nav channel are quite different. Therefore, rpTx1 fused to the C-terminus of these beta subunits may be difficult to bind to the common site on Nav channels to exert its effect. However, we propose the following possibility: the recombinant proteins of beta1, beta2, or beta4 fused with rpTx1 are transiently expressed in cells, leading to overexpression. Some of the fusion proteins localize to the cell membrane and interact with Nav channels, while others exist in the cytoplasm in a free state. They can bind to an intracellular region of Nav channels in an rpTx1-dependent manner, with minimal involvement of the beta subunits. However, this requires a prerequisite: the linker sequence between the beta subunits and rpTx1 must have sufficient flexibility so as not to restrict rpTx1 from binding to the intracellular region of Nav channels. Data from eGFP-rpTx1 also support this hypothesis.

Revised Supplementary Figure 6. The eGFP-rpTx1 fusion protein maintained rpTx1 on the intracellular side of cells.

(a) Representative western blotting images of recombinant eGFP in cell lysates from HEK293T cells (line 1), HEK293T cells expressing eGFP (line 2), and HEK293T cells expressing eGFP-rpTx1 (line 3), respectively (n=3). (b) Confocal images show that eGFP-rpTx1 is primarily distributed within the cytoplasm. DiD and DAPI were used as membranous and nuclei markers, respectively. Scale bar: 10 μm .

Please show a time course of peptide activity following extracellular application.

Response: Thank you for pointing out this comment. We have added this result in Revised Fig. 1c of the reviewed manuscript.

Here, we also attached this figure (Revised Fig. 1c) for your reviewing.

Revised Figure 1c. The time course of rpTx1 (10 μM) activity on Nav1.7 following extracellular application, different colors indicate recordings from different cells (n=11). The inset shows representative current traces exhibiting an rpTx1-induced response following 30 minutes of extracellular treatment with 10 μM rpTx1. These currents were elicited by a 50-ms depolarization to 0 mV from the holding potential of -90 mV, applied at a frequency of 0.1 Hz.

It is also not clear why treatment with 5 μM recombinant peptide (for how long?) only caused a response in a small subset of cells (Fig 2c). This is in contrast to the near uniform permeability implied by the experiment with FITC-labelled peptide. This is also in contrast to the effect observed after 30 minutes with Nav1.7 (unless a responding cell was cherry-picked).

Response: Thank you for your valuable comments. In the revised manuscript, we have performed experiments and increased the number of cells tested (exceeding 59 cells per group). ND7/23 cells expressing rNav1.8 were incubated with the recombinant rpTx1 at different concentrations for 30 minutes, then these cells were randomly selected for patch-clamp recording, showing that only a subset of cells (approximately 30.6% for 5 μM rpTx1 and 18.6% for 1 μM rpTx1) exhibited a significant inhibition of the fast inactivation of rNav1.8 in response to rpTx1 (Revised Fig. 2c). The results are similar to data on rNav1.8 obtained previously, and to the data on hNav1.7 which show that some HEK293T cells expressing hNav1.7 (4 out of 11 cells) had delayed fast inactivation of Na^+ currents with the extracellular treatment of 10 μM rpTx1. This is consistent with our findings from the FITC-labelled peptide cell permeability experiments, which were statistically analyzed and revealed that approximately 29.8% of the cells demonstrated notable FITC fluorescence after treatment with FITC-labelled rpTx1 (5 μM) for 30 minutes (Revised Fig. 2a). Therefore, we have rewritten these results in the revised manuscript.

Here, we also attached this figure (**Revised Fig. 2**) for your reviewing.

Revised Figure 2. Rptx1 contains two functional domains. (a) *Left panel*, confocal images of 30 min cellular uptake of 5 μ M FITC-labelled rptx1 or rptx1-8K/A or rptx1-Ntermini on HEK293T cells. Cell membranes and nuclei were labelled by DiI or DAPI, respectively. Scale bar: 20 μ m (n=3 independent experiments). *Right panel*, quantitative fluorescence analysis was performed based on the data from (a) (one-way ANOVA with post hoc analysis using the Dunnett's multiple comparisons test, n = 8-10). (b) Representative current traces and scatter dot plot showing the effect of rptx1-8K/A on rNav1.8 upon intracellular (*upper*) (n=6, nonparametric Mann-Whitney two-tailed test) or extracellular application (*lower*) (n=5). (c) Comparison of the persistent currents of rNav1.8-expressing ND7/23 cells incubated with or without rptx1 or rptx1-8K/A. The cells were pre-treated for 30 min at 25 $^{\circ}$ C (n=21 for vehicle, n=15 for 1 μ M rptx1, n=22 for 5 μ M rptx1, and n=14 for 5 μ M rptx-8K/A). (d) Representative current traces and scatter dot plot showing the effect of rptx1-Ntermini on rNav1.8 upon intracellular (*left*) (n=5) or extracellular application (*right*) (n=5). (e) Representative current traces and scatter dot plot showing the effect of rptx1-Ctermini on rNav1.8 upon intracellular (*upper*) (n=6, nonparametric Mann-Whitney two-tailed test) or extracellular application (*lower*) (n=5). (f) Inhibition of the fast inactivation of rNav1.8 by various concentrations of WT-rptx1, M45A, E60A and T66A mutants, respectively (n=3-17 per concentration). (g) The locations of the three key residues M45, E60 and T66 are shown on the rptx1 structure. Data are presented as mean \pm S.E.M. **p < 0.01, ***p < 0.001.

The labelling is also confusing - is rptx1 the same as rptx1-N? presumably it is the N-terminal half, what was the exact sequence of this and the C-terminal peptide?

Response: Thanks! They are different. Rptx1-N refers to the N-terminal portion of rptx1, which we now name it as rptx1-Ntermini, comprising amino acid residues 1-42 of rptx1 full sequence (rptx1-Ntermini sequence is LKVEDLEPPTSYMKELKLALDKDKKDPKLOAYIKKIFPENRK), while rptx1-C refers to the C-terminal portion of rptx1, which we now name it as rptx1-Ctermini, comprising amino acid residues 43-83 (rptx1-Ctermini sequence is SCMTNCKHVEGCFLLSPECCPKMTSTCLELDIVKEHMKKTK). We have mentioned this in the **Revised Fig. 1b** and the results section.

While identifying residues that contribute to activity is interesting, a 6-22 fold reduction in potency

could hardly be termed "critical".

Response: Thank you for your valuable comments. We have corrected them in the revised manuscript.

Importantly, the view that the IFM motif binds to a region in the pore is a now outdated view, with the binding site having been identified to lie adjacent to the pore. This makes activity of both b1 and b2-fused peptide even more surprising as a longer linker would be expected to be required.

Response: We thank you for this valuable comment. Indeed, recent cryo-EM studies have demonstrated that the receptor site targeting by the IFMT motif is formed by residues positioned in the S4-S5 linkers of DIII and DIV, as well as the intracellular termini of the S5 and S6 segments in DIV. The targeting by the motif leads to the closure of the intracellular gate through allosteric effects, rather than through direct blocking. We added this in the Discussion section of revised manuscript.

The reviewer expresses concern that beta1/2 may be spatially distant from the IFMT region, potentially preventing rpTx1 from accessing and functioning within the IFMT region. We gave our assumption described above. At the same time, there is also another possibility. The intracellular segment of beta1/2 comprises approximately 35 amino acid residues. When combined with the linker and the N-terminus of rpTx1, there are nearly 80 amino acid residues in total. Furthermore, based on the currently resolved complex structure of the beta1-Nav channel, the C-terminal of beta1's intracellular segment is localized near the DIII, suggesting that rpTx1 might spatially be able to access the IFMT region.

The concentration-response curves shown in Fig 3 C are concerning. For example, Fig. 3A shows that the I1434A mutant has decreased fast inactivation (consistent with this residue being part of the IFM cluster important for fast inactivation). Indeed, this mutant has a persistent current shown as ~0.5 of peak. (As a side note, what is missing is quantification of the effect of these mutations on persistent current.) Now, a very low concentration of peptide should accordingly be similar to control, i.e. $\sim 0.5 I_{45\text{ms}}/I_{\text{peak}}$ - but what is shown in the CRC is ~ 0.2 . There are two possibilities, either the representative current trace isn't very representative, or the inactivation particle of Nav1.8 actually does not contribute much to inactivation.

Response: Sorry for this misunderstanding. The representative current trace was not very representative, so we have corrected it in the revised manuscript (**Revised Fig. 3a**). In the revised manuscript, we have added quantification of the effect of these mutations on the persistent currents and observed that the mutant I1434A of rNav1.8 significantly increased the persistent currents, with $I_{45\text{ms}}/I_{\text{peak}}$ values ranging from 0.17 to 0.396 (mean \pm sem: 0.25 ± 0.03), while the $I_{45\text{ms}}/I_{\text{peak}}$ values of the wild-type (WT) currents were between 0.07 and 0.149 (mean \pm sem: 0.1 ± 0.01) (**Revised Supplementary Fig. 9b**). These findings suggest that I1434 is crucial for the fast inactivation of rNav1.8. Furthermore, our results also indicate that compared to the WT, the I1434A mutant reduced the effect of rpTx1 on fast inactivation. Upon the addition of 0.1 μM rpTx1 intracellularly, the $I_{45\text{ms}}/I_{\text{peak}}$ values of the I1434 mutant and WT currents were increased to 0.62 ± 0.06 and 0.85 ± 0.03 , respectively ($p=0.007$). In addition, while the M1436A and T1437A mutations themselves did not significantly affect the fast inactivation of the channel, these two mutations significantly reduced the impact of rpTx1 on the channel. Generally speaking, mutations in the inactivation particle can affect the fast inactivation, but this depends on the specific mutations.

For instance, mutating IFM to QQQ within the IFMT region completely abolishes the fast inactivation, and similarly, mutating F to Q will also have such an effect, but mutating F to W does not (Jiang, Banh et al., 2021, Kellenberger, West et al., 1997, McPhee, Ragsdale et al., 1995). In our study, we also found mutating F to T completely removes fast inactivation (**Fig. R1**).

Revised Supplementary Fig. 9b and 9c. (b) The scatter dot plot shows the effect of rNav1.8 mutants I1434A (n=7), M1436A (n=8), and T1437A (n=7) on the fast inactivation of the channel (Non-parametric test using Dunn's multiple comparisons test). Data are presented as mean \pm S.E.M. ** $p < 0.01$.

Figure for reviewers removed

Jiang D, Banh R, Gamal El-Din TM, Tonggu L, Lenaeus MJ, Pomès R, Zheng N, Catterall WA (2021) Open-state structure and pore gating mechanism of the cardiac sodium channel. *Cell* 184: 5151-5162.e11
Kellenberger S, West JW, Scheuer T, Catterall WA (1997) Molecular analysis of the putative inactivation particle in the inactivation gate of brain type IIA Na⁺ channels. *J Gen Physiol* 109: 589-605
McPhee JC, Ragsdale DS, Scheuer T, Catterall WA (1995) A critical role for transmembrane segment IVS6 of the sodium channel α subunit in fast inactivation. *J Biol Chem* 270: 12025-34

As an additional control - there are some very good antibodies against the inactivation particle. What is the effect of this antibody (pan-NaV, e.g. from Sigma Aldrich) on persistent current? And does the antibody still bind in presence of rpTx1?

Response: We thank you for this valuable comment. Based on your suggestion, since the delivery of antibodies from Sigma Aldrich takes four months, we have chosen to use the Pan-Nav antibody from Absolute Antibody (Ab02113-10.0), which is specific for a conserved sequence (TEEQKKYYNAMKGLGSKK) of the intracellular III-IV loop of vertebrate Nav channels. Consequently,

we supplemented our investigation to examine the effect of the Pan-Nav antibody on the fast inactivation current of Nav channels. We found that the addition of 100 $\mu\text{g/ml}$ Pan-Nav antibody intracellularly had no impact on Nav1.5 and Nav1.8 currents (**Fig. R2a and R2b**). When co-administered with rpTx1 in the intracellular solution, it also did not affect rpTx1's inhibition of Nav1.8's fast inactivation (**Fig. R2b**). To address this issue, we initially suspected that the purchased antibody might not be functional. Therefore, we verified it through WB. The results showed that this antibody specifically labels Nav1.5 channels expressed in HEK293T cells, both in the absence and presence of rpTx1 (**Fig. R2c**). Since this pan-Nav channel antibody is specific for a conserved sequence present in all vertebrate Nav channel subunits. It binds to all Nav channel subunits. We conclude that the antibody functions normally in WB. There are two possible reasons why Pan-Nav antibody does not affect the fast inactivation of Nav channels intracellularly: (1) The epitope may adopt a specific structure in the native (undenatured) state, which prevents recognition or binding by the antibody, while the epitope sequence is denatured in WB assay; (2) Although the epitope sequence is located near the inactivation particle sequence, the binding of the antibody may not influence the function of the inactivation particle. Furthermore, through cellular immunofluorescence detection, we found that the Pan-Nav antibody was unable to stain native hNav1.5 (**Fig. R2d**). Therefore, based on our results, we believe the former reason is more likely.

Using antibody experiments to demonstrate that rpTx1 binds to the IFMT and thereby blocks fast inactivation is an excellent idea. Therefore, we attempted this experiment using the available antibodies, but it did not achieve the desired outcome. However, this does not compromise the main results and conclusions of this study, as we already have multiple lines of evidence supporting them.

Figure for reviewers removed

The schematic in Fig 6a is very confusing; but if the hypothesis is that rpTx1 prevents transition from the open to inactivated state then some kinetic electrophysiological studies should be conducted that can address these questions - eg recovery from inactivation. The voltage protocol used in Fig 6 are not clear.

Response: Thank you for point out this. In the revised manuscript, in order to make our hypothesis clear, we reformatted the schematic as shown in **Revised Fig. 6a**. In the **Revised Fig. 6a**, we attempt to emphasize that rpTx1 treatment disrupts the normal transitions of Nav channels among the resting, activated, and inactivated states, hindering the binding of the IFMT motif to its receptor site, thereby leading to a persistent open state. We think our data can support this hypothesis, while more data from some kinetic electrophysiological studies will be more helpful, as you suggested. If rpTx1 can keep the IFMT motif in the unbound state, the recovery of the Nav currents from the fast inactivation would be faster in the presence of rpTx1. This was proved by our new experiments, as shown in the following figure. To determine the recovery from fast inactivation, a 50-ms prepulse at 20 mV was used to move channels into the fast inactivated state, followed by a pulse at -90 mV with increased duration to allow channels to recover from fast inactivation before a 50-ms test pulse at 20 mV to measure the available currents. For the rNav1.8 currents in the absence of rpTx1, the recovery from fast inactivation was gradually achieved with recovery duration increased. However, for the rNav1.8 currents in the presence of $0.2 \mu\text{M}$ rpTx1, recovery occurred instantaneously, and it appeared that no inactivation of the currents took place. RpTx1 treatment might cause rNav1.8 to remain in a non-inactivated state, essentially a persistent open state. These data were added in the **revised Supplementary Fig. 4b**. These results suggest that rpTx1 may prevent transition of the inactivation gate from the open to inactivated state.

Revised Supplementary Figure 4b. (left) Representative inactivation recovery current traces from rNav1.8 channels in the absence or presence of $0.2 \mu\text{M}$ rpTx1, (right) time course of recovery from fast inactivation of rNav1.8 in the absence ($n=9$) or presence ($n=5$) of $0.2 \mu\text{M}$ rpTx1. Data are presented as mean \pm S.E.M.

In terms of proposed mechanism - although it could make sense that the apparent potency of rpTx1 is increased at NaV1.8 because the inactivation particle binds less strongly to the receptor site. However, by that logic, the inactivation particle of NaV1.9 would bind even less strongly to the receptor site, and the peptide should be even more potent.

Response: We appreciate the question you raised. In fact, we are also aware of this point. The

explanation we proposed applies to Nav1.8 but may not be applicable to Nav1.9. This is because the mechanism of Nav channel fast inactivation is highly complex. As we mentioned in the discussion, it is influenced by many factors. For Nav1.8, the low affinity between the IFMT motif and its receptor site may be an important factor contributing to its slow inactivation. However, for Nav1.9, other factors may play a more significant role. Furthermore, the fast inactivation mechanism of Nav channels is likely more complex than currently understood.

Additionally, our latest molecular dynamics simulations have revealed that, apart from the IFMT motif, more amino acid residues near this motif also participate in binding with rpTx1. This may also influence the varying activity of rpTx1 on different Nav channel subtypes. Therefore, we must acknowledge that even for Nav1.8, the mechanism we proposed is a simplified model. Of course, research is always conducted in stages, and we cannot expect to understand everything in a single study. The deeper mechanisms are worth further investigation in the future.

This also highlights the value of rpTx1. A detailed exploration of the molecular mechanism of rpTx1's action will contribute to a better understanding of the complex problem of Nav channel fast inactivation. RpTx1 will serve as a useful molecular tool for studying the molecular mechanisms of Nav channel fast inactivation.

Was Fig 1e) obtained following extra- or intracellular peptide application?

Response: Thanks! It is intracellular peptide application, and we have mentioned it in the revised manuscript.

Please provide more information on FITC labelling: what is the reaction, what was the efficiency, where was FITC attached?

Response: We thank you for pointing this out. The experiment of FITC labeling was performed according to manufacturer's instructions (FluoroTag FITC Conjugation Kit, FITC1, Sigma). FITC conjugation occurs through the free amino groups of peptides, forming a stable thiourea bond. Therefore, FITC may be labeled on the N-terminus of the peptides and the side chain of lysine (Lys, K) amino acid residues. These details were mentioned in the materials and methods section of the revised manuscript.

There is no information on the hNav1.9 clone. It is also notable that no other lab in the world has successfully managed to express hNav1.9; including some of world's leaders on Nav channels. Can you elaborate on why this clone expresses, and whether or not anyone else is able to replicate your findings? The lack of reproducibility of this astounding result, particularly the current size, is somewhat concerning.

Response: We thank you for this valuable comment.

As you mentioned, heterologous expression of Nav1.9 is extremely challenging, which significantly limits research on Nav1.9. Therefore, considerable efforts have been devoted to establishing convenient and reproducible methods. Evidence suggests that low-temperature culture of transfected cells and modification of the Nav1.9 C-terminal facilitate functional expression of Nav1.9 (Vanoye et al., 2013; Goral et al., 2015), inspiring us to create a fusion protein channel. We successfully established a heterologous expression system for Nav1.9 by adding an eGFP protein to the C-terminal of Nav1.9 via a linker (ARDPPAA) (Front Pharmacol. 2017,8:852. doi: 10.3389/fphar.2017.00852). It is noted that the Waxman laboratory, experts in the field of sodium

channels, discovered a 49-residue sequence motif in the C-terminus of Nav1.9 that regulates trafficking of the channel to the plasma membrane. Deleting these 49 amino acids significantly increased channel current density in HEK293T cells (Sizova et al. 2020, doi: 10.1074/jbc.RA119.011424). Therefore, we believe that the C-terminal of Nav1.9 is crucial for its functional expression, and our established Nav1.9-GFP may promote plasma membrane expression by the fused GFP affecting the function of the C-terminal. Furthermore, a group from Pfizer Inc. established a Nav1.9 stably expressed HEK 293 system by coexpression of Nav1.9 with $\beta 1/\beta 2$ subunits (Lin et al., 2016), which was used for Nav1.9 modulator screening. Compared with other expression systems, our Nav1.9-GFP heterologous expression system offers the following advantages: (1) It expresses large and stable currents (>1 nA) in ND7/23 cells, which is essential for experimental data collection. (2) It is capable of functional expression in various cell types, including ND7/23, HEK293T, and CHO cells. Using this Nav1.9-GFP system, we identified two toxins targeting Nav1.9 from spider venom for the first time, revealing the crucial role of Nav1.9 in pain signaling pathways, gating mechanism and its structure-function relationship (*Nat Commun.* 2020,11(1):2293, doi: 10.1038/s41467-020-16210-y; *J Biol Chem.* 2024:108060, doi: 10.1016/j.jbc.2024.108060. *Front Pharmacol.* 2021,12:778534, doi: 10.3389/fphar.2021.778534). Therefore, we believe our results are reproducible. Information regarding the Nav1.9-GFP construct has been mentioned in the "Methods and Materials" section.

The manuscript will need to be proof-read for language carefully as there are several errors throughout.

Response: Thanks! The revised manuscript has been proofread by a native English speaker.

Referee #2:

Voltage-gated sodium channels (Navs) are critical players in neurotransmission. Animal toxins have long been used to understand Nav structure and function. Fast inactivation is a hallmark of the kinetics of mammalian Navs and is critical for repetitive firing of action potentials. The short intracellular linker between DIII and DIV (DIII-DIV linker) is a key structural element for fast inactivation with a hydrophobic motif Ile-Phe-Met-Thr (the IFMT motif), located in the N-terminal of this linker, known as the "inactivation particle". This study identified a novel centipede toxin, rpTx1 from the Chinese red-headed centipede *Scolopendra subspinipes mutilans*, that appears to penetrate the cell membrane (via an N-terminal alpha helical domain) and specifically binds to the inactivation particle to inhibit the fast inactivation of Nav channels (via a C-terminal toxin domain). As such, this toxin represents a novel tool to study Nav channel inactivation and represents a beautiful example of the diverse mechanisms evolved by venomous animals to subdue their prey and defend against predators. This is a very exciting and relevant study that will be of broad interest to the EMBO J readership and the scientific community. However, I have several major concerns that need to be addressed before accepting this work for publication. Briefly, the result section lacks details and some of the claims are not well supported by rigorous data. This includes the claim that rpTx1 crosses the cell membrane. The electrophysiological data strongly suggests that this is indeed the case but the authors fail to

provide compelling data using an orthogonal approach. Since cell penetration is one of the key findings of this study, there has to be strong unambiguous data supporting this finding. The FITC-labeled HEK cell penetration data is weak and not convincing. Additional reasons for why there needs to be strong supporting data is that the discovered toxin could represent an unprocessed precursor of the actual mature toxin that only consists of the disulfide rich domain. There is an obvious dibasic cleavage site (RK) separating the two domains, suggesting that the toxin may be further processed in the venom. Furthermore, the toxin appears to only constitute a very small fraction of the HPLC trace and the authors do not discuss how the venom was extracted (was the entire gland lysed?) and whether they also found or at least looked for a version of the toxin lacking the N-terminus. The electrophysiological data on the action of the toxin on the intracellular domain appears convincing but toxins can get inside cells by the action of other toxins present in the venom cocktail. It is also somewhat surprising that such a rather big toxin can diffuse into the cell via linkage to a helical peptide. Given all these observations, there needs to be strong data supporting the claim that trpTx1 can indeed efficiently penetrate the cell membrane. There are various methods to quantitatively measure cell penetration and the authors need to show at least one of these methods in a robust manner.

A more detailed explanations of this and my other concerns is provided below.

Major concerns essential to be addressed to support the conclusions:

The results lack a lot of the information needed to evaluate the claims made by the authors. While some of the information is provided in the method and SI sections it is difficult for the reader to find this information and it should therefore be included in the result section alongside the claims.

Response: Thank you very much for recognizing the value of our work. Due to the large amount of data obtained in this study and considering the limitations of the article's length, we have included some of the data in the supplementary materials. We agree with your viewpoint that this approach may indeed cause inconvenience to readers. Therefore, in the revised manuscript, we have provided as much detailed description as possible of the data included in the main text while ensuring that the data in the supplementary materials is clearly described.

Specific comments:

1. The language should be changed in a couple of places to avoid misleading interpretations of the significance of the results. Currently, the following two sentences could suggest that the toxin identified a previously unknown site on the channel:

√Lines 25-27: "This study revealed the inherent inactivation particle of Nav channels as a new neurotoxin binding site, utilizing a novel centipede peptide toxin called rpTx1."

Change to: Utilizing a novel centipede peptide toxin called rpTx1, this study revealed that the inherent inactivation particle of Nav channels represent a binding site for a neurotoxin.

√Lines 395-397: "we identified the inherent inactivation particle of Nav channels as a new receptor site for a centipede peptide toxin that can modify Nav channel activity."

Change to: we identified that the centipede toxin modifies Nav channel activity via binding to the inactivation particle.

Response: Thanks! We have corrected these in the revised manuscript.

v2. Similarly, the title needs to be changed to clarify that the study identified a single toxin with this mode of action. Suggesting the existence of other toxins is fine but does not warrant the use of plural in the title. Change "neurotoxins" to "a neurotoxin"

Response: Thanks! We have corrected it.

3. The result section lacks important details and, in some places, experiments lack rigor and results/conclusions are questionable.

For example, a major concern is the lack of information on how the new toxin relates to previously described, similar sequences. The authors state that BLAST did not identify similar sequences but some similar sequences are shown in Figure S2d but the source of these sequences is not provided. Were any of these characterized before? This needs to be better described for transparency.

Response: Sorry for this negligence. In the revised manuscript, we have clarified the issue as follows: "According to a BLAST search, rpTx1 exhibits 35-73% sequence identities with several centipede peptide toxins from centipedes. Among these toxins, Ssd1a and Ssd1b are isolated from the venom of *Scolopendra subspinipes dehaani* (Liu, Zhang et al., 2012), and the other are derived from the transcriptome analysis of venom glands of *Scolopendra morsitans* (Sm1a, Sm2a and Sm3a), *Cormocephalus westwoodi* (Cw1a and Cw2a), *Scolopendra alternans* (Sa2a), and *Ethmostigmus rubripes* (Er1a) (Undheim, Jones et al., 2014). Their functions remain unknown so far. The sequence comparison shows they share the same sequence architecture, composing of the long flexible N-terminal sequences and the C-terminal cysteine-rich sequences, and the arrangement patterns of cysteine residues are identical (**Supplementary Fig. 2d**)."

Another example are the results for the FITC-labeled cell penetration assay. These are not well described and somewhat confusing. Since cell-penetration is an essential claim of the paper the FITC data needs to be more thoroughly analyzed or repeated with higher confidence. Only very few cells were analyzed and data is not quantitative and completely lacks any statistical analysis.

Response: We thank you for this valuable comment. We also believe that the entry of rpTx1 into the cell membrane is crucial in this study. Using fluorescence imaging and patch-clamp techniques, we investigated this again by comparing the cellular entry and ability to inhibit Nav fast inactivation of rpTx1, rpTx-8K/A, rpTx-Ntermi, and rpTx-Ctermi. Please note rpTx-8K/A is the mutant of rpTx1 with 8 lysine residues in the N-terminal replaced by alanine residues, while rpTx-Ntermi and rpTx-Ctermi are peptide fragments of rpTx1 corresponding to the N- and C-terminal of the peptide, respectively. On one hand, we confirmed that rpTx1 can cross cell membranes and localize in cytoplasm by using rpTx1 labeled with FITC fluorescence, and quantitative analysis indicates that after treatment with FITC-labelled rpTx1 (5 μ M) for 30 min, approximately 29.8% of the cells exhibited obvious FITC fluorescence. Similarly, after treatment with FITC-labelled rpTx1-Ntermi (5 μ M) for 30 min, about 27.2% of the cells also showed notable FITC fluorescence. There is no significant difference between them. However, the mutant rpTx1-8K/A treatment could lead to only $8.3 \pm 1.8\%$ of the cells labeled with FITC fluorescence, that is, rpTx1-8K/A hardly has the ability to enter cells. On the other hand, patch clamp experiments also reveal that entry of cells by rpTx1 is required for its activity on Nav channels. For example, because both rpTx1-8K/A and rpTx1-Ctermi loss the cell entry ability, accordingly, they cannot affect Nav currents from the extracellular

side, although their intracellular application actually take effect. The patch-clamp experiments are described in detail in the main text and will not be repeated here.

Why is there no quantitative comparison between the data for rpTx1, rpTx1-8K/A, and rpTx-N? If anything, currently it seems that rpTx1 had better cell penetration than rpTx-N and the image for rpTx-N looks like the experiment may not have worked.

Response: We thank you for this valuable comment. Please see the above point response.

Why were such few cells analyzed and how could such a small cell number possibly give statistically significant results (i.e., 8/22 cells for 5uM rpTx1 and 2/15 cells for 1uM)? Perhaps this is why there is no statistical analysis here.

Response: We sincerely appreciate your comment. As shown in **Revised Fig. 2c**, we conducted this experiment again and counted more cells. As a result, we recorded data from more than 59 cells in each group and conducted statistical analysis. We observed a significant increase in the total number of cells responding to 1 μ M rpTx1, 5 μ M rpTx1, or 5 μ M rpTx1-8K/A, compared to the vehicle control, when ND7/23 cells expressing rNav1.8 were incubated with either rpTx1 or rpTx1-8K/A for 30 min. However, when compared to 5 μ M rpTx1, there was a notable reduction in both the percentage of cells exhibiting responses and the intensity of inhibition of fast inactivation for 5 μ M rpTx1-8K/A.

Revised Figure 2c. Comparison of the persistent currents of rNav1.8-expressing ND7/23 cells incubated with or without rpTx1 or rpTx1-8K/A. The cells were pre-treated for 30 min at 25 $^{\circ}$ C (n=63 for vehicle, n=59 for 1 μ M rpTx1, n=62 for 5 μ M rpTx1, and n=60 for 5 μ M rpTx-8K/A) (one-way ANOVA with post hoc analysis using the Dunnett's multiple comparisons test). Data are presented as mean \pm S.E.M. ** p < 0.01, *** p < 0.001, **** p < 0.0001.

4. The identification of potentially similar two-domain toxins as presented in the discussion and Figure 11 is intriguing but very immature and not supported by any data. This part should be removed from the paper given the complete lack of evidence of dual action in any of the peptides shown in Figure 11. For example, it has long been known that cone snail toxins are translated as propeptides that contain an N-terminal pro-region. Several roles have been attributed to this region (assistance in folding, binding of modifying enzymes, and binding of secretory trafficking molecules). Using alphafold these proregions that can contain positive charges are often predicted

to be helical which could suggest that these may penetrate cells, although all known cone snail toxins have extracellular targets. This is not to say that the mechanism described for rpTx1 does not exist in cone snail toxins and other toxins but it is to say that seemingly having two domains like rpTx1 on the predicted propeptide region of a toxin is NOT suggestive of the same mode of action as seen for rpTx1. Again, in the absence of any evidence, figure 11 needs to be removed. Instead, the authors could suggest that similar sequences to rpTx1 from related centipede species, as shown in figure S2d, may act like rpTx1 and that other toxins with similar mechanisms may also exist in other animal venoms.

Response: We thank you for this valuable comment. In our previous manuscript, our intention was to identify a series of toxin sequences with structural characteristics similar to rpTx1 in the venoms of other toxic animals. This was meant to emphasize that such a similar mechanism of action is widely present in toxins and represents a predation and defense mechanism broadly adopted by venomous animals. Of course, this is a hypothesis lacking solid evidence. We agree that including this figure in the manuscript is inappropriate in the absence of robust evidence, so we have removed Figure 11 and related context in the revised manuscript, and added some discussions as follows: “ Based on the BLAST result (**Fig. S2d**), peptide toxins with characteristics similar to rpTx1 are distributed across various centipede venoms, suggesting that they may have similar action mechanism, and other toxins with the similar mechanism may also exist in other animal venoms.”

Minor concerns that should be addressed:

1. General: Insert space before citations.

Response: Thanks! We have revised accordingly.

2. Introduction: When discussing the usefulness of toxins that target intracellular sites, it would be useful to list a few examples of other toxins that target other ion channels at intracellular sites. These are discussed in the discussion section but could already be mentioned in the introduction.

Response: We thank you for pointing this out, we have added some text in the introduction as follows:” Furthermore, it has also been found that venom peptide toxins can enter cells to act on the intracellular regions of ion channels or ion channels located on organelles, such as scorpion toxins WaTx targeting TRPA1, and Maurocalcine, Hadrucalcin and IpTxa targeting ryanodine receptor type 1 (RyR1).”

3. Results: It is difficult to see what HPLC fraction the asterisk in Figure 1a belongs to. Clarify this by using an arrow or something else that more clearly shows the active fraction.

Response: thanks! We have corrected it.

4. Results: In the caption of Figure 1a include information on how much venom was loaded on this HPLC run and how much of this was estimated to be rpTx1. Provide information on whether the venom was "milked" or extracted from homogenized glands? Also provide information on whether the toxin was found in its "truncated version". There's a dibasic cleavage site between the two domains and it would not be surprising but highly relevant if the C terminus "toxin"-like part was found without the N-terminal "carrier".

Response: Thanks. We agree that the point you raised is important for this study, and it is necessary

for us to clarify how rpTx1 was obtained. First, we added some information in the Figure 1 legend as follows: "The C18 RP-HPLC profile of ~ 3 mg crude venom which was collected from the centipede *S. subspinipes mutilans* by electrical stimulation was exhibited. The red arrow indicates the peak containing rpTx1, which accounts for approximately 1% of the crude venom."

Second, regarding whether rpTx1 is a mature toxin peptide or a precursor peptide containing an N-terminal precursor sequence and a C-terminal mature peptide sequence, we believe the former is more likely, although the latter cannot be entirely ruled out. No.1, the venom collection method used was electrical stimulation rather than venom gland homogenization. This is similar to how centipedes eject venom stored in venom gland, avoiding contamination by immature peptides enclosed within venom gland cells. It is generally assumed that the peptide toxins stored in venom gland are mature toxin peptides. No.2, mass spectrometry analysis of components separated by RP-HPLC did not reveal any molecular weight that could match the potential mature peptide sequences (truncated version) corresponding to rpTx1. The results of the mass spectrometry data are described in the supplementary documents (**Revised Supplementary Table 3**).

5. Results: 5'RACE requires information on related DNA sequences for at least one end. What sequences were used as the basis of primer design (degenerative primer)? This information should be provided in the results. What portion of the sequence was confidently identified by Edman sequencing? Presumably not all residues and not the entire toxin was sequenced by Edman. Similarly, RACE may not have provided the entire toxin region. This information should be provided.

Response: We apologize for the oversight caused on our part. It is 3'RACE but not 5'RACE. We have clarified the sequence determination of rpTx1 clearly in the revised manuscript as follows. To determine the protein sequence of rpTx1, we employed a combined approach of Edman degradation sequencing, 3' Rapid Amplification of cDNA Ends (3' RACE), and mass spectrometry. Initially, Edman degradation sequencing yielded the N-terminal 20 amino acid sequence: LKVEDLEPPTSYMKELKLAL (revised Supplementary data). Subsequently, based on the N-terminal sequence VEDLEPPT, degenerate primers (5'-GTNGARGAYTNGARCCNCCNAC-3', where N=A, C, G, or T; Y=C or T; R=A or G) were designed for 3' RACE. PCR was conducted using Tks Gflex DNA Polymerase (TaKaRa), and the PCR products were cloned into the T-Vector pMDTM20. DNA sequencing was then used to determine the gene sequence of the mature rpTx1 (revised Supplementary data). The theoretical relative molecular mass of the determined sequence was in full agreement with that identified by mass spectrometry. Therefore, we successfully obtained the full-length sequence of rpTx1. We have added these information in the Results section.

6. Results: The rationale for naming the toxin is unclear ("We named it rpTx1 (rational nomenclature: δ -SLPTX10-Ssm1a(King, Gentz et al., 2008)).") and should be better described. Is this based on other centipede peptides and/or on the activity at sodium channels etc.?

Response: Thank you for pointing this out. The rationale nomenclature for naming the toxin is based on the rules for peptide toxins from venomous animals introduced by King et al., which incorporate information about the biological origin of the peptide, its molecular target, and its relationship to known paralogs and orthologs into each toxin name. In our manuscript, based on BLAST results, we have determined that this toxin belongs to the centipede peptide toxin SLPTX10 family. Furthermore, according to our experimental results, it has been found to exhibit activity on the fast inactivation of Nav channels. These are the reasons for the naming of this toxin, and we

have already supplemented these descriptions in the results section as follows: “Based on the BLAST results, its function described below, and the rational nomenclature rules introduced by Glenn King et al. , the toxin is also named δ -SLPTX10-Ssm1a.”

7. Results: If BLAST did not identify any similar sequences, how were the other sequences (shown in Figure S2d) identified? This section is unclear and needs further clarification. The sequence of rpTx1 should be included in the alignment.

Response: Sorry for this misunderstanding. In the revised manuscript, we have clarified the issue as follows: “According to a BLAST search, rpTx1 exhibits 35-73% sequence identities with several centipede peptide toxins from centipedes. Among these toxins, Ssd1a and Ssd1b are isolated from the venom of *Scolopendra subspinipes dehaani* (Liu, Zhang et al., 2012), and the other are derived from the transcriptome analysis of venom glands of *Scolopendra morsitans* (Sm1a, Sm2a and Sm3a), *Cormocephalus westwoodi* (Cw1a and Cw2a), *Scolopendra alternans* (Sa2a), and *Ethmostigmus rubripes* (Er1a) (Undheim, Jones et al., 2014). Their functions remain unknown so far. The sequence comparison shows they share the same sequence architecture, composing of the long flexible N-terminal sequences and the C-terminal cysteine-rich sequences, and the arrangement patterns of cysteine residues are identical (**Supplementary Fig. 2d**).”

8. Results: Provide alphafold confidence values rather than just saying a high confidence structure was predicted or show a figure with the color code of alphafold confidence value in the Supporting information. There are typically regions of high and low confidence in alphafold predicted structures. This should be clarified.

Response: We thank you for this valuable comment. The three-dimensional structure of rpTx1 was predicted by AlphaFold2, and the prediction results showed that the overall structure of rpTx1 was with very high confidence, with a per-residue model confidence score (pLDDT) exceeding 90 for most regions, except for the C-terminal Lys, where the pLDDT ranged between 70 and 50. In the revised manuscript, we have added this data (Revised Supplementary Fig. 2e).

Revised Supplementary Figure 2e. Model confidence shows a high confidence in in Alphafold2 predicted rpTx1 structure.

9. Results: The authors state that the disulfide connectivity was "derived from the experimental data and predicted structure was...". Please describe what experimental data was obtained on the disulfide connectivity and whether this was done on the venom peptide or the recombinant peptide.

Response: We thank you for pointing this out, we have added some text in the results as follows: In order to determine the disulfide linkage of rpTx1, the truncated rpTx1 with short N-terminal was

yielded by expression in *Escherichia coli*. Partial reduction of the truncated rpTx1 with Tris (2-carboxyethyl) phosphine (TCEP) resulted in the obtaining of partially reduced peptides through RP-HPLC (**Supplementary Fig. 3a**). These peptides were subsequently alkylated with iodoacetamide and then subjected to Edman degradation, demonstrating the disulfide linkage of rpTx1: C1-C3, C2-C5, and C4-C6 (where the numbers represent the relative positions of the cysteine residues in the peptide sequences), which is consistent with the disulfide connectivity in the AlphaFold2 predicted structure.

Considering the limitations of the article's length, we have included some of the data in the supplementary materials (**Supplementary Fig. 3b-c**),

10. Results: For EC50 values, state what {plus minus} refers to when showing the results in the result section (SEM, SDEV, 95% confidence?).

Response: It is SEM, we have mentioned in the result section and Figure legends.

11. Supporting Information: Show the HPLC trace of the final, purified product (Figure S5c presumably only shows the pre-purified toxin).

Response: Thanks! We have added these in the revised manuscript.

12. Supporting Information: Show MS spectra before deconvolution, including all m/z precursor ions. The deconvoluted spectra is somewhat meaningless as the authors just labeled the peak with the deconvoluted mass.

Response: Thanks! We have corrected them accordingly.

13. Supporting Information: Spelling error in Figure S8e ("Sequence")

Response: Thanks! We have corrected it accordingly.

Referee #3:

In this manuscript, the authors report the discovery of a unique peptide toxin, rpTx1, derived from centipede venom. They utilized a comprehensive, multi-method approach involving electrophysiology, fluorescence imaging, mutagenesis, surface plasmon resonance, mutant cycle analysis, and molecular dynamics (MD) simulations to reveal the molecular interactions between the toxin and Nav channels. Notably, the authors identified the intrinsic inactivation particle of Nav channels as a novel receptor site for this centipede-derived peptide toxin, thereby demonstrating its capacity to modulate Nav channel activity. The functional assays are extensive and adequately illustrate the binding and modulatory mechanism.

Response: We thank you for your positive comments. Accordingly, we have addressed your concerns as follows, and made corrections in the revised manuscript.

Nonetheless, several aspects require further clarification or adjustment:

1. The toxin rpTx1 contains two functional domains: an N terminal domain for cell penetration and a C terminal domain for modulating channel activity. The C terminal domain has three disulfide

bonds. What's the function of these disulfide bonds, considering they would be broken when the toxin enters the cytosol? The authors also attempted to co-express the toxin with Nav channels and obtained similar electrophysiological results. Perhaps they could mutate the Cys residues of the toxin when co-expressing with Nav channels.

Response: We thank you for your comments. Disulfide-rich peptide toxins are the dominant compounds in most animal venoms and they are the major contributors to the venom's activity, characterized by a diversity of structural motifs. Disulfide bonds not only provide these peptide toxins with exceptional chemical and thermal stability but are also crucial for their structure and function. Numerous evidences indicate that disrupting the disulfide bonds in the toxins completely abolishes their activity. We have also supplemented our study with the recombinant expressed mutant rpTx1-C61A&C62A, which contains cysteine mutations. Intracellular application of rpTx1-C61A&C62A revealed that its ability to inhibit the fast inactivation of rNav1.8 was completely lost (**Revised Supplementary Fig.7c**). Similarly, disrupting the toxin's disulfide bonds in the β 4-rpTx1-C61A&C62A also rendered it completely inactive (**Revised Supplementary Fig.7e**). Therefore, we believe that the disulfide bonds in the C-terminal domain of rpTx1 are critical for its inhibition of Nav channels fast inactivation and are relatively stable within cells. Considering the importance of the disulfide bonds on the structure and function of rpTx1, we believe that the rpTx1 enters cell membranes and localizes inside cytoplasm, its disulfide bonds should remain intact.

Revised Supplementary Figure 7c and 7e. (c) Potency of 0.1 μ M WT rpTx1 and mutants measured on rNav1.8. Note that M45A, E60A, T66A and C61A&C62A mutations remarkably reduced toxin's availability on the channel (one-way ANOVA with post hoc analysis using Dunnett's multiple comparisons test, $n = 5-18$). (e) Representative traces of current families were recorded from ND7/23 cells expressing rNav1.8 co-expressed with 3.5 μ g β 4-rpTx1 ($n=5$) or β 4-rpTx1-C61A&C62A vector ($n=6$). Scale bar: 0.5 nA, 10 ms. Data are presented as mean \pm S.E.M. **** $p < 0.0001$.

2. In Figure 2, control current traces are not shown alongside the low-concentration toxin traces. Please add control traces for comparison or provide an explanation for their absence.

Response: We thank you for pointing this out, we have added the control traces in the revised manuscript.

3. In Figure 3, the IFMT-related mutants abolished the effect of rpTx1 on the Nav1.8 persistent current. Given that rpTx1 also induces a shift in the voltage-dependent inactivation of Nav1.8, do these IFMT-related mutants similarly affect the inactivation V_{half} shift caused by the toxin? Please provide further details on this point.

Response: We thank you for this valuable comment. We test the effect of rpTx1 on the voltage-

dependent inactivation of mutant channels (Nav1.8 I1434A, M1436A and T1437A). As shown in the **Revised Supplementary Fig. 9c**, rpTx1 did not significantly shift the voltage-dependent inactivation of these mutant channels, but it enhanced the no-inactivation component in the I1434A mutant. These results are consistent with the effect of rpTx1 on the persistent current of these mutant channels.

Revised Supplementary Figure 9c. The effect of rpTx1 on the steady-state inactivation of rNav1.8 mutants I1434A (n = 9 for control, n = 8 for rpTx1), M1436A (n = 5 for control, n = 6 for rpTx1) and T1437A (n = 5 for control, n = 6 for rpTx1). Data are presented as mean \pm S.E.M.

4. For the MD simulation section, additional data should be presented. Specifically, it would be beneficial to include the interaction frequency or distance change of T66-T1436 (1437) and T66-I1433 (1434) over the 200 ns simulation period. For the metadynamics simulations, it would be helpful to explain a bit how these five structures (B1, B2, T1, U1 and U2) were generated and if they are at their free energy minima?

Response: We appreciate you for raising this insightful question. In the revised manuscript, we performed the MD simulation again and a new **Fig.5** was prepared. In our investigation of the dynamics of the IFMT motif, we found that the region of hNav1.8 subsequent to the residue 1730 plays a significant role in the conformational change of the IFMT motif (Clairfeuille, Cloake et al., 2019). However, this region had been overlooked due to its absence in the selected cryo-EM structure. Therefore, we remodeled the structure by adding residues 1729–1956 in the MD simulations, which led to a more accurate free energy landscape for the IFMT motif than the previous version. According to the free energy landscape and the subsequent conformational clustering, the motion of the IFMT motif should involve three major states (B1, B2, and U) and a transition state between B2 and U (**Fig. 5a-d**); For the U state, three major representative conformations (U1-3) were identified. Indeed, the different representative conformations of each state correspond to the same minimum within the energy landscape, and their classification was achieved through conformational clustering. The positions of these states in the defined conformational space, along with their corresponding free energy values, are shown in **Fig. 5e**. The characteristics and clustering of these states are presented in Supplementary **Table 2**. Please note that conformational clustering in each minimum was conducted based on heavy atoms of specific pocket residues (V1269, V1270, A1273, P1605, A1606, N1609, I1610, N1715, I1718, A1719) along with I1433, F1434 and M1435, using clustering threshold of 3.0 Å.

Additionally, we reconducted hNav1.8 and rpTx1 complex docking by using a new docking program

and found that rpTx1 prefers to dock to the IFMT motif in the U state, and one of the three representative conformations (U3) produced a more expected binding complex with rpTx1 (**Fig. 5f**). We further recorded the distance changes between specific residues during the 200 ns MD simulation based on the docking complex (rpTx1-U3), as shown in **Fig. 5g**.

Revised Figure 5. The docking-based prediction of rpTx1-hNav1.8 complex. (a-d) The representative conformation of different states of the IFMT motif generated by metadynamics simulation, including the Bound State I (B1), the Bound State II (B2), the Transitional state (T), and the Unbound state (U). (e) The free energy landscape given by metadynamics simulation illustrating different states of the IFMT motif. (f) The global and local view of the docking complex of rpTx1-Nav1.8-U-3, and local view of the binding between rpTx1 and the IFMT motif by the end of 200 ns MD simulation. (g) The distance of two atom pairs illustrating the continuous contact between rpTx1-T66 and hNav1.8-I1433/T1436. In local view of (a-d), four residues I-F-M-T of the IFMT motif are colored in pink, while the linker region and the IFMT receptor site are colored in green and blue, respectively. In global view of (f), the rpTx1 molecule is represented by cyan cartoon, and hNav1.8 is drawn as surface with different region in varied colors, i.e. DI pore in dar orange, VSD IV in sienna, DIV pore in yellow, IFMT motif in pink, and the linker region in green, respectively. As for membrane components, PA and OL molecules are given as violet stick models, while PC molecules are given as light violet surfaces.

Clairfeuille T, Cloake A, Infield DT, Llongueras JP, Arthur CP, Li ZR, Jian Y, Martin-Eauclaire MF, Bougis PE, Ciferri C, Ahern CA, Bosmans F, Hackos DH, Rohou A, Payandeh J (2019) Structural basis of alpha-scorpion toxin action on Nav channels. *Science* 363

5. Functional assays indicate specific interactions within T66-I1434 and T66-T1437. However, these interactions appear absent in the initially predicted structure. Could you please provide a reasonable explanation for this discrepancy? Additionally, the distances of the specific interactions T66-T1436 (1437) and T66-I1433 (1434) are 5.5 Å or 6.8 Å which may be too far to form the strong interaction between the toxin and the Nav1.8. Further elaboration on these points is recommended.

Response: We would like to thank the reviewer for their valuable suggestions. After making the revisions described in response to point 4, we obtained more reasonable results, as shown in **Fig. 5f and 5g**. The interaction between rpTx1-T66 and hNav1.8-I1433/T1436 is likely to be primarily driven by hydrophobic interactions. This is because the side-chain hydroxyl group of rpTx1-T66 folds inward and forms a highly stable hydrogen bond with the backbone atoms of rpTx1-E60, leaving the hydrophobic part of T66's side chain exposed at the protein-protein interface, potentially facilitating hydrophobic interactions between rpTx1-T66 and hNav1.8-I1433/T1436 (**Fig. 5f**). This point has been clarified in the revised manuscript.

To further investigate the strength of the hydrophobic interaction between rpTx1-T66 and hNav1.8-I1433/T1436, we measured the distance between the CG2 atom of T66 and the CB atom of I1433, as well as the distance between the CG2 atom of T66 and the CG2 atom of T1436. As shown in **Fig. 5g**, these two distances fluctuate between 4 and 8 Å during the simulation, which is within the typical range for hydrophobic interactions. It is noted that considering the alkyl side chain of hNav1.8-I1433, this residue is more likely to play a dominant role in hydrophobic interactions at the interface. The distance between the CG2 atom of rpTx1-T66 and the CB atom of hNav1.8-I1433 remained within 6 Å for the majority of the MD simulation, suggesting a sustained hydrophobic interaction between these two residues. These results support the experimental data, and suggested that rpTx1-T66 may interact directly with hNav1.8-I1433/T1436.

6. Please re-check the residue number of IFMT in Figure 4, Figure 5 and the manuscript. Please verify and correct as necessary.

Response: We thank you for pointing this out. After careful verification, I confirm that the residue number of IFMT in Figures 4 and 5 is correctly labeled. The reason for the difference in the residue number of IFMT between Figure 4 and Figure 5 is that the channel we tested using the patch-clamp technique is rat Nav1.8, whereas the channel used for molecular docking and MD simulations is human Nav1.8. Both channels share very high sequence identity, but there is a one-position difference in the amino acid number of the IFMT motif; specifically, I1434A in rat Nav1.8 corresponds to I1433A in human Nav1.8. Therefore, to distinguish between them, we have added species information before the channel notation. We apologize again for any confusion caused.

ADDITIONAL SPECIFIC COMMENTS BY REFEREE #1

For me the most concerning result is the "inhibition" of extracellular peptide shown after incubation for 1 hour. This is shown in Fig 2c (and was also noted by Reviewer #2), where despite the high concentration and long incubation period, only 13.3 - 36.4% of cells showed significant inhibition of inactivation.

Response: Thanks! This is indeed a point worth noting. In the revised manuscript, we conducted this experiment again. We increased cell numbers tested and performed quantitative analysis. We

treated ND7/23 cells expressing rNav1.8 by adding 5 μ M rpTx1 into the bath and incubating for 30 min and found that approximately 30.6% cells exhibited delayed fast inactivation compared with control (Fig 2c), which was consistent with the fluorescent data indicating approximately 28.9% cells were labelled with rpTx1-FITC (Fig 2a). These data are actually similar to previous ones. Our repeated experimental observations consistently show that only a small subset of cells exhibits a noticeable effect. This may be explained by the cell penetrating efficiency of the peptide. Crossing the cell membrane is inherently challenging for macromolecules. Numerous studies have demonstrated that the cell membrane crossing efficiency of cell-penetrating peptides (CPPs) is generally low; even the most effective CPPs identified to date do not achieve 100% efficiency. Furthermore, as commonly observed in liposome-mediated transient transfection experiments, transfection efficiency typically hovers around 30%. For rpTx1, the N-terminal sequence of rpTx1 functions as a cell-penetrating peptide, carrying the C-terminal sequence as a cargo into cells. Compared to most CPPs, its molecular weight is significantly larger, which understandably results in lower cell penetrating efficiency. Consequently, only a small number of cells are able to internalize a sufficient amount of rpTx1 to exhibit a significant inhibition of fast inactivation. This, in turn, supports the conclusion that adequate intracellular entry of rpTx1 and its binding to intracellular targets are prerequisites for its functional effects.

Combined with lack of convincing data showing penetration of the peptide intracellularly, the proposed mechanism of action is not strongly supported by the experimental data. This data is also at odds with the (single current trace) shown in Fig 1c that implies inhibition of inactivation after 30 min incubation.

At a minimum I would require additional experiments showing this effect; and more experimental detail in general.

Response: We understand and acknowledge your concerns. Therefore, building on the original data, we have conducted a substantial number of additional experiments to obtain evidence from multiple perspectives to support the conclusions of this study: rpTx1 is a novel bifunctional peptide toxin that can cross cell membranes to act on the inactivation particle, inhibiting the fast inactivation of Nav channels. This represents a novel mechanism by which peptide toxins regulate Nav channels, with the inactivation particle identified as a new binding site for peptide toxins.

1. FITC-labeled rpTx1 localized within cells provides direct evidence that rpTx1 can cross the membrane into the intracellular environment. Furthermore, the proportion of cells displaying intracellular fluorescence signals matches the proportion of cells in which the same concentration of rpTx1 induces significant inhibition of fast inactivation of rNav1.8 currents from the extracellular side. On the other hand, we found that the N-terminal sequence of rpTx1 is critical for its cell penetrating activity. Mutants lacking cell penetrating capability also fail to affect Nav channel currents from the extracellular side.

2. Direct intracellular application of rpTx1, even at low concentrations (e.g. 0.1 μ M), can immediately and completely inhibit the fast inactivation of rNav1.8 currents. Additionally, intracellular fusion expression of rpTx1 with eGFP or the beta subunits significantly suppresses fast inactivation. In contrast, extracellular application of rpTx1, even at high concentrations (e.g. 5 μ M) and long durations (30 min), results in only a small proportion of cells responding. This stark difference between intracellular and extracellular application indicates that the entry of rpTx1 into the cell is a prerequisite for its functional activity.

3. Through comprehensive experiments, we have demonstrated that the IFMT motif is the direct binding site of rpTx1 on Nav channels. Once rpTx1 interacts with the IFMT motif, it captures the motif in the unbound state, thereby restricting its interaction with the receptor site and obstructing fast inactivation. This represents the molecular mechanism by which this toxin inhibits the fast inactivation of Nav channels. Furthermore, disrupting the interaction between rpTx1 and the IFMT motif abolishes the ability of rpTx1 to inhibit fast inactivation, even when it is present intracellularly. These findings indicate that rpTx1's entry into the cell and its binding to the IFMT motif located in the intracellular region of the Nav channels are sufficient conditions for its functional activity. Compared to activity analysis and binding site analysis, the direct evidence for rpTx1's cell penetrating activity is relatively sparse. This may inevitably raise doubts about the authenticity of rpTx1's ability to cross the membrane. However, rpTx1's cell penetrating activity should not be considered in isolation. Instead, it should be analyzed comprehensively alongside its activity and binding site interactions. The data on rpTx1's activity and binding site, in fact, provide indirect but sufficient evidence to support its cell penetrating activity, demonstrating that rpTx1 does undergo this process. While the details of the membrane crossing process and its molecular mechanism remain unclear, this does not detract from the validity of our primary conclusions.

ADDITIONAL SPECIFIC COMMENTS BY REFEREE #3

After thoroughly reviewing the comments provided by Referee #1 and Referee #2, I would like to offer my perspective and contribute to the discussion regarding the minimal additional experiments and analyses required to support the authors' claims within a feasible time frame of approximately three months.

Recommendations for Minimal Additional Experiments:

Based on the comments from all three reviewers, I believe the following minimal set of experiments and analyses would be necessary to support the authors' conclusions while being realistically achievable within three months:

1. Venom Processing: Investigate the possibility that rpTx1 may be a precursor peptide, including investigating whether a truncated version of rpTx1 exists in venom and an exploration of potential cleavage sites. This is critical, as Reviewer 2 noted that the N-terminal domain may be cleaved in the mature toxin.

Response: Thanks! This is really important. We have provided a detailed description of the venom collection process and the separation of the peptide, as well as determined the molecular weights of the venom fractions separated by RP-HPLC. Based on this, we have provided a thorough explanation in response to Point 4 raised by Reviewer 2, which will not be repeated here.

2. Electrophysiological Assays: Additional electrophysiological measurements to include control current traces in Figure 2 and to explore the $V_{1/2}$ shift in the IFMT-related mutants (point 3 of my review) are necessary. This will clarify the functional impact of the toxin on voltage-dependent inactivation and help address the reviewers' concerns about the thoroughness of the functional assays. Meanwhile, perform detailed electrophysiological studies, including: Time course of peptide activity following extracellular application; Concentration-response experiments with proper controls, ensuring that the differences between mutants (such as I1434A) and wild-type are statistically validated; Quantification of persistent currents to better characterize the peptide's effects on Nav channels; Performing kinetic studies such as "recovery from inactivation" to clarify

the peptide's effects on sodium channels.

Response: Thanks! We have supplemented corresponding experiments to address these questions. Please refer to the responses under the respective points mentioned above.

3. Cell Penetration Quantification: As raised by reviewer 1 and 2, provide robust, quantitative evidence of rpTx1 cell penetration, using alternative methods to FITC labeling. This could include quantitative flow cytometry or biochemical uptake assays. These are essential, as cell penetration is a critical claim of the study.

Response: Thanks! We have supplemented corresponding experiments to address these questions. Please refer to the responses under the respective points mentioned above.

4. MD Simulation Data: Expanding the molecular dynamics data to include the interaction frequencies or distance changes between specific residues (point 4 of my review) will provide deeper insight into the toxin-channel interactions, addressing concerns from all reviewers about the validity of these computational models. Additionally, clarifying the structural energy minima for the meta dynamics simulations (point 4 of my review) will be helpful in interpreting the simulation results in relation to the experimental data. More robust MD simulation data and additional electrophysiological assays could offer deeper insights into alternative toxin mechanisms or support further hypotheses regarding potential dual actions of the toxin.

Response: Thanks! We have supplemented corresponding experiments to address these questions. Please refer to the responses under the respective points mentioned above.

5. Clarification of Discrepancies in Interaction Distances: Providing further elaboration on the discrepancies in interaction distances between T66-T1436 (1437) and T66-I1433 (1434) (point 5 of my review) will strengthen the manuscript's mechanistic claims. Although this may require some additional analysis or computational refinement, it is a necessary step to resolve concerns about the structural data.

Response: Thanks! We have supplemented corresponding experiments to address these questions. Please refer to the responses under the respective points mentioned above.

Dear Prof. Liu,

Thank you for submitting your manuscript for consideration by the EMBO Journal. It has now been seen by the three original referees from the previous round whose comments are enclosed. As you will see, all three referees express interest in your manuscript and are broadly in favour of publication, pending satisfactory minor revision.

Given the referees' positive recommendations, I would like to invite you to submit a revised version of the manuscript, addressing the comments of all three reviewers.

Additionally please keep special attention to the more technical comments provided below by our editorial assistants and properly address them in your revision.

I should add that it is EMBO Journal policy to allow only a single round of revision, and acceptance of your manuscript will therefore depend on the completeness of your responses in this revised version.

We generally allow three months as standard revision time. As a matter of policy, competing manuscripts published during this period will not negatively impact on our assessment of the conceptual advance presented by your study. However, we request that you contact the editor as soon as possible upon publication of any related work, to discuss how to proceed.

Thank you for the opportunity to consider your work for publication. I look forward to your revision.

Yours sincerely,

Yehu Moran
Academic Editor
The EMBO Journal

We realize that it is difficult to revise to a specific deadline. In the interest of protecting the conceptual advance provided by the work, we recommend a revision within 3 months (27th May 2025). Please discuss the revision progress ahead of this time with the editor if you require more time to complete the revisions. Use the link below to submit your revision:

Specific comments by Editorial Assistant:

*AUTHOR CHECKLIST: included but several fields have not been completed. Please fix.

*Figures in separate files: No. The main figures should be removed from the manuscript text and uploaded as separate, high resolution figure files. The legends should stay in the manuscript, after the References. The supplementary figures should be renamed "Figure EV1" - 11, also be removed from the manuscript text and uploaded as individual figure files. Their legends should also be in the manuscript text, after the main figure legends, and under the heading "Expanded View Figure Legends"

*AUTHORS: Qiangling Xie is missing in the system. Please correct when submitting your revision.

*ORCID ID: missing for Zhou. A link was sent by one of our assistants. Please make sure to fix this and contact us should you encounter any problems.

*DATA AVAILABILITY SECTION: in but incorrect format. Please correct.

*FUNDING: please enter all funding information in the comments, including project numbers, into the lines provided above.

*Author Contributions: Please remove from the manuscript text and provide only via the system.

*Conflict of Interest: Please rename "Disclosure and competing interests statement".

*REFERENCE FORMAT: limit the number of author names listed to 10 before et al.

*DATASET EV LEGENDS: Table 1 has a second sheet - is this source data? If this is the case, please remove it from the Table 1 file and upload it as a source data file. The supplementary table 3 is uploaded as supplementary table 2. Please rename this table "Table EV1". The the third supplementary table is source data (for Table EV1?), please upload it as a source data file.

*APPENDIX 1 FILE WITH Table of Contents: There is a file with supplementary information on "N-terminal Edman degradation sequencing" - this file should be renamed "Appendix", a table contents should be added to the first page. The information should be named "Appendix Supplementary Information". A short legend with a description should be added.

*SOURCE DATA: please upload a completed source data checklist, and upload the requested source data as one file per figure, i.e. one file for Figure 1, one for Figure 2, etc.

*REAGENT TABLE: missing, please upload as a separate file using the template provided in the author guidelines.

*SYNOPSIS IMAGE: Not provided. Please provide.

*SYNOPSIS TEXT: Not provided. Please provide.

Notes: "Materials and Methods" should be renamed "Methods".

- Figure Legends (main + EV): 1. Please note that the exact p values are not provided in the legends of figures 2B, C, E; 3B, F. These must be provided in the revision.

2. Please note that information related to n is missing in the legend of figure 4A. Please provide this in the revision.

Referee #1:

The manuscript by Zhou et al describe the discovery and activity of a novel centipede venom-derived peptide that modulates inactivation of NaV channels via interaction with the intracellular inactivation particle. The peptide is proposed to cross cell membrane via an N-terminal domain, while the C-terminal domain interacts with the inactivation gate to inhibit fast inactivation. Overall, it is a interesting finding that further extends the toolkit of venom peptides with activity at NaV channels.

I thank the authors for taking on board comments and suggestions. The manuscript is overall much improved and the proposed mechanism of action is intriguing. Unfortunately there are still several areas of concern - most of which could be addressed by more careful wording, and less definitive claims, by the authors .

The immunofluorescence experiment do not, and cannot, show that "eGFP-rpTx1 is primarily localized within the cytoplasm on the inner side of the plasma membrane". The resolution is simply not sufficient, and it is unclear that overlap of fluorescence signal (or lack thereof) from DiD could determine this either. The issue is not whether or not the peptide is expressed on the intracellular side (of course transfection with a plasmid encoding this peptide will at some point be intracellular) - the issue is the claim that the eGFP fusion tethers the peptide close to the intracellular side of the membrane specifically. For beta subunit fusion proteins - it is difficult to see how these would exist in a "free" state, or freely diffusible. It is more likely they are associated with intracellular membranes. It is also unclear why a membrane permeable peptide would not be able to cross from the intracellular to the extracellular compartment - it would just be easier to visualise intracellular peptide as any extracellular peptide would likely diffuse away. This issue is one that could be addressed with careful wording. Discussion on the low number of cells that show fluorescence is also important. At a minimum, there should be robust discussion and acknowledgement that only a small % of cells show a toxin effect, and that other mechanisms of cell penetration cannot be ruled out without further

experiments.

I still find the explanation for preferential activity of the peptide for NaV1.8 (as opposed to NaV1.9) problematic - yes it is true that our understanding of channel gating is incomplete and this may be much more complicated - but your explanation that this is due to a lower affinity of the IFM motif for NaV1.8 - ignoring the even slower fast inactivation of NaV1.9 - feels contrived. It would be much more reasonable, rather than speculating on the binding affinity being a driving force, to state that there is currently no good explanation for the observation that the peptide preferentially inhibits fast inactivation of NaV1.8. You could speculate that this may be due to binding affinities, noting that NaV1.9.

Overall I strongly suggest that the manuscript is worded much more cautiously, included the ability of the peptide to cross cell membranes, and mechanisms proposed for subtype selective effects.

Referee #2:

I greatly appreciate that the authors addressed all my concerns by either providing new, compelling data that supports the major claims or by providing more details on the experimental procedures and results. As stated in my previous review, I think that this toxin represents a novel tool to study Nav channel inactivation and represents a beautiful example of the diverse mechanisms evolved by venomous animals to subdue their prey and defend against predators. This is a very exciting and relevant study that will be of broad interest to the EMBO J readership and the scientific community.

I only have a few minor comments left:

Please clarify if the results outlined under "RpTx1 contains two functional domains" were from a single experiment or replicate studies. Currently it just states the number of cells but not whether these were done with biological replicates.

Line 26, insert "that": this study revealed that the..."

Line 88: change to "such as the scorpion toxin WaTx targeting TRPA1,.... References are needed for these prior findings.

Line 134: delete "centipede": "several peptide toxins..."

Line 140: delete "so far"

Line 145: remove "Glenn": "by King et al."

Line 158-159: change "peptide" to "peptide isomers" or just isomers" as these are the same peptide in its different forms

Line 187: insert "higher": "10-fold higher potency for"

Line 235: delete "just"

Line 269-271: Please remove "chemical stability" and possibly also "thermal stability" unless this was tested in CD experiments". Although it is a fair assumption that the disulfides provide chemical stability there is not data supporting this and the disulfides may merely serve to provide the correct fold needed for activity rather than stability.

Line 271-272: Clarify that this statement refers to other, disulfide-containing toxins. This is not clear as there is no reference provided here.

Line 534-537: Please revise grammar.

Referee #3:

In the revised manuscript, the authors have conducted a comprehensive series of experiments and analyses to address the reviewers' concerns and substantiate their conclusions. The results are thorough and effectively demonstrate the binding and modulatory mechanisms, providing insight into the interaction between the toxin and Nav channels. Furthermore, the new MD simulation data offer strong support for the proposed hypotheses regarding the potential dual actions of the toxin. Based on these revisions, I have no further concerns.

Referee #1:

The manuscript by Zhou et al describe the discovery and activity of a novel centipede venom-derived peptide that modulates inactivation of NaV channels via interaction with the intracellular inactivation particle. The peptide is proposed to cross cell membrane via an N-terminal domain, while the C-terminal domain interacts with the inactivation gate to inhibit fast inactivation. Overall, it is a interesting finding that further extends the toolkit of venom peptides with activity at NaV channels.

I thank the authors for taking on board comments and suggestions. The manuscript is overall much improved and the proposed mechanism of action is intriguing. Unfortunately there are still several areas of concern - most of which could be addressed by more careful wording, and less definitive claims, by the authors .

The immunofluorescence experiment do not, and cannot, show that "eGFP-rpTx1 is primarily localized within the cytoplasm on the inner side of the plasma membrane". The resolution is simply not sufficient, and it is unclear that overlap of fluorescence signal (or lack thereof) from DiD could determine this either. The issue is not whether or not the peptide is expressed on the intracellular side (of course transfection with a plasmid encoding this peptide will at some point be intracellular) - the issue is the claim that the eGFP fusion tethers the peptide close to the intracellular side of the membrane specifically. For beta subunit fusion proteins - it is difficult to see how these would exist in a "free" state, or freely diffusible. It is more likely they are associated with intracellular membranes. It is also unclear why a membrane permeable peptide would not be able to cross from the intracellular to the extracellular compartment - it would just be easier to visualise intracellular peptide as any extracellular peptide would likely diffuse away. This issue is one that could be addressed with careful wording. Discussion on the low number of cells that show fluorescence is also important. At a minimum, there should be robust discussion and acknowledgement that only a small % of cells show a toxin effect, and that other mechanisms of cell penetration cannot be ruled out without further experiments.

I still find the explanation for preferential activity of the peptide for NaV1.8 (as opposed to NaV1.9) problematic - yes it is true that our understanding of channel gating is incomplete and this may be much more complicated - but your explanation that this is due to a lower affinity of the IFM motif for NaV1.8 - ignoring the even slower fast inactivation of NaV1.9 - feels contrived. It would be much more reasonable, rather than speculating on the binding affinity being a driving force, to state that there is currently no good explanation for the observation that the peptide preferentially inhibits fast inactivation of NaV1.8. You could speculate that this may be due to binding affinities, noting that NaV1.9.

Overall I strongly suggest that the manuscript is worded much more cautiously, included the ability of the peptide to cross cell membranes, and mechanisms proposed for subtype selective effects.

Referee #2:

I greatly appreciate that the authors addressed all my concerns by either providing new, compelling data that supports the major claims or by providing more details on the experimental procedures and results. As stated in my previous review, I think that this toxin represents a novel tool to study Nav channel inactivation and represents a beautiful example of the diverse mechanisms evolved by venomous animals to subdue their prey and defend against predators. This is a very exciting and relevant study that will be of broad interest to the EMBO J readership and the scientific community.

I only have a few minor comments left:

Please clarify if the results outlined under "RpTx1 contains two functional domains" were from a single experiment or replicate studies. Currently it just states the number of cells but not whether these were done with biological replicates.

Line 26, insert "that": this study revealed that the..."

Line 88: change to "such as the scorpion toxin WaTx targeting TRPA1,... References are needed for these prior findings.

Line 134: delete "centipede": "several peptide toxins..."

Line 140: delete "so far"

Line 145: remove "Glenn": "by King et al."

Line 158-159: change "peptide" to "peptide isomers" or just "isomers" as these are the same peptide in its different forms

Line 187: insert "higher": "10-fold higher potency for"

Line 235: delete "just"

Line 269-271: Please remove "chemical stability" and possibly also "thermal stability" unless this was tested in CD experiments". Although it is a fair assumption that the disulfides provide chemical stability there is not data supporting this and the disulfides may merely serve to provide the correct fold needed for activity rather than stability.

Line 271-272: Clarify that this statement refers to other, disulfide-containing toxins. This is not clear as there is no reference provided here.

Line 534-537: Please revise grammar.

Referee #3:

In the revised manuscript, the authors have conducted a comprehensive series of experiments and analyses to address the reviewers' concerns and substantiate their conclusions. The results are thorough and effectively demonstrate the binding and modulatory mechanisms, providing insight into the interaction between the toxin and Nav channels. Furthermore, the new MD simulation data offer strong support for the proposed hypotheses regarding the potential dual actions of the toxin. Based on these revisions, I have no further concerns.

Inherent fast inactivation particle of Nav channels as a new binding site for a neurotoxin

Xi Zhou^{1,2,4,#,*}, Haiyi Chen^{3,5#}, Shuijiao Peng^{1,2,4}, Yuxin Si^{1,2,4}, Gaoang Wang³, Li Yang^{1,2,4}, Qing Zhou^{1,2,4}, Minjuan Lu^{1,2,4}, Qiaoling Xie^{1,2,4}, Xi He^{1,2,4}, Meijing Wu^{1,2,4}, Xin Xiao^{1,2,4}, Xiaoqing Luo^{1,2,4}, Xujun Feng^{1,2,4}, Wenxing Wang^{1,2,4}, Sen Luo^{1,2,4}, Yaqi Li^{1,2,4}, Jiaxin Qin^{1,2,4}, Minzhi Chen^{1,2,4}, Qianqian Zhang^{1,2,4}, Weijun Hu^{1,2,4}, Songping Liang^{1,2,4}, Tingjun Hou^{3,*}, Zhonghua Liu^{1,2,4,*}

Our revisions are described below in a point-by-point manner. For easier reading, the reviewers' comments are marked in black, responses are marked in red. Questions are copy-pasted from the EMBO J Editor's decision correspondence. Changes in the main text of the revised manuscript are highlighted in red.

Referee #1:

The manuscript by Zhou et al describe the discovery and activity of a novel centipede venom-derived peptide that modulates inactivation of NaV channels via interaction with the intracellular inactivation particle. The peptide is proposed to cross cell membrane via an N-terminal domain, while the C-terminal domain interacts with the inactivation gate to inhibit fast inactivation. Overall, it is a interesting finding that further extends the toolkit of venom peptides with activity at NaV channels.

I thank the authors for taking on board comments and suggestions. The manuscript is overall much improved and the proposed mechanism of action is intriguing. Unfortunately there are still several areas of concern - most of which could be addressed by more careful wording, and less definitive claims, by the authors .

Response: Thanks for your comments. According to your suggestions, we have revised the relevant description and conclusion to make them more reasonable in the revised manuscript.

The immunofluorescence experiment do not, and cannot, show that "eGFP-rpTx1 is primarily localized within the cytoplasm on the inner side of the plasma membrane". The resolution is simply not sufficient, and it is unclear that overlap of fluorescence signal (or lack thereof) from DiD could determine this either. The issue is not whether or not the peptide is expressed on the intracellular side (of course transfection with a plasmid encoding this peptide will at some point be intracellular) - the issue is the claim that the eGFP fusion tethers the peptide close to the intracellular side of the membrane specifically. For beta subunit fusion proteins - it is difficult to see how these would exist in a "free" state, or freely diffusible. It is more likely they are associated with intracellular membranes.

Response: Thanks for your comments. In the revised manuscript, we have toned down this part of the description and only stated that rpTx1 can be expressed intracellularly through fusion expression. Additionally, the relevant description you mentioned in your concerns was something we included in our response to your first-round review comments and was not mentioned in the main text. We apologize for any confusion this may have caused.

It is also unclear why a membrane permeable peptide would not be able to cross from the intracellular to the extracellular compartment - it would just be easier to visualise intracellular peptide as any extracellular peptide would likely diffuse away. This issue is one that could be addressed with careful wording.

Response: Thanks for your comments. In the revised manuscript, we added the description between the lines 221-224. "Considering rpTx1 added intracellularly could possibly pass through the cell membrane into the extracellular space, we cannot exclude the possibility that intracellular rpTx1 exits into the extracellular space and exerts its effect of inhibiting the fast inactivation of Nav channels."

Discussion on the low number of cells that show fluorescence is also important. At a minimum, there should be robust discussion and acknowledgement that only a small % of cells show a toxin effect, and that other mechanisms of cell penetration cannot be ruled out without further experiments.

Response: Thanks for your comments. In the revised manuscript, we added the discussions between the lines 510-521. "Our observations show that only a small subset of cells exhibit sufficient entry of rpTx1 and a noticeable inhibition of fast inactivation, which indicates the cell penetrating efficiency of the peptide is not high. Crossing the cell membrane is inherently challenging for macromolecules. Numerous studies have demonstrated that the cell membrane crossing efficiency of cell-penetrating peptides (CPPs) is generally low; even the most effective CPPs identified to date do not achieve 100% efficiency. The N-terminal sequence of rpTx1 functions as a CPP, carrying the C-terminal sequence as a cargo into the cell. Compared to most CPPs, its molecular weight is significantly large, which understandably results in low cell penetrating efficiency. The cell penetrating efficiency of rpTx1 is influenced by various factors, and whether it shares the same mechanism as CPPs or operates through a different one is crucial for explaining the effect of rpTx1. This requires us to conduct further research."

I still find the explanation for preferential activity of the peptide for Nav1.8 (as opposed to Nav1.9) problematic - yes it is true that our understanding of channel gating is incomplete and this may be much more complicated - but your explanation that this is due to a lower affinity of the IFM motif for Nav1.8 - ignoring the even slower fast inactivation of Nav1.9 - feels contrived. It would be much more reasonable, rather than speculating on the binding affinity being a driving force, to state that there is currently no good explanation for the observation that the peptide preferentially inhibits fast inactivation of Nav1.8. You could speculate that this may be due to binding affinities, noting that Nav1.9.

Overall I strongly suggest that the manuscript is worded much more cautiously, included the ability of the peptide to cross cell membranes, and mechanisms proposed for subtype selective effects.

Response: Thanks for your comments. In the revised manuscript, we added discussions between the lines 541-556. "Thereby, it provides a possible explanation for the stronger activity of rpTx1 on Nav1.8. Considering Nav1.9 has slower fast inactivation than Nav1.8, rpTx1 should be more potent on Nav1.9 than the latter, but our data are the opposite. This suggests that the explanation mentioned above may not be applicable to Nav1.9, and other factors may play a more significant role. Additionally, our molecular dynamics simulations have revealed that, apart from the IFMT motif, more amino acid residues near this motif also participate in binding with rpTx1. This may also influence the varying activity of rpTx1 on different Nav channel subtypes. Therefore, we must acknowledge that even for Nav1.8, the mechanism we proposed is a simplified model. Actually, the fast inactivation mechanism of Nav channels is likely more complex than currently understood, and the underlying mechanisms responsible for rpTx1's selectivity on Nav channel subtypes remain unclear, necessitating further studies to elucidate the precise mechanism of action of the inactivation gate."

Referee #2:

I greatly appreciate that the authors addressed all my concerns by either providing new, compelling data that supports the major claims or by providing more details on the experimental procedures and results. As stated in my previous review, I think that this toxin represents a novel tool to study Nav channel inactivation and represents a beautiful example of the diverse

mechanisms evolved by venomous animals to subdue their prey and defend against predators. This is a very exciting and relevant study that will be of broad interest to the EMBO J readership and the scientific community.

Response: We greatly appreciate the reviewer for the positive comment.

I only have a few minor comments left:

Please clarify of the results outlined under "RpTx1 contains two functional domains" were from a single experiment or replicate studies. Currently it just states the number of cells but not whether these were done with biological replicates.

Response: Thanks for this suggestion. It is three independent experiments, and we have mentioned in the Figure legend of the revised manuscript.

Line 26, insert "that": this study revealed that the..."

Line 88: change to "such as the scorpion toxin WaTx targeting TRPA1,... References are needed for these prior findings.

Line 134: delete "centipede": "several peptide toxins..."

Line 140: delete "so far"

Line 145: remove "Glenn": "by King et al."

Line 158-159: change "peptide" to "peptide isomers" or just isomers" as these are the same peptide in its different forms

Line 187: insert "higher": "10-fold higher potency for"

Line 235: delete "just"

Line 269-271: Please remove "chemical stability" and possibly also "thermal stability" unless this was tested in CD experiments". Although it is a fair assumption that the disulfides provide chemical stability there is not data supporting this and the disulfides may merely serve to provide the correct fold needed for activity rather than stability.

Line 271-272: Clarify that this statement refers to other, disulfide-containing toxins. This is not clear as there is no reference provided here.

Line 534-537: Please revise grammar.

Response: We thank you for your thorough reading of the manuscript. We have corrected in the reviewed manuscript.

Referee #3:

In the revised manuscript, the authors have conducted a comprehensive series of experiments and analyses to address the reviewers' concerns and substantiate their conclusions. The results are thorough and effectively demonstrate the binding and modulatory mechanisms, providing insight into the interaction between the toxin and Nav channels. Furthermore, the new MD simulation data offer strong support for the proposed hypotheses regarding the potential dual actions of the toxin. Based on these revisions, I have no further concerns.

Response: We greatly appreciate the reviewer for the positive comment.

Dear Prof. Liu,

I am pleased to inform you that your manuscript has been accepted for publication in the EMBO Journal. Congratulations!

Yours sincerely,

Yehu Moran
Academic Editor
The EMBO Journal
